# GENERALIZATION BOUNDS FOR FEDERATED LEARNING: FAST RATES, UNPARTICIPATING CLIENTS AND UNBOUNDED LOSSES

**Xiaolin Hu[1,2], Shaojie Li[1,2], Yong Liu[1,2,*]**
[1]Gaoling School of Artificial Intelligence, Renmin University of China, Beijing, China
[2]Beijing Key Laboratory of Big Data Management and Analysis Methods, Beijing, China
{xiaolinhu,lishaojie95,liuyonggsai}@ruc.edu.cn

## ABSTRACT

In federated learning, the underlying data distributions may be different across clients. This paper provides a theoretical analysis of generalization error of federated learning, which captures both heterogeneity and relatedness of the distributions. In particular, we assume that the heterogeneous distributions are sampled from a meta-distribution. In this two-level distribution framework, we characterize the generalization error not only for clients participating in the training but also for unparticipating clients. We first show that the generalization error for unparticipating clients can be bounded by participating generalization error and participating gap caused by clients' sampling. We further establish fast learning bounds of order $\mathcal{O}(\frac{1}{mn} + \frac{1}{m})$ for unparticipating clients, where $m$ is the number of clients and $n$ is the sample size at each client. To our knowledge, the obtained fast bounds are state-of-the-art in the two-level distribution framework. Moreover, previous theoretical results mostly require the loss function to be bounded. We derive convergence bounds of order $\mathcal{O}(\frac{1}{\sqrt{mn}} + \frac{1}{\sqrt{m}})$ under unbounded assumptions, including sub-exponential and sub-Weibull losses.

## 1 INTRODUCTION

In federated learning, a common model is trained based on the collaboration of the participating clients holding local data samples (McMahan et al., 2017). Typically, the underlying distributions vary across clients since the data-generating processes are affected by the local environment. Federated learning is heterogeneous in the scenario where local distributions are different (Wang et al., 2021). Most existing experimental and theoretical results focus on the convergence of optimization on training datasets (Li et al., 2020b; Karimireddy et al., 2020; Mitra et al., 2021; Mishchenko et al., 2022; Yun et al., 2022). The generalization error, which is more natural and important in machine leanring, seems not to have been carefully examined in heterogeneous federated learning.

As a key performance indicator of the machine learning model, generalization error measures the performance of a trained model by its population risk with the corresponding distribution. However, existing generalization results are generally derived for clients participating in the training, which only captures the performance of the learned model on seen distributions during training (Mohri et al., 2019; Chen et al., 2021; Masiha et al., 2021).

In practice, the probability that a client participates in the federated training is affected by many factors such as the reliability of network connections or the availability of the client. The realistic participation ratio may be slow and a variety of clients never have a chance to participate during the training process (Kairouz et al., 2021; Li et al., 2020a; Yuan et al., 2021). Though the training process is operated only on participating clients, the trained model will be used by both unparticipating and participating clients. Since the data distributions of unparticipating clients are different from that of participating clients, it is natural and emergent to ask the following question:

**Would the unparticipating clients benefit from the model trained by participating clients?**

---

[*]Corresponding Author.

To answer this question theoretically, we take the participation gap into account in the analysis of generalization error, which is generally ignored by existing works.

In addition to the ignored participating gap, existing theoretical results on the generalization error of heterogeneous federated learning have two more limitations to our knowledge. First, all previous learning rates in probability form are of the order $\mathcal{O}(\frac{1}{\sqrt{mn}})$, where $m$ is the number of clients and $n$ is the sample size at each client (Mohri et al., 2019). We note that faster rates of order $\mathcal{O}(\frac{1}{mn})$ are derived in (Chen et al., 2021). However, their learning rates are in expectation form. Faster learning rates in probability form haven't been derived even only for participating clients. The guarantees in-expectation form reflect the average performance of the model trained based on the randomly sampled datasets. The theoretical bounds in probability form, which we focus on in this paper, reflect the performance of a single sampling on datasets (Klochkov & Zhivotovskiy, 2021; Kanade et al., 2022; Sefidgaran et al., 2022a). Second, most previous generalization bounds are derived by assuming that the loss function is bounded. However, there are a variety of learning problems that do not satisfy this assumption. This includes regression problems where unbounded noise is added to labels (Kuchibhotla & Patra, 2022; Kuchibhotla & Chakrabortty, 2018; Zhang & Zhou, 2018), clustering tasks with heavy-tailed distribution (Paul et al., 2021; Vellal et al., 2022), domain adaptation, and so on. Notable exception works in this direction include (Barnes et al., 2022) and (Sefidgaran et al., 2022b). However, their results are established under the assumption that local clients are homogeneous, which is highly restrictive in the general federated scenario.

In this paper, we assume that data distributions of participating and unparticipating clients are drawn from a meta-distribution $P$. We argue that this assumption is reasonable in practice. For instance, in cross-device federated learning, the number of total clients is generally large and it is natural to assume that there exists a meta-distribution (Reisizadeh et al., 2020; Wang et al., 2021). In this learning scenario, we assume that the total number of clients is $M$. Among all these $M$ clients, only $m$ clients have a chance to participate in the training phase, which means that the training process only involves the $m$ distributions $\{D_i\}_{i=1}^m$. Note that the total number $M$ and the number of unparticipating clients/distributions is generally larger than $m$ (Hu et al., 2022; Xu & Wang, 2020; Yang et al., 2020). Practically, the model is trained based on datasets $\{S_i\}_{i=1}^m$, where $S_i$ is the dataset located in client $i$ and is sampled from $D_i$. This two-level framework not only captures the heterogeneity of clients' distributions but also reflects the relatedness of the distributions. Thanks to this framework, we are allowed to characterize the generalization performance of both participating distributions and unparticipating distributions. A similar framework has been used by recent literature (Yuan et al., 2021; Reisizadeh et al., 2020; Wang et al., 2021). However, these works mainly focus on the optimization performance or only involve experimental results on the generalization. The objective of this work is to provide theoretical results on generalization error in this framework. Our contributions are summarized as follows.

- We provide a systematic analysis of the generalization error of federated learning in the two-level framework, which captures the missed participating gap in the existing works. This two-level framework captures both heterogeneity and relatedness of clients' distributions. Moreover, all learning bounds presented in this paper are in probability form instead of expectation form.

- We derive fast learning rates in the empirical risk minimization setting. The unparticipating error is bounded by two terms. One is participating error. The other is the participation gap results from missing clients in the training. Our participating bounds and unparticipating bounds are of order $\mathcal{O}(\frac{1}{mn})$ and $\mathcal{O}(\frac{1}{mn} + \frac{1}{m})$, respectively.

- We study the learning bounds for unbounded loss functions, including sub-gaussian, sub-exponential, and heavy-tailed losses. Small-ball methods and concentration inequalities for unbounded random variables are used in the unbounded setting. Our bounds are comparable with the existing results with bounded assumptions.

The rest of the paper is organized as follows. In Section 2, we describe the two-level distribution framework and provide basic theoretical results in this framework. In Section 3, we derive fast generalization bounds. In Section 4, we go beyond the bounded assumption and provide the generalization bounds for unbounded losses such as sub-exponential and sub-Weibull losses. In Section 5, we discuss related work on the generalization analysis of heterogeneous federated learning. Finally, we conclude this paper in Section 6. All proofs are postponed to the appendix.

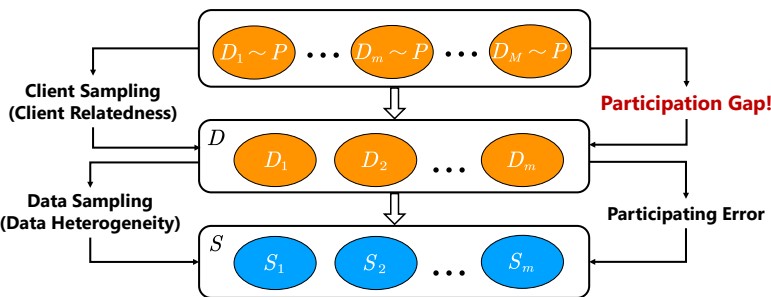

Figure 1: Illustration of the participation gap and participation error.

## 2 TWO-LEVEL DISTRIBUTION FRAMEWORK

Let $\mathcal{X}$ denote the input space and $\mathcal{Y} \subset \mathbb{R}$ the output space. For simplicity, we denote $Z = (X, Y)$ the random variable with support $\mathcal{Z} = \mathcal{X} \times \mathcal{Y}$. Let $\mathcal{D}$ denote the set of all probability distributions on $\mathcal{Z}$ and $P$ is a meta-distribution on $\mathcal{D}$. The assumption of meta-distribution is reasonable especially in cross-device federated learning scenario, where the local devices may be a large population of mobile phones. As shown in Figure 1, in this two-level distribution framework, we assume the total number of clients is $M$ (may be infinity) and the number of clients participating in training is $m$. It is worth emphasizing that $M$ is generally larger than $m$ owing to unreliable network connections. We denote by $D_i$ the distribution associated to client $i$ and assume $\{D_1, \cdots, D_m\}$ are independently sampled from $\mathcal{D}$ according to $P$. Data sample $S_i = \{Z_i^j\}_{j=1}^n$ located on participating client $i$ is made of $n$ i.i.d realizations of $Z$ following $D_i$. The global model is trained based on $\{S_i\}_{i=1}^m$ and will be used by all $M$ clients. Two-level distribution framework allows us to measure the performance of the global model with respect to clients' distribution $P$, which quantifies both the participation gap (caused by client sampling) and participating error (caused by data sampling from participating distributions). Throughout the paper we denote $\mathcal{F}$ by $\mathcal{F} = \{z \mapsto \ell(h, z) : h \in \mathcal{H}\}$. Moreover, we use $Z_i = (X_i, Y_i)$ to represent the random variables across two-level framework. That is, $\mathbb{E}[Z_i] = \mathbb{E}_{D_i \sim P}[\mathbb{E}_{Z_i \sim D_i}[Z_i]]$.

Let the hypothesis space $\mathcal{H}$ be a family of real-valued functions defined on $\mathcal{X}$. The loss function $\ell : \mathcal{Y} \times \mathcal{Y} \to \mathbb{R}^+$ is a non-negative function. We denote the population risk $\mathcal{L}_P(h)$ by $\mathcal{L}_P(h) = \mathbb{E}_{D_i \sim P}[\mathbb{E}_{Z \sim D_i}[\ell(h(X), Y)]]$, where $h \in \mathcal{H}$ represents the global hypothesis shared by all local clients. The population risk minimizer $h^*$ associated to population risk $\mathcal{L}_P(h)$ is define as $h^* = \arg\min_{h \in \mathcal{H}} \mathcal{L}_P(h)$. However, it is impossible to minimize $\mathcal{L}_P(h)$ directly because the exact meta distribution and client local distributions are unknown to us. We have access to only a finite number of clients and finite training data at each client. The global objective function defined as population risk is often optimized by the form of empirical risk minimization (ERM) objective function defined as: $\mathcal{L}_S(h) = \frac{1}{m} \sum_{i=1}^m \frac{1}{n} \sum_{j=1}^n \ell(h(X_i^j), Y_i^j)$, where $(X_i^j, Y_i^j)$ represents the $j$-th training data point at $i$-th participating client. For simplicity, we denote $Z_i^j = (X_i^j, Y_i^j)$ the data point. Let $S_i = \{Z_i^j\}_{j=1}^n$ denotes the local training set at $i$-th participating client and $S = S_i \bigcup \cdots \bigcup S_m$ represent the global training set across all participating clients. The empirical risk minimizer $\widehat{h}$ condition on dataset $S$ is define as $\widehat{h} = \arg\min_{h \in \mathcal{H}} \mathcal{L}_S(h)$. To analyze the generalization in our two-level framework, we further define semi-empirical distribution $D$ and the corresponding semi-empirical risk $\mathcal{L}_D(h)$ by $D = \frac{1}{m} \sum_{i=1}^m D_i$ and $\mathcal{L}_D(h) = \frac{1}{m} \sum_{i=1}^m \mathbb{E}_{Z \sim D_i}[\ell(h, Z)]$. We extend the previous definitions and denote by $\widehat{h}^*$ the semi-empirical risk minimizer $\widehat{h}^* = \arg\min_{h \in \mathcal{H}} \mathcal{L}_D(h)$.

The semi-excess risk for participating clients is defined as: $\mathcal{L}_D(\widehat{h}) - \mathcal{L}_D(\widehat{h}^*)$. Semi-excess risk indicates the performance of the learned model $\widehat{h}$ on the unseen data associated with semi-empirical distribution $D$. The excess risk for unparticipating clients is defined as: $\mathcal{L}_P(\widehat{h}) - \mathcal{L}_P(h^*)$. Excess risk indicates the performance of the learned model $\widehat{h}$ on the unseen clients distributed according to $P$. Note that the the excess risk $\mathcal{L}_P(\widehat{h}) - \mathcal{L}_P(h^*)$ is defined across two-level distribution framework. It will be shown that, in our analysis, all upper bound of excess risk $\mathcal{L}_P(\widehat{h}) - \mathcal{L}_P(h^*)$ involves semi-

excess risk $\mathcal{L}_D(\widehat{h}) - \mathcal{L}_D(\widehat{h}^*)$ or its upper bound. To understand this framework better, we present our basic results of excess risk as follows.

**Definition 1** (VC dimension). *Let $(\mathcal{X}, \mathcal{H})$ be a set system that consists of a set and a class H of subsets of X. A set system $(\mathcal{X}, \mathcal{H})$ shatters a set A if each subset of A can be expressed as $A \cap h$ for some h in $\mathcal{H}$. The VC-dimension of $\mathcal{H}$ is the size of the largest set shattered by $\mathcal{H}$.*

**Definition 2** (VC subgraph of real valued function). *The subgraph of a function $h(\in \mathcal{H}) : \mathcal{X} \to \mathbb{R}$ is the subset of $\mathcal{X} \times \mathbb{R}$ given by $\{(x, t) : t < h(x)\}$. Then the $VC$-dimension of the function class $\mathcal{F}$ is defined as the $VC$-dimension of the set of subgraphs of functions in $\mathcal{H}$.*

**Theorem 1** (Generalization error for unparticipating clients). *Let $\mathcal{F}$ be a family of functions related to hypothesis space $\mathcal{H} : \mathcal{F} = \{z \mapsto \ell(h, z) : h \in \mathcal{H}\}$. For the VC subgraph class $\mathcal{F}$ with VC dimension $d$. If the loss function $\ell$ is bounded by $b$, it follows that with probability at least $1 - 2\delta$,*

$$\mathcal{L}_P(\widehat{h}) - \mathcal{L}_P(h^*) \leq c_1 b \sqrt{\frac{d}{mn}} + b\sqrt{\frac{\ln(1/\delta)}{2mn}} + c_2 b \sqrt{\frac{d}{m}} + b\sqrt{\frac{\ln(1/\delta)}{2m}},$$

*where $c_1$ and $c_2$ are constants.*

**Remark 1.** *Assume the total number of clients is $M$ and $P$ is a concrete meta-distribution on $M$ different clients' distributions. The global model $\widehat{h}$ is trained based on $m$ participating clients. The excess risk measures the average performance of $\widehat{h}$ on total $M$ clients, which include participating and unparticipating clients. Theorem 1 indicates that, increasing the number of participating clients $m$ leads to the decrease of excess risk $\mathcal{L}_P(\widehat{h}) - \mathcal{L}_P(h^*)$. In cross-device federated learning, the number of participating clients $m$ may be large enough such that the excess risk approaches zero. Based on these discussions, we can give a positive answer to the question asked in Introduction. This is, from the perspective of average performance, unparticipating clients would benefit from the model trained by participating clients.*

**Remark 2.** *Our theoretical results show that under some assumptions we are able to bound the excess risk $\mathcal{L}_P(\widehat{h}) - \mathcal{L}_P(h^*)$. In the cases when every client is completely different, $\mathcal{L}_P(h^*)$ will be large. Thus, the generalization error for unparticipating clients $\mathcal{L}_P(\widehat{h})$ will be large. This observation indicates that, we can not expect one commen model works well when heterogeneity is high. We provide experimental results on EMINIST (Cohen et al., 2017) and synthetic data in appendix. Though Theorem 1 is derived for VC class, experiments of neural network justify our theory.*

## 3  FAST LEARNING RATES IN TWO-LEVEL DISTRIBUTION FRAMEWORK

In this section, we present fast learning rates in our two-level distribution framework. Recall that $\widehat{h}$ is the empirical risk minimizer and $h^*$ is the population risk minimizer. Our goal is to bound the semi-excess risk for participating clients $\mathcal{L}_D(\widehat{h}) - \mathcal{L}_D(\widehat{h}^*)$ and excess risk for unparticipating clients $\mathcal{L}_P(\widehat{h}) - \mathcal{L}_P(h^*)$. To get faster learning rates in our two-level distribution framework, we start by making some assumptions on loss function $\ell$, hypothesis space $\mathcal{H}$, semi-empirical distribution $D$, and meta distribution $P$.

**Assumption 1.** *Loss function $\ell$ is L-Lipschitz in its first argument: $|\ell(y_1, y) - \ell(y_2, y)| \leq L|y_1 - y_2|$.*

**Definition 3** (Bernstein condition). *Let $\mu$ be a distribution supported on $\mathcal{X} \times \mathcal{Y}$ and let $\ell$ be a loss function with domain $\mathcal{Y} \times \mathcal{Y}$. The tuple $(\mu, \ell, \mathcal{H}, h^*)$ satisfies the $(\beta, B)$-Bernstein condition with parameter $B > 0$ if the following holds for any $h \in \mathcal{H}$:*

$$\mathbb{E}\left(h(X) - h^*(X)\right)^2 \leq B\mathbb{E}\left[\ell(h(X), Y) - \ell(h^*(X), Y)\right]^\beta.$$

It is well known that fast learning rates require extra assumptions. Bernstein condition is widely used to get fast learning rates in the learning theory community (Xu & Zeevi, 2020; van Erven et al., 2015; Wu et al., 2022). We emphasize that it is not too restrictive. For example, it is directly implied by the boundedness property of functions with any probability distribution (Bartlett et al., 2004). Moreover, regression problems with strictly convex loss function satisfy the Bernstein condition if the function class is convex (Lecué & Mendelson, 2013). Other examples include excess risk functions with minimizer of population risk when the loss function is strongly convex and Lipschitz (Klochkov & Zhivotovskiy, 2021).

**Assumption 2.** *Theoretical analyses in our two-level distribution framework involve different types of Bernstein conditons:*

(a) *The tuple $(D, \ell, \mathcal{H}, \widehat{h}^*)$ satisfies the Bernstein condition with parameter $B' \geq 1, 0 < \beta' \leq 1$. That is, for any $h \in \mathcal{H}$, $\frac{1}{m} \sum_{i=1}^{m} \mathbb{E}[h(X_i^1) - \widehat{h}^*(X_i^1)]^2 \leq B'(\mathcal{L}_D(h) - \mathcal{L}_D(\widehat{h}^*))^{\beta'}$.*

(b) *The tuple $(P, \ell, \mathcal{H}, h^*)$ satisfies the Bernstein condition with parameter $B'' \geq 1, 0 < \beta'' \leq 1$. That is, for any $h \in \mathcal{H}$, $\mathbb{E}_{D_i \sim P}[\mathbb{E}_{X \sim D_i}[h(X) - h^*(X)]^2] \leq B''(\mathcal{L}_P(h) - \mathcal{L}_P(h^*))^{\beta''}$.*

For our purposes, we need to check that both (a) and (b) in Assumption 2 hold. A typical example satisfying Assumption 2 is quadratic loss with convex function class $\mathcal{H}$. We provide some examples satisfying Assumption 2 in appendix. For more details, we refer to (Xu & Zeevi, 2020; Wu et al., 2022; van Erven et al., 2015).

**Assumption 3** (Uniform entropy number[1]). *Let $\mathcal{H}$ be a family of bounded functions with uniformly entropy number $\log \mathcal{N}(\epsilon, \mathcal{H}, \|\cdot\|_2)$. Assume that there exist positive numbers $\gamma, d$ and $p$ such that $\log \mathcal{N}(\epsilon, \mathcal{H}, \|\cdot\|_2) \leq d \log^p(\gamma/\epsilon)$ for any $0 < \epsilon \leq \gamma$.*

Assumption 3 is a mild assumption if the function classes are bounded. We list some popular function classes satisfying Assumption 3: (a) If the VC-dimension of $\mathcal{H}$ is finite, then $\mathcal{H}$ satisfies assumption 3. For instance, the function class of $k$-means methods has finite VC dimension. For more details, we refer the reader to (Devroye et al., 2013). (b) When we set $\epsilon \in (0, 1)$, then all the unit Euclidean ball $\mathcal{B} \subset \mathbb{R}^d$ satisfy assumption 3. (c) If $\mathcal{H}$ is a RKHS with kernel $k$ and the rank of $k$ is $d$, then $\mathcal{H}$ satisfies assumption 3.

### 3.1 FAST LEARNING RATES FOR PARTICIPATING CLIENTS

In this subsection, we provide fast learning rates for participating clients in high probability. To obtain faster convergence rates, we focus on semi-excess risk $\mathcal{L}_D(\widehat{h}) - \mathcal{L}_D(\widehat{h}^*)$.

**Theorem 2** (Semi-excess risk for participating clients). *Let $\mathcal{F}$ be a family of functions bounded by $b$. Under assumptions 1, 3 and (a) of Assumption 2, when $mn \geq cd \log^p(mn)$, it follows that with probability at least $1 - \delta$,*

$$\mathcal{L}_D(\widehat{h}) - \mathcal{L}_D(\widehat{h}^*) \leq c_1 \left(\frac{\log^p(mn)}{mn}\right)^{\frac{1}{2-\beta'}} + c_2 \left(\frac{\log(1/\delta)}{mn}\right)^{\frac{1}{2-\beta'}},$$

*where $c_1$ and $c_2$ are constants depending on $\gamma, p, L, \beta'$ and $B_1, b, \beta'$ respectively.*

**Remark 3.** *Theorem 2 shows that the convergence rate of semi-empirical excess risk ranges from $\mathcal{O}(\frac{1}{\sqrt{mn}})$ to faster order $\mathcal{O}(\frac{1}{mn})$, which corresponds to $\beta' = 0$ and $\beta' = 1$, respectively. It indicates that, under Bernstein condition, semi-empirical excess risk convergences faster when we increase the number of clients $m$ or the size $n$ of local dataset. We emphasize that our bounds in Theorem 2 is in high probability form, which is more emergent and challenging, when compared to the previous results in expectation form (Chen et al., 2021; Fallah et al., 2021). The learning bounds in Theorem 2 are conducted for the empirical risk minimizer $\widehat{h}$. For the inexact minimizer of $\widehat{h}$, the proof technique and the final bounds only involve an extra optimization term. For more details about the optimization error, we refer to (Wang et al., 2021; Su et al., 2021; Khaled et al., 2019).*

### 3.2 FAST LEARNING RATES FOR UNPARTICIPATING CLIENTS

In this subsection we provide fast learning rates for **unparticipating clients** in high probability. To the best of our knowledge, this is the first result derived for unparticipating clients in heterogeneous federated learning.

**Theorem 3.** *Let $\mathcal{F}$ be a family of functions bounded by $b$. Under assumptions 1, 3 and (b) of Assumption 2, when $m \geq cd \log^p(m)$, for any $\delta > 0$, it follows that with probability at least $1 - \delta$,*

$$\mathcal{L}_P(\widehat{h}) - \mathcal{L}_P(h^*) \leq c_0 \left(\mathcal{L}_D(\widehat{h}) - \mathcal{L}_D(\widehat{h}^*)\right) + c_1 \left(\frac{\log^p m}{m}\right)^{\frac{1}{2-\beta''}} + c_2 \left(\frac{\log(1/\delta)}{m}\right)^{\frac{1}{2-\beta''}},$$

*where $c_0 = \frac{K}{K-\beta''}$, $c_1$ and $c_2$ are constants depending on $\gamma, p, L, \beta''$ and $B_2, b, \beta''$ respectively.*

---

[1]The definition of uniform entropy number is provided in appendix C.

**Remark 4.** *Theorem 3 is developed across the two-level distribution framework, which brings extra challenges to the analysis. It is shown that the upper bound derived in 3 include semi-empirical excess risk term $\mathcal{L}_D(\widehat{h}) - \mathcal{L}_D(\widehat{h}^*)$, which is an outcome of excess risk decomposition across two-level framework. Recall that $\beta'$ and $\beta''$ are constants defined in Assumption 2. In the cases where $\beta' = 1$ and $\beta'' = 1$, it can be shown that excess risk is of order $\mathcal{O}(\frac{1}{mn} + \frac{1}{m})$ with high probability. To present Theorem 3, we must construct a sub-root function that links the expected local Rademacher complexity associated with meta-distribution $P$ and uniform entropy number. However, the conventional Dudley's integral bounds are built under empirical constraints (Boucheron et al., 2013). We tackle this challenge by extending the techniques developed in (Lei et al., 2016) to our two-level distribution framework.*

## 4 LEARNING RATES FOR SUB-WEIBULL LOSSES

In this section, we provide generalization error bounds for unbounded losses in two-level distribution framework. In particular, we focus on loss functions satisfying sub-Weibull condition.

**Definition 4** (Sub-Weibull random variables). *A random variable $X$ is said to be sub-Weibull if there is constant $\|X\|_{\psi_\alpha} < \infty$, such that*

$$\mathbb{P}(|X| \geq t) \leq 2\exp(-t^\alpha/\|X\|_{\psi_\alpha}^\alpha), \text{ for all } t \geq 0.$$

*Sub-Gaussian and sub-exponential random variables are two special cases of Sub-Weibull random variables, which correspond to $\alpha = 2$ and $\alpha = 1$, respectively.*

The learning rates derived in two-level framework for sub-exponential losses are deferred to the appendices. In the following we use small-ball method to establish learning rates for more heavy-tailed losses, where two-side concentration inequalities may fail to hold.

This subsection aims at establishing generalization bounds for unbounded losses that have heavier tails than sub-exponential distribution. Since the two side inequalities for empirical process fail to hold when the losses are heavy-tailed, the analysis of heavy-tailed losses require new method to relate empirical risk and population risk. In this subsection, we establish generalization bounds for heterogeneous federated learning by extending the small-ball method from i.i.d setting to our two-level distribution framework. We consider the quadratic loss function in this section. The extension to general losses can be achived by using the techniques presented in (Mendelson, 2018).

In what follows, we denote by $\|h\|_{L_2(\mu)}$ for Banach spaces $L_2(\mathcal{X}, \mu)$. Recall that $D$ is the semi-empirical distribution and $P$ is meta-distribution. In particular, we have $\|h\|_{L_2(D)} = (\frac{1}{m}\sum_{i=1}^m \mathbb{E}_{X \sim D_i}[h(X)]^2)^{1/2}$ and $\|h\|_{L_2(P)} = (\mathbb{E}_{D_i \sim P}\mathbb{E}_{X \sim D_i}[h(X)]^2)^{1/2}$. For the sake of clear exposition, we first introduce the small-ball condition.

**Assumption 4** (Small-ball condition). *Let $\mathcal{H} \subset L_2(D)$ be a closed and convex class of functions and $\mathcal{H} - \mathcal{H} := \{h - h' : h, h' \in \mathcal{H}\}$.*

    (a) *Let $Q_{mn}(\tau) = \inf_{h \in \mathcal{H} - \mathcal{H}} \mathbb{P}(|h(X_i^1)| \geq \tau\|h\|_{L_2(D)})$, where $X_i^1$ represent the random sample at $i$-th participating client. There is a $\tau \geq 0$ for which $Q_{mn}(\tau) > 0$.*

    (b) *Let $Q_m(\tau) = \inf_{h \in \mathcal{H} - \mathcal{H}} \mathbb{P}\left(|\mathbb{E}[h(X_i^1)]| \geq \tau\|h\|_{L_2(P)}\right)$, where $X_i^1$ represent the random sample at $i$-th participating client. There is a $\tau \geq 0$ for which $Q_m(\tau, P) > 0$.*

Assumption 4, small-ball condition, has been assumed for i.i.d and dependent data-generating process. To obtain high-probability theoretical guarantees, concentration techniques are widely used in the analysis of generalization error (Boucheron et al., 2013). Intuitively, empirical risk will concentrate around population risk with high probability only when the loss function has well-behaved moments. However, this condition may fail to hold for heavy-tailed losses (Mendelson, 2015). Assumption 4 appears first in the work of (Mendelson, 2015). Losses with any sort of moment equivalence satisfy small-ball condition, which is weaker than concentration condition and can be used to model heavy-tailedness. For example, even weak condition $\|h\|_{L_2(P)} \leq c\|h\|_{L_1(P)}$ yields nontrivial small-ball estimate. Moreover, the equivalence between higher-order moments and second-order moment such as $\|h\|_{L_p(P)} \leq c\|h\|_{L_2(P)}$ also leads to small-ball condition (Lecué & Mendelson, 2016). Based on these observations, condition (b) of Assumption 4 is generally implied when we

consider each local distribution $D_i$ as a random variable according to client distribution $P$. Let us discuss condition (a) of Assumption 4 in more detail. Note that the establishment of this assumption 4 only requires $\mathcal{H} \subset L_2(D)$ with high probability, where $D$ is the semi-empirical distribution. This requirement is not too restrictive since the elements of $D$ are i.i.d sampled from $P$. To our knowledge, this is the first time that small-ball condition is used under heterogeneous data generating assumption.

## 4.1 LEARNING RATES FOR PARTICIPATING CLIENTS WITH SMALL-BALL CONDITION

We first describe the basic idea of generalization analysis for participating clients. Recall that $\widehat{h}^*$ is the minimizer of semi-empirical risk $\mathcal{L}_D(h)$ in $\mathcal{H}$. In this subsection we focus on the measure $\|h - \widehat{h}^*\|_{L_2(D)}^2$, which represents the distance between $h$ and $\widehat{h}^*$ with respect to semi-empirical distribution $D$. For quadratic loss and every $h \in \mathcal{H}$, we have

$$\mathcal{L}_S(h) - \mathcal{L}_S(\widehat{h}^*) = \frac{1}{mn} \sum_{i=1}^{m} \sum_{j=1}^{n} \left[ (h(X_i^j) - Y_i^j)^2 - (\widehat{h}^*(X_i^j) - Y_i^j)^2 \right] \tag{1}$$

$$= \frac{1}{mn} \sum_{i=1}^{m} \sum_{j=1}^{n} (h - \widehat{h}^*)^2 (X_i^j) + \frac{2}{mn} \sum_{i=1}^{m} \sum_{j=1}^{n} \xi_i^j (h - \widehat{h}^*)(X_i^j), \tag{2}$$

where $\xi_i^j = \widehat{h}^*(X_i^j) - Y_i^j$. Since $\widehat{h}$ is the minimizer of empirical risk $\mathcal{L}_S(h)$, we have $\mathcal{L}_S(\widehat{h}) - \mathcal{L}_S(\widehat{h}^*) \leq 0$. If on an event $\|h - \widehat{h}^*\|_{L_2(D)}$ is large, then the summation of two terms in (2) is larger than 0 with high probability. It follows that with high probability $\|\widehat{h} - \widehat{h}^*\|_{L_2(D)}$ is small since $\mathcal{L}_S(\widehat{h}) - \mathcal{L}_S(\widehat{h}^*) \leq 0$.

Let $\{(X_i^j, Y_i^j)\}_{(i,j)=(1,1)}^{(m,n)}$ be global data samples whose elements $\{(X_i^j, Y_i^j)\}_{j=1}^{n}$ are i.i.d random pairs at $i$-th client. The analysis of the first term in (2) involves the following definition of Rademacher complexity.

**Definition 5.** *We define $\mathcal{H} - \mathcal{H} = \{h - h' : h, h' \in \mathcal{H}\}$ and denote by $B_2^m$ the $L_2(D)$ unit ball entered at $\widehat{h}^*$, that is $B_2^m = \{h \in \mathcal{H} : \|h - \widehat{h}^*\|_{L_2(D)} \leq 1\}$. For every $\eta > 0$, define*

$$\omega_{mn}(\eta) := \inf \left\{ s > 0 : \mathbb{E}\left[ \sup_{h \in (\mathcal{H} - \mathcal{H}) \cap sB_2^m} \left| \frac{1}{mn} \sum_{i=1}^{m} \sum_{j=1}^{n} \sigma_i^j h(X_i^j) \right| \right] \leq \eta s \right\},$$

*where $\sigma_i^j$ are Rademacher random variables.*

The quantity $\omega_{mn}(\eta)$ measures the Rademcher complexity of the localized function set $\{\mathcal{H} - \mathcal{H} \cap sB_2^m\}$. Note that $\omega_{mn}(\eta)$ depends only on the hypothesis class $\mathcal{H}$ and global input samples are drawn from semi-empirical distribution $D$.

**Theorem 4.** *Fix $\tau > 0$ for which $Q_m(2\tau) > 0$ and set $\eta < \tau^2 Q_{mn}(2\tau)/32$. If every random variable $V_i^j = \xi_i^j h(X_i^j) - \mathbb{E}[\xi_i^j h(X_i^j)]$ for all $h \in \mathcal{H} - \widehat{h}^*$ is Sub-Weibull. For sufficiently large $mn$, with probability at least $1 - \delta_{mn} - \exp\left(-mn Q_{mn}^2(2\tau)/2\right)$ one has*

$$\|\widehat{h} - \widehat{h}^*\|_{L_2(D)} \leq 2 \max \left\{ \omega_{mn}(\tau Q_{mn}(2\tau)/16), (mn)^{-\frac{1}{4}+\iota} \right\},$$

*where $0 < \iota < \frac{1}{4}$ and $\delta_{mn} = \exp\{-(\frac{c_1 \eta^2 (mn)^{4\iota}}{\frac{1}{mn} \sum_{i=1}^{m} \sum_{j=1}^{n} \|V_i^j\|_{\psi_\alpha}^2} \wedge \frac{c_2 \eta^\alpha (mn)^{\alpha(1/2+2\iota)}}{\max_{(1,1) \leq (i,j) \leq (m,n)} \|V_i^j\|_{\psi_\alpha}^\alpha})\}$.*

**Remark 5.** *To the best of our knowledge, Theorem 4 provides the first result on the generalization error of heterogeneous federated learning with heavy-tailed losses. It suggests that both hypothesis 'size' and noise level play important roles in the generalization error of heterogeneous learning problems. In Theorem 4, the expression of $\delta_{mn}$ dependents on the tail of sub-Weibull random variables $V_i^j$. Specifically, the heavier the tail of $V_i^j$, the larger $\delta_{mn}$ will be. Note that in addition to $\|V_i^j\|_{\psi_\alpha}$, $\delta_{mn}$ also depends on $(mn)^{4\iota}$. That is, the larger $mn$ and $\iota$ are, the smaller $\delta_{mn}$ will be. To ensure that Theorem 4 holds with high probability, we must ensure that $\delta_{mn}$ is small enough. Therefore, when $mn$ is fixed, the heavier the tail of $V_i^j$, the larger $\iota$ should be. It can be seen from Theorem 4 that the larger $\iota$ is, the slower the convergence rate of $\|\widehat{h} - \widehat{h}^*\|_{L_2(D)}$ is.*

**Corollary 1.** *Under the same conditions of Theorem 4, for convex function class $\mathcal{H}$ and sufficiently large $mn$, with probability at least $1 - \delta_{mn} - \exp\left(-mnQ_{mn}^2(2\tau)/2\right)$ one has*

$$\mathcal{L}_D(\widehat{h}) - \mathcal{L}_D(\widehat{h}^*) \leq (2 + \frac{\tau^2}{4}Q_{mn}(2\tau))\max\left(\omega_{mn}^2(\tau Q_{mn}(2\tau)/16), (mn)^{-\frac{1}{2}+\iota}\right),$$

*where $0 < \iota < \frac{1}{2}$.*

**Remark 6.** *Corollary 1 provides the convergence rate of semi-empirical excess risk for Sub-Weibull losses. Compared to Theorem 2, it shows that the convergence rate of excess risk is slower than $\mathcal{O}(\frac{1}{\sqrt{mn}})$.*

## 4.2 LEARNING RATES FOR UNPARTICIPATING CLIENTS WITH SMALL-BALL CONDITION

In the analysis of generalization error for unparticipating clients, we focus on the measure $\|h - h^*\|_{L_2(P)}^2$, which represents the distance between $h$ and $h^*$ with respect to meta-distribution $P$. The analysis of generalization error for unparticipating clients follows a similar path to the previous analysis. Let $\{(X_i, Y_i)\}_{i=1}^m$ be dataset whose elements are sampled across the two-level framework, that is $\mathbb{E}[(X_i, Y_i)] = \mathbb{E}_{D_i \sim P}\mathbb{E}_{(X_i, Y_i) \sim D_i}[(X_i, Y_i)]$. We present the different definitions of Rademacher complexity terms in the following.

**Definition 6.** *We define $\mathcal{H} - \mathcal{H} = \{h - h' : h, h' \in \mathcal{H}\}$ and denote by $B_2$ the $L_2(P)$ unit ball entered at $h^*$. For every $\eta > 0$, define*

$$\omega_m(\eta) := \inf\left\{s > 0 : \mathbb{E}\left[\sup_{h \in (\mathcal{H}-\mathcal{H}) \cap sB_2}\left|\frac{1}{m}\sum_{i=1}^m \sigma_i h(X_i)\right|\right] \leq \eta s, \right\}$$

*where $\sigma_i$ are Rademacher random variables.*

The quantity $\omega_m(\eta)$ measures the localized complexity of $\{(\mathcal{H} - \mathcal{H}) \cap sB_2\}$.

**Theorem 5.** *Fix $\tau > 0$ for which $Q_m(2\tau) > 0$ and set $\eta < \tau^2 Q_m(2\tau)/32$. If for all $h \in \mathcal{H} - h^*$ the random variable $V_i = \mathbb{E}[\xi_i^1 h(X_i^1)] - \mathbb{E}[\xi_i h(X_i)]$ is Sub-Weibull. For sufficiently large $m$, with probability at least $1 - \delta_m - \exp\left(-mQ_m^2(2\tau)/2\right)$ one has*

$$\|\widehat{h}^* - h^*\|_{L_2(P)} \leq 2\max\left\{\omega_m(\tau Q_m(2\tau)/16), m^{-\frac{1}{4}+\iota}\right\},$$

*where $0 < \iota < \frac{1}{4}$ and $\delta_m = \exp\{-(\frac{c_1\eta^2 m^{4\iota}}{\frac{1}{m}\sum_{i=1}^m \|V_i\|_{\psi_\alpha}^2} \wedge \frac{c_2\eta^\alpha m^{\alpha(1/2+2\iota)}}{\max_{1 \leq i \leq m}\|V_i\|_{\psi_\alpha}^\alpha})\}$.*

**Remark 7.** *Theorem 5 provides the first result on the generalization error of unparticipating clients in heterogeneous federated learning with heavy-tailed losses.*

**Corollary 2.** *Assume for all $h \in \mathcal{H} - \widehat{h}^*$ the random variable $V_i' = \mathbb{E}[h^2(X_i^j)] - \mathbb{E}[h^2(X_i)]$ is Sub-Weibull and the noise $h^*(X_i) - Y_i$ is independent of $X_i$. Under the same conditions of Theorem 5, for $0 < \eta < 1$ and sufficiently large $mn$, with probability at least $1 - \delta' - \exp\left(-mnQ_{mn}^2(2\tau)/2\right) - \exp\left(-mQ_m^2(2\tau)/2\right)$ one has*

$$\mathcal{L}_P(\widehat{h}) - \mathcal{L}_P(h^*) \leq c_0\max\left(\omega_{mn}^2(\frac{\tau Q_{mn}(2\tau)}{16}), (mn)^{-\frac{1}{2}+\iota}\right) + 2\max\left\{\omega_m^2(\frac{\tau Q_m(2\tau)}{16}), m^{-\frac{1}{2}+\iota}\right\},$$

*where $c_0 = \frac{2}{1-\eta}, 0 < \iota < \frac{1}{2}$ and $\delta' = \delta_{mn} + \delta_m + \exp\{-(\frac{c_1\eta^2 m^{4\iota}}{\frac{1}{m}\sum_{i=1}^m \|V_i'\|_{\psi_\alpha}^2} \wedge \frac{c_2\eta^\alpha m^{\alpha(1/2+2\iota)}}{\max_{1 \leq i \leq m}\|V_i'\|_{\psi_\alpha}^\alpha})\}$.*

**Remark 8.** *Corollary 2 provides the convergence rate of excess risk for Sub-Weibull losses. Compared to Theorem 3, it shows that the convergence rate of excess risk is slower than $\mathcal{O}(\frac{1}{\sqrt{mn}} + \frac{1}{\sqrt{m}})$.*

## 5 RELATED WORK

**Generalization Error for Heterogeneous Federated Learning.** Several attempts have been made in the analysis of generalization error for heterogeneous federated learning. We compare our results with most related works in Table 1. Complexity-based bounds for participating clients are derived in the work of (Mohri et al., 2019), who present high probability slow rates of order $\mathcal{O}(\frac{1}{\sqrt{mn}})$

Table 1: **Generalization Bounds for Heterogeneous Federated Learning**. SC, Pro, and Exp denote Strong convexity, In probability, and In expectation. Sub-expon denotes sub-exponential.

| Reference | Loss | Assumption | Part | Unpart | Type |
|-----------|------|------------|------|--------|------|
| Mohri et al. (2019) | Bounded | Bi-Classifier | $\mathcal{O}(\frac{1}{\sqrt{mn}})$ | / | Pro |
| Chen et al. (2021) | Bounded | Smooth, SC | $\mathcal{O}(\frac{1}{mn})$ | / | Exp |
| Fallah et al. (2021) | Bounded | Smooth, SC | $\mathcal{O}(\frac{1}{mn})$ | / | Exp |
| Our Results | Sub-expon | Lipschitz | $\mathcal{O}(\frac{1}{\sqrt{mn}})$ | $\mathcal{O}(\frac{1}{\sqrt{mn}} + \frac{1}{\sqrt{m}})$ | Pro |
| Our Results | Bounded | Bernstein Con | $\mathcal{O}(\frac{1}{mn})$ | $\mathcal{O}(\frac{1}{mn} + \frac{1}{m})$ | Pro |
| Our Results | Sub-Weibull | Small-ball | $\mathcal{O}\left((mn)^{\frac{2\iota-1}{2}}\right)$ | $\mathcal{O}((mn)^{\frac{2\iota-1}{2}} + m^{\frac{2\iota-1}{2}})$ | Pro |

for bounded losses. Fast rates bounds are obtained based on stability tools in (Chen et al., 2021; Fallah et al., 2021). However, their results are in expectation form. Among the existing theoretical work, different measurements are used to model the heterogeneity of local distributions. These measurements include gradient dissimilarity and parameter dissimilarity of local optimal models. Here we argue that it is more natural to make an assumption from the perspective of the data-generating process. Therefore, in this paper, we assume that the local distributions are sampled from a higher meta-distribution.

A similar two-level distribution framework has been used in the analysis of meta-learning. However, the learning scenarios and objectives of federated learning are different from that of meta-learning. The goal of meta-learning is to choose an optimal hypothesis space $\mathcal{H}$ from the hypothesis space family $\mathbb{H}$. Ideally, the chosen hypothesis $\mathcal{H}$ should contain good hypothesis $h \in \mathcal{H}$ for each distribution $D_i$ sampled from the meta distribution $P$. In this paper, we focus on the performance of common model $\hat{h}$ trained by participating clients. The performance of the common model is measured by the population risk with respect to meta distribution $P$. Another line of research closely related to heterogeneous federated learning is domain adaptation/generalization. In this line, possibly the results in (Li et al., 2022) are most relevant to ours.

**Generalization error for Unbounded losses.** The unbounded assumption brings two major challenges to complexity-based generalization analysis. One is that the two-side concentration inequalities do not hold when the losses are heavy-tailed. The other is that the standard techniques used to upper bound the complexity of hypothesis space are developed for bounded losses. The straightforward way to avoid these two challenges is to assume there exists an envelope function with respect to the underlying distribution and hypothesis class (Adamczak, 2008; Lecué & Mendelson, 2012). Small-ball method is first proposed to replace the concentration tools for empirical process in (Mendelson, 2015) and further developed in (Mendelson, 2018). Inspired by the small-ball method, Offset Rademacher complexity-based method provides another replacement for two-side concentration inequality (Liang et al., 2015). However, most existing generalization bounds for unbounded losses are derived in the i.i.d setting. Roy et al. (2021) extend the small-ball method in the dependent data setting. In this paper, we focus on the heterogeneous federated learning scenario with unbounded losses, where the samples are independent but non-identically distributed.

## 6 CONCLUSION

We present a systematic generalization analysis of heterogeneous distributed learning. Our analysis captures the generalization performance of the learned model on both participating and unparticipating clients. To our knowledge, this is the first theoretical analysis under the assumption that the local distributions are sampled from a meta-distribution. We recover the current state of art guarantees without using bounded assumptions. Moreover, under the empirical risk minimization setting, we derive fast generalization rates in our two-level distribution setting.

## ACKNOWLEDGEMENTS

We appreciate all the anonymous reviewers for their insightful and constructive comments, especially one reviewer's suggestion to add more discussion and explanation. This work is supported by National Natural Science Foundation of China NO. 62076234; the Beijing Natural Science Foundation No. 4222029; the Intelligent Social Governance Interdisciplinary Platform, Major Innovation & Planning Interdisciplinary Platform for the "Double-First Class" Initiative, Renmin University of China; the Beijing Outstanding Young Scientist Program NO.BJJWZYJH012019100020098; the Beijing Key Laboratory of Big Data Management and Analysis Methods, Gaoling School of Artificial Intelligence, Renmin University of China, Beijing 100872, China; the Public Computing Cloud, Renmin University of China; the Fundamental Research Funds for the Central Universities, and the Research Funds of Renmin University of China NO. 2021030199; the Huawei-Renmin University joint program on Information Retrieval; the Unicom Innovation Ecological Cooperation Plan; and the CCF-Huawei Populus Grove Fund.

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

CONTENTS

# A    SETTING, DEFINITIONS

We denote the population risk $\mathcal{L}_P(h)$ by

$$\mathcal{L}_P(h) = \mathbb{E}_{D_i \sim P}\left[\mathbb{E}_{Z \sim D_i}\left[\ell(h(X), Y)\right]\right],$$

where $h \in \mathcal{H}$ represents the global hypothesis shared by all local clients, $\ell : \mathcal{Y} \times \mathcal{Y} \to \mathbb{R}^+$ is the non-negative loss function. The population risk minimizer $h^*$ is define as

$$h^* = \underset{h \in \mathcal{H}}{\arg\min}\, \mathcal{L}_P(h).$$

In practice, the global objective function is often optimized by the form of empirical risk minimization (ERM) objective function, which is defined as:

$$\mathcal{L}_S(h) = \frac{1}{m}\sum_{i=1}^{m}\frac{1}{n}\sum_{j=1}^{n}\ell(h(X_i^j), Y_i^j),$$

where $(X_i^j, Y_i^j)$ represents the $j$-th training data point at $i$-th participating client. We also use $Z_i^j = (X_i^j, Y_i^j)$ to denote the data point. Let $S_i = \{Z_i^j\}_{j=1}^n$ denotes the local training set at $i$-th participating client and $S = S_i \bigcup \cdots \bigcup S_m$ be the global training set. The empirical risk minimizer $\widehat{h}$ is define as

$$\widehat{h} = \underset{h \in \mathcal{H}}{\arg\min}\, \mathcal{L}_S(h).$$

The semi-empirical distribution $D$ is defined as $D = \frac{1}{m}\sum_{i=1}^{m} D_i$. Moreover, the corresponding semi-empirical risk $\mathcal{L}_D(h)$ is defined as:

$$\mathcal{L}_D(h) = \frac{1}{m}\sum_{i=1}^{m}\mathbb{E}_{Z \sim D_i}\left[\ell(h, Z)\right].$$

We denote by $\widehat{h}^*$ the semi-empirical risk minimizer

$$\widehat{h}^* = \underset{h \in \mathcal{H}}{\arg\min}\, \mathcal{L}_D(h).$$

The semi-excess risk for participating clients is defined as:

$$\mathcal{L}_D(\widehat{h}) - \mathcal{L}_D(\widehat{h}^*).$$

Semi-excess risk indicates the performance of the learned model $\widehat{h}$ on the participating clients.

The excess risk for unparticipating clients is defined as:

$$\mathcal{L}_P(\widehat{h}) - \mathcal{L}_P(h^*).$$

Excess risk indicates the performance of the learned model $\widehat{h}$ on the unparticipating clients.

In the following, we denote $\mathcal{F}$ by $\mathcal{F} = \{z \mapsto \ell(h, z) : h \in \mathcal{H}\}$. Moreover, we use $Z_i = (X_i, Y_i)$ to represent the random variables across two-level framework. That is,

$$\mathbb{E}[Z_i] = \mathbb{E}_{D_i \sim P}[\mathbb{E}_{Z_i \sim D_i}[Z_i]].$$

# B    PROOF OF THEOREM 1

## B.1    PROOF OF THEOREM 1

We first show that semi-excess risk $\mathcal{L}_D(\widehat{h}) - \mathcal{L}_D(h^*)$ can be upper bounded by supremum of the empirical process indexed by $\mathcal{H}$:

$$\mathcal{L}_D(\widehat{h}) - \mathcal{L}_D(\widehat{h}^*) \le \mathcal{L}_D(\widehat{h}) - \mathcal{L}_S(\widehat{h}) + \mathcal{L}_S(\widehat{h}^*) - \mathcal{L}_D(\widehat{h}^*) \le 2\sup_{h \in \mathcal{H}}|\mathcal{L}_D(h) - \mathcal{L}_S(h)|, \quad (3)$$

where the first inequality follows from the fact that $\widehat{h}$ is the empirical minimizer.

Next, we decompose excess risk $\mathcal{L}_P(\widehat{h}) - \mathcal{L}_P(h^*)$ in two-level framework:

$$\mathcal{L}_P(\widehat{h}) - \mathcal{L}_P(h^*) \leq 2 \sup_{h \in \mathcal{H}} |\mathcal{L}_P(h) - \mathcal{L}_D(h)| + 2 \sup_{h \in \mathcal{H}} |\mathcal{L}_D(h) - \mathcal{L}_S(h)|.$$

In details:

$$
\begin{aligned}
\mathcal{L}_P(\widehat{h}) - \mathcal{L}_P(h^*) &\leq \mathcal{L}_P(\widehat{h}) - \mathcal{L}_P(h^*) - (\mathcal{L}_S(\widehat{h}) - \mathcal{L}_S(h^*)) \\
&= \mathcal{L}_P(\widehat{h}) - \mathcal{L}_D(\widehat{h}) + \mathcal{L}_D(h^*) - \mathcal{L}_P(h^*) + \mathcal{L}_D(\widehat{h}) - \mathcal{L}_S(\widehat{h}) + \mathcal{L}_S(h^*) - \mathcal{L}_D(h^*) \\
&\leq 2 \sup_{h \in \mathcal{H}} \{|\mathcal{L}_P(h) - \mathcal{L}_D(h)|\} + 2 \sup_{h \in \mathcal{H}} \{|\mathcal{L}_D(h) - \mathcal{L}_S(h)|\},
\end{aligned}
$$

where the first inequality uses $\mathcal{L}_S(\widehat{h}) - \mathcal{L}_S(h^*) \leq 0$ since $\widehat{h}$ is the minimizer of $\mathcal{L}_S(h)$. Note that the first term $\sup_{h \in \mathcal{H}} |\mathcal{L}_P(h) - \mathcal{L}_D(h)|$ in the upper bound of excess risk quantifies the participation gap. The second term $\sup_{h \in \mathcal{H}} |\mathcal{L}_D(h) - \mathcal{L}_S(h)|$ quantifies the generalization error for participating clients.

Based on these observations, we first provide theoretical bounds on the term $\sup_{h \in \mathcal{H}} |\mathcal{L}_D(h) - \mathcal{L}_S(h)|$. Then we move to the participating gap $\sup_{h \in \mathcal{H}} |\mathcal{L}_P(h) - \mathcal{L}_D(h)|$.

Let $\boldsymbol{\sigma} = \{\sigma_i^j\}_{i \in [m], j \in [n]}$ be a collection of independent Rademacher variables, this is random variables taking values uniformly in $\{+1, -1\}$. We define the generalized rademacher complexity $\mathcal{R}_{mn}(\mathcal{F})$ used in heterogeneous federated learning as is as follows:

$$\mathcal{R}_{mn}(\mathcal{F}) = \mathbb{E}_{\boldsymbol{\sigma}} \left[ \sup_{h \in \mathcal{H}} \frac{1}{mn} \sum_{i=1}^{m} \sum_{j=1}^{n} \sigma_i^j \ell(h, Z_i^j) \right].$$

**Lemma 1** (Generalization error for participating clinets). *Let $\mathcal{F}$ be a family of functions related to hypothesis space $\mathcal{H} : \mathcal{F} = \{z \mapsto \ell(h, z) : h \in \mathcal{H}\}$. Assume that loss function $\ell$ is bounded by $M$. Then, for any $\delta \geq 0$, with probability at least $1 - \delta$, we have*

$$\sup_{h \in \mathcal{H}} |\mathcal{L}_D(h) - \mathcal{L}_S(h)| \leq 2\mathcal{R}_{mn}(\mathcal{F}) + 3M \sqrt{\frac{\ln(1/\delta)}{2mn}}.$$

Lemma 1 provide high probability theoretical guarantees for participating clients under bounded assumption. Compared to the classical i.i.d setting in learning theory, the results in Lemma 1 in under independent but non-identically distributed setting.

Since each local distribution $D_i$ is sampled independently from $P$, we regard local population risk $\{\mathbb{E}_{Z \sim D_i}[\ell(h, Z)]\}_{i \in [m]}$ as a collection of iid random variables. Let $\boldsymbol{\sigma} = \{\sigma_i\}_{i \in [m]}$ be a collection of independent Rademacher variables, the rademacher complexity used in the analysis of participating gap is defined as:

$$\mathcal{R}_m(\mathcal{F}) = \mathbb{E}_{\substack{D_1, \cdots, D_m \\ \boldsymbol{\sigma}}} \left[ \sup_{h \in \mathcal{H}} \frac{1}{m} \sum_{i=1}^{m} \sigma_i \mathbb{E}_{Z \sim D_i}[\ell(h, Z)] \right].$$

**Lemma 2** (Participation gap). *Let $\mathcal{F}$ be a family of functions related to hypothesis space $\mathcal{H} : \mathcal{F} = \{z \mapsto \ell(h, z) : h \in \mathcal{H}\}$. Assume that loss function $\ell$ is bounded by $b$. Then, for any $\delta \geq 0$, with probability at least $1 - \delta$, we have*

$$\sup_{h \in \mathcal{H}} |\mathcal{L}_P(h) - \mathcal{L}_D(h)| \leq 2\mathcal{R}_m(\mathcal{F}) + 3b \sqrt{\frac{\ln(1/\delta)}{2m}}.$$

The participation gap in Lemma 2 quantifies the error caused by missing clients during training. The Rademacher complexity term $\mathcal{R}_m(\mathcal{F})$ indicates the 'size' of hypothesis class, which can be future bounded by convering number or VC dimension of the choosen hypothesis.

PROOF OF LEMMA 1

*Proof.* We first create sample set $S'$ from $S$ by changing one sample point. Without loss generality, we assume the data point $Z_s^k$ in $S$ is changed into $Z_s'^k$. Next we define $\Phi(S)$ by $\Phi(S) = $

$\sup_{h \in \mathcal{H}} |\mathcal{L}_D(h) - \mathcal{L}_S(h)|$. By the definition of $S'$ and $S$, we have

$$\Phi(S') - \Phi(S) = \sup_{h \in \mathcal{H}} \{|\mathcal{L}_D(h) - \mathcal{L}_{S'}(h)|\} - \sup_{h \in \mathcal{H}} \{|\mathcal{L}_D(h) - \mathcal{L}_S(h)|\}$$

$$\leq \sup_{h \in \mathcal{H}} \{|\{\mathcal{L}_D(h) - \mathcal{L}_{S'}(h)\} - \{\mathcal{L}_D(h) - \mathcal{L}_S(h)\}|\}$$

$$= \sup_{h \in \mathcal{H}} \{|\mathcal{L}_S(h) - \mathcal{L}_{S'}(h)|\} = \frac{1}{mn} \sup_{h \in \mathcal{H}} |\ell(h, Z_s^k) - \ell(h, Z_s'^k)|.$$

By assuming that loss function $\ell$ is bounded by $b$, then we have $|\Phi(S') - \Phi(S)| \leq \frac{b}{mn}$. We apply McDiarmid's inequality and obtain, for any fixed $\delta \in (0,1)$, it follows that with probability at least $1 - \delta$

$$\Phi(S) \leq \mathop{\mathbb{E}}_{S_1, \cdots, S_m} [\Phi(S)] + b\sqrt{\frac{\ln(2/\delta)}{2mn}}$$

Next we consider the expectation of $\Phi(S)$. By symmetrization, we have

$$\mathop{\mathbb{E}}_{S_1, \cdots, S_m} [\Phi(S)] = \mathop{\mathbb{E}}_{S_1, \cdots, S_m} \left[ \sup_{h \in \mathcal{H}} \mathcal{L}_D(h) - \mathcal{L}_S(h) \right]$$

$$= \mathop{\mathbb{E}}_{S_1, \cdots, S_m} \left[ \sup_{h \in \mathcal{H}} \frac{1}{m} \sum_{i=1}^m \left( \mathbb{E}_{Z \sim D_i} [\ell(h, Z)] - \frac{1}{n} \sum_{j=1}^n \ell(h, Z_i^j) \right) \right]$$

$$= \mathop{\mathbb{E}}_{S_1, \cdots, S_m} \left[ \sup_{h \in \mathcal{H}} \frac{1}{m} \sum_{i=1}^m \mathop{\mathbb{E}}_{S_1', \cdots, S_m'} \left[ \frac{1}{n} \sum_{j=1}^n \left( \ell(h, Z_i'^j) - \ell(h, Z_i^j) \right) \right] \right]$$

$$\leq \mathop{\mathbb{E}}_{\substack{S_1, \cdots, S_m \\ S_1', \cdots, S_m'}} \left[ \sup_{h \in \mathcal{H}} \frac{1}{m} \sum_{i=1}^m \frac{1}{n} \sum_{j=1}^n \left( \ell(h, Z_i'^j) - \ell(h, Z_i^j) \right) \right]$$

$$= \mathop{\mathbb{E}}_{\substack{S_1, \cdots, S_m \\ S_1', \cdots, S_m'}} \left[ \sup_{h \in \mathcal{H}} \mathop{\mathbb{E}}_{\sigma} \frac{1}{mn} \sum_{i=1}^m \sum_{j=1}^n \sigma_{ij} \left( \ell(h, Z_i'^j) - \ell(h, Z_i^j) \right) \right]$$

$$\leq \mathop{\mathbb{E}}_{\substack{S_1, \cdots, S_m \\ S_1', \cdots, S_m' \\ \sigma}} \left[ \sup_{h \in \mathcal{H}} \frac{1}{mn} \sum_{i=1}^m \sum_{j=1}^n \sigma_{ij} \left( \ell(h, Z_i'^j) - \ell(h, Z_i^j) \right) \right]$$

$$\leq \mathop{\mathbb{E}}_{S_1', \cdots, S_m', \sigma} \left[ \sup_{h \in \mathcal{H}} \frac{1}{mn} \sum_{i=1}^m \sum_{j=1}^n \sigma_{ij} \ell(h, Z_i'^j) \right]$$

$$+ \mathop{\mathbb{E}}_{S_1, \cdots, S_m, \sigma} \left[ \sup_{h \in \mathcal{H}} \frac{1}{mn} \sum_{i=1}^m \sum_{j=1}^n -\sigma_{ij} \ell(h, Z_i'^j) \right]$$

$$= 2 \mathop{\mathbb{E}}_{S_1, \cdots, S_m, \sigma} \left[ \sup_{h \in \mathcal{H}} \frac{1}{mn} \sum_{i=1}^n \sum_{j=1}^m \sigma_{ij} \ell(h, Z_i^j) \right]$$

$$= 2 \mathop{\mathbb{E}}_{S_1, \cdots, S_m} [\mathcal{R}_{mn}(\mathcal{F})]$$

where $\mathcal{R}_{mn}(\mathcal{F}) = \mathop{\mathbb{E}}_{\sigma} \left[ \sup_{h \in \mathcal{H}} \frac{1}{mn} \sum_{i=1}^n \sum_{j=1}^m \sigma_{ij} \ell(h, Z_i^j) \right]$. Replacing one data point of $S = S_1 \bigcup \cdots \bigcup S_m$ make $\mathcal{R}_{mn}(\mathcal{F})$ vary at most $\frac{b}{mn}$, then applying McDiarmid's inequality again, with probability at least $1 - \delta$ we have

$$\mathop{\mathbb{E}}_{S_1, \cdots, S_n} [\mathcal{R}_{mn}(\mathcal{F})] \leq \mathcal{R}_{mn}(\mathcal{F}) + b\sqrt{\frac{\ln(2/\delta)}{2mn}}$$

$\square$

PROOF OF LEMMA 2

*Proof of Lemma 2.* We create distributions set $D'$ from $D$ by changing one client distribution. Without loss generality, we assume the $k$-th $D_k$ in $D$ is changed into $D'_k$, then we define $\Phi(D)$ by

$$\Phi(D) = \sup_{h \in \mathcal{H}} \mathcal{L}_P(h) - \mathcal{L}_D(h).$$

From the definition of $D'$ and $D$, we have

$$
\begin{aligned}
&\Phi\left(D'\right) - \Phi(D) \\
&= \sup_{h \in \mathcal{H}} \left\{\mathcal{L}_P(h) - \mathcal{L}_{D'}(h)\right\} - \sup_{h \in \mathcal{H}} \left\{\mathcal{L}_P(h) - \mathcal{L}_D(h)\right\} \\
&\leq \sup_{h \in \mathcal{H}} \left\{\left\{\mathcal{L}_P(h) - \mathcal{L}_{D'}(h)\right\} - \left\{\mathcal{L}_P(h) - \mathcal{L}_D(h)\right\}\right\} \\
&= \sup_{h \in \mathcal{H}} \left\{\mathcal{L}_D(h) - \mathcal{L}_{D'}(h)\right\} = \sup_{h \in \mathcal{H}} \frac{1}{m} \left(\mathbb{E}_{Z \sim D_k}\left[\ell(h, Z)\right] - \mathbb{E}_{Z \sim D'_k}\left[\ell(h, Z)\right]\right)
\end{aligned}
$$

By assuming that loss function $f$ is bounded by $b$, then we have $|\Phi\left(D'\right) - \Phi(D)| \leq \frac{b}{m}$. According to McDiarmid's inequality, for all $\delta \in (0, 1)$, with probability at least $1 - \delta$, we have

$$\Phi(D) \leq \mathbb{E}_{D_1, \cdots, D_m}[\Phi(D)] + b\sqrt{\frac{\ln(2/\delta)}{2m}}$$

Now we deal with the expectation of $\Phi(D)$. By symmetrization, we have

$$
\begin{aligned}
\mathbb{E}_{D_1, \cdots, D_m}[\Phi(D)] &= \mathbb{E}_{D_1, \cdots, D_m}\left[\sup_{h \in \mathcal{H}} \mathcal{L}_P(h) - \mathcal{L}_D(h)\right] \\
&= \mathbb{E}_{D_1, \cdots, D_m}\left[\sup_{h \in \mathcal{H}} \left(\mathbb{E}_{D' \sim P^m}\left[\mathcal{L}_{D'}(h)\right] - \mathcal{L}_D(h)\right)\right] \\
&\leq \mathbb{E}_{\substack{D_1, \cdots, D_m \\ D'_1, \cdots, D'_m}}\left[\sup_{h \in \mathcal{H}} \left(\mathcal{L}_{D'}(h) - \mathcal{L}_D(h)\right)\right] \\
&= \mathbb{E}_{\substack{D_1, \cdots, D_m \\ D'_1, \cdots, D'_m}}\left[\sup_{h \in \mathcal{H}} \frac{1}{m} \sum_{i=1}^{m} \left(\mathcal{L}_{D'_i}(h) - \mathcal{L}_{D_i}(h)\right)\right] \\
&= \mathbb{E}_{\substack{D_1, \cdots, D_m \\ D'_1, \cdots, D'_m}}\left[\sup_{h \in \mathcal{H}} \frac{1}{m} \sum_{i=1}^{m} \sigma_i\left(\mathcal{L}_{D'_i}(h) - \mathcal{L}_{D_i}(h)\right)\right] \\
&\leq \mathbb{E}_{D_1, \cdots, D_m, \sigma}\left[\sup_{h \in \mathcal{H}} \frac{1}{m} \sum_{i=1}^{m} \sigma_i \mathcal{L}_{D'_i}(h)\right] + \mathbb{E}_{D'_1, \cdots, D'_m, \sigma}\left[\sup_{h \in \mathcal{H}} \frac{1}{m} \sum_{i=1}^{m} -\sigma_i \mathcal{L}_{D_i}(h)\right] \\
&\leq 2 \mathbb{E}_{D_1, \cdots, D_m, \sigma}\left[\sup_{h \in \mathcal{H}} \frac{1}{m} \sum_{i=1}^{m} \sigma_i \mathcal{L}_{D_i}(h)\right]
\end{aligned}
$$

Thus, we have

$$
\begin{aligned}
\mathbb{E}_{D_1, \cdots, D_m}[\Phi(D)] &= 2 \mathbb{E}_{D_1, \cdots, D_m, \sigma}\left[\sup_{h \in \mathcal{H}} \frac{1}{m} \sum_{i=1}^{m} \sigma_i \mathcal{L}_{D_i}(h)\right] \\
&= 2 \mathbb{E}_{D_1, \cdots, D_m, \sigma}\left[\sup_{h \in \mathcal{H}} \frac{1}{m} \sum_{i=1}^{m} \sigma_i \mathbb{E}_{Z_i \sim D_i}\left[\ell(h, Z_i)\right]\right] \\
&\leq 2 \mathbb{E}_{Z_1, \cdots, Z_m, \sigma}\left[\sup_{h \in \mathcal{H}} \frac{1}{m} \sum_{i=1}^{m} \sigma_i\left[\ell(h, Z_i)\right]\right] \\
&= 2 \mathbb{E}_{Z_1, \cdots, Z_m} \mathcal{R}_m(\mathcal{L})
\end{aligned}
$$

where $\mathcal{R}_m(\mathcal{L}) = \mathbb{E}_\sigma \left[ \sup_{h\in\mathcal{H}} \frac{1}{m} \sum_{i=1}^m \sigma_i \ell(h, Z_i) \right]$. Replacing one point of $Z = \{Z_i\}_{i=1}^m$ make $\mathcal{R}_m(\mathcal{L})$ vary at most $\frac{M}{m}$, then applying McDiarmid's inequality again, with probability at least $1 - \delta$ we have

$$\mathbb{E}_{Z_1,\cdots,Z_m} [\mathcal{R}_m(\mathcal{L})] \leq \mathcal{R}_m(\mathcal{L}) + b\sqrt{\frac{\ln(2/\delta)}{2m}}.$$

$\square$

The goal of Theorem 1 is to present theoretical bounds of excess risk $\mathcal{L}_P(\widehat{h}) - \mathcal{L}_P(h^*)$, it has been shown that excess risk can be bounded by participation gap $\sup_{h\in\mathcal{H}} \{|\mathcal{L}_P(h) - \mathcal{L}_D(h)|\}$ and participating error $\sup_{h\in\mathcal{H}} \{|\mathcal{L}_D(h) - \mathcal{L}_S(h)|\}$. To show the explicit form of the rademacher complexity terms in Lemma 1 and Lemma 2, we take VC class as an example.

PROOF OF THEOREM 1

*Proof.* We use subgraph dimension(aka pseudo dimension) and Dudley's theorem to bound the empirical rademacher complexity $\mathcal{R}_{mn}(\mathcal{H})$. According to Dudley (1978); Pollard (2012); Haussler (2018), the uniform matric entropy of real valued function class $\mathcal{F}$ with subgraph dimension $\mathrm{VC}(\mathcal{F}) = d$ can be bounded

$$\log \mathcal{N}(\epsilon, \mathcal{F}, \|\cdot\|_2) \leq cd\log(1/\epsilon).$$

where $F$ is the envolop function of $\mathcal{F}$. Then by Dudley's Theorem (Dudley, 1978; van der Vaart & Wellner, 1996), we have

$$\mathcal{R}_{mn}(\mathcal{F}) \leq cb\sqrt{\frac{d}{mn}}, \qquad \mathcal{R}_m(\mathcal{F}) \leq cb\sqrt{\frac{d}{m}}.$$

Combining the results of Lemma 1 and 2 leads to the final results. $\square$

## C PROOFS OF FAST RATES WITH BOUNDED LOSSES

Covering number can be used to give tighter estimate on the hypothesis size. Here we provide the definition of convering number and uniform entropy number.

**Definition 7** (Convering number)**.** *Let $(\mathcal{G}, \rho)$ be a metric space and $\mathcal{F} \subseteq \mathcal{G}$. For any $\epsilon \geq 0$, $\mathcal{F}_\epsilon$ is an $\epsilon$-cover of $\mathcal{F}$ with respect of $\rho$ if for all $f \in F$, we can find $f' \in \mathcal{F}_\epsilon$ such that $\rho(f, f') \leq \epsilon$. The covering number $\mathcal{N}(\epsilon, \mathcal{F}, \rho)$ is defined as the minimum size of an $\epsilon$-cover:*

$$\mathcal{N}(\epsilon, \mathcal{F}, \rho) := \min\{|\mathcal{F}_\epsilon| : \mathcal{F}_\epsilon \text{ is an } \epsilon\text{-cover of } \mathcal{F} \text{ w.r.t } \rho\}.$$

**Definition 8** (Uniform entropy number)**.** *The entropy number is defined as the logarithm of the covering number. Let $(\mathcal{G}, \rho)$ be a normed space with $\rho(f, f') = \|f - f'\|$. Let $F$ be an envelope function of $\mathcal{F}$ such that $|f(Z)| \leq F(Z)$, for all $Z$ and $f$. We further define uniform entropy number of $\mathcal{F}$ as: $\log \mathcal{N}(\epsilon, \mathcal{F}, \|\cdot\|_2) = \sup_Q \log \mathcal{N}(\epsilon, \mathcal{F}, \|\cdot\|_{L_2(Q)})$, where $Q$ is taken over all probability measures with $0 \leq QF^2 \leq \infty$.*

**Definition 9.** *A function $\varphi(r) : [0, \infty) \mapsto [0, \infty)$ is sub-root function if it is nondecreasing and $r \mapsto \varphi(r)/\sqrt{r}$ is nonincreasing for $r > 0$.*

### C.1 EXAMPLES SATISFYING ASSUMPTION 2

#### C.1.1 EXAMPLE 1

In this subsection, we show that when the hypothesis class contains the $\widehat{h}^*$ and $h^*$, Assumption 2 holds.

$$\mathbb{E}_{(X,Y)\sim D}[\ell(h(X), Y) - \ell(\widehat{h}^*(X), Y)]^2 = \frac{1}{m} \sum_{i=1}^m \mathbb{E}\left[\left|\left(h(X_i^1) - Y_i^1\right)^2 - (\widehat{h}^*(X_i^1) - Y_i^1)^2\right|^2\right]$$

$$\leq \frac{L^2}{m} \sum_{i=1}^m \mathbb{E}\left[h(X_i^1) - \widehat{h}^*(X_i^1)\right]^2.$$

Next we show that

$$
\begin{aligned}
\mathbb{E}_{(X,Y)\sim D}[\ell(h(X),Y) - \ell(\widehat{h}^*(X),Y)] &= \frac{1}{m}\sum_{i=1}^{m}\mathbb{E}\left[(h(X_i^1) - Y_i^1)^2 - (\widehat{h}^*(X_i^1) - Y_i^1)^2\right] \\
&= \frac{1}{m}\sum_{i=1}^{m}\mathbb{E}\left[h(X_i^1) - \widehat{h}^*(X_i^1)\right]^2.
\end{aligned}
$$

Thus, if $\widehat{h}^* \in \mathcal{H}$, we have

$$
\begin{aligned}
\mathbb{E}_{(X,Y)\sim D}[\ell(h(X),Y) - \ell(\widehat{h}^*(X),Y)]^2 &\le \frac{L^2}{m}\sum_{i=1}^{m}\mathbb{E}\left[h(X_i^1) - \widehat{h}^*(X_i^1)\right]^2 \\
&= L^2 \mathbb{E}_{(X,Y)\sim D}[\ell(h(X),Y) - \ell(\widehat{h}^*(X),Y)].
\end{aligned}
$$

Thus, the tuple $\left(D, \ell, \mathcal{H}, \widehat{h}^\star\right)$ satisfies the Bernstein condition with parameter $B_2 = 1$ such that

$$
\frac{1}{m}\sum_{i=1}^{m}\mathbb{E}\left[h(X_i^1) - \widehat{h}^*(X_i^1)\right]^2 \le B_2 \mathbb{E}_{(X,Y)\sim D}[\ell(h(X),Y) - \ell(\widehat{h}^*(X),Y)].
$$

Similarly, we can prove that

$$
\begin{aligned}
\mathbb{E}_{(X,Y)\sim P}[\ell(h(X),Y) - \ell(h^*(X),Y)]^2 &\le L^2\mathbb{E}(h(X_i) - h^*(X_i))^2 \\
&= L^2\mathbb{E}_{(X,Y)\sim P}[\ell(h(X),Y) - \ell(h^*(X),Y)].
\end{aligned}
$$

Thus, the tuple $(P, f, \mathcal{H}, h^\star)$ satisfies the Bernstein condition with parameter $B_1 = 1$ such that

$$
\mathbb{E}(h(X_i) - h^*(X_i))^2 \le B_1\mathbb{E}_{(X,Y)\sim P}[\ell(h(X),Y) - \ell(h^*(X),Y)].
$$

### C.1.2 EXAMPLE 2

In this example, we show that even when the hypothesis class does not include $\widehat{h}^*$ and $h^*$, Assumption 2 still holds. If $\mathcal{F}$ is a nonempty, closed and convex subset of a Hilbert space with inner product

$$
\langle f, g \rangle = \mathbb{E}_{(X,Y)\sim D}(f(X,Y)g(X,Y)),
$$

where $\widehat{h}^*$ is the projection of $Y$ in the space $\mathcal{F}$. By the definition we have $\langle Y - \widehat{h}^*(X), h(X) - \widehat{h}^*(X)\rangle \le 0$.

$$
\begin{aligned}
\mathbb{E}_{(X,Y)\sim D}[\ell(h(X),Y) - \ell(\widehat{h}^*(X),Y)] &= \mathbb{E}_{(X,Y)\sim D}\left[(Y - h(X))^2 - (Y - \widehat{h}^*(X))^2\right] \\
&= \left\|(h(X) - \widehat{h}^*(X))^2\right\|_{L_2}^2 + \langle Y - \widehat{h}^*(X), \widehat{h}^*(X) - h(X)\rangle \\
&\ge \frac{1}{m}\sum_{i=1}^{m}\mathbb{E}\left[h(X_i^1) - \widehat{h}^*(X_i^1)\right]^2.
\end{aligned}
$$

Thus, for regression problems with square loss function, where $\left(\widehat{h}^* \notin \mathcal{H}\right)$, we have

$$
\begin{aligned}
\mathbb{E}_{(X,Y)\sim D}[\ell(h(X),Y) - \ell(\widehat{h}^*(X),Y)]^2 &\le \frac{L^2}{m}\sum_{i=1}^{m}\mathbb{E}\left[h(X_i^1) - \widehat{h}^*(X_i^1)\right]^2 \\
&\le L^2 \mathbb{E}_{(X,Y)\sim D}[\ell(h(X),Y) - \ell(\widehat{h}^*(X),Y)].
\end{aligned}
$$

Similarly, we can prove that

$$
\begin{aligned}
\mathbb{E}_{(X,Y)\sim P}[\ell(h(X),Y) - \ell(h^*(X),Y)]^2 &\le L^2\mathbb{E}(h(X_i) - h^*(X_i))^2 \\
&= L^2\mathbb{E}_{(X,Y)\sim P}[\ell(h(X),Y) - \ell(h^*(X),Y)].
\end{aligned}
$$

## C.2   PROOF OF THEOREM 2

**Lemma 3** (Theorem 2.1 in (Bartlett et al., 2005)). *Let $\mathcal{F}$ be a family of functions satisfying $\forall f \in \mathcal{F}, \forall Z \in \mathcal{Z}, |f(Z)| \leq b$. If $\sup_{f \in \mathcal{F}} \frac{1}{m} \sum_{i=1}^{m} \mathbb{E}[f(Z_i)]^2 \leq r$, then, for any $\delta \geq 0$ and any $\alpha \geq 0$, with probability at least $1 - \delta$,*

$$\sup_{f \in \mathcal{F}} \left[ \frac{1}{m} \sum_{i=1}^{m} \left[ \mathbb{E}[f(Z_i)] - f(Z_i) \right] \right] \leq 2(1+\alpha)\mathbb{E}[\mathcal{R}_m(\mathcal{F})] + \sqrt{\frac{2r \ln(1/\delta)}{m}} + \left( \frac{1}{3} + \frac{1}{\alpha} \right) \frac{b \ln(1/\delta)}{m},$$

*where $\mathcal{R}_m(\mathcal{F}) = \mathbb{E}_{\sigma} \left[ \sup_{f \in \mathcal{F}} \frac{1}{m} \sum_{i=1}^{m} \sigma_i f(Z_i) \right]$.*

**Lemma 4** (Theorem 1 in (Yousefi et al., 2018)). *Let $\mathcal{F}$ be a family of functions satisfying $\forall f \in \mathcal{F}, \forall Z \in \mathcal{Z}, |f(Z)| \leq b$. If $\sup_{f \in \mathcal{F}} \frac{1}{mn} \sum_{i=1}^{m} \sum_{j=1}^{n} \mathbb{E}[f(Z_i^j)]^2 \leq r$, then, for any $\delta \geq 0$ and any $\alpha \geq 0$, with probability at least $1 - \delta$,*

$$\sup_{f \in \mathcal{F}} \left[ \frac{1}{mn} \sum_{i=1}^{m} \sum_{j=1}^{n} \left[ \mathbb{E}[f(Z_i^j)] - f(Z_i^j) \right] \right] \leq 2(1+\alpha)\mathbb{E}[\mathcal{R}_{mn}(\mathcal{F})] + \sqrt{\frac{8r \ln(1/\delta)}{mn}} + \left( 1 + \frac{2}{\alpha} \right) \frac{4b \ln(1/\delta)}{mn},$$

*where $\mathcal{R}_{mn}(\mathcal{H}) = \mathbb{E}_{\sigma} \left[ \sup_{f \in \mathcal{F}} \frac{1}{mn} \sum_{i=1}^{m} \sum_{j=1}^{n} \sigma_i^j f(Z_i^j) \right]$.*

**Lemma 5** (Lemma B.1 in (Yousefi et al., 2018)). *Let $c_1, c_2 > 0$ and $s > q > 0$. Then the equation $x^s - c_1 x^q - c_2 = 0$ has a unique positive solution $x_0$ satisfying*

$$x_0 \leq \left[ c_1^{\frac{s}{s-1}} + \frac{sc_2}{s-q} \right]^{\frac{1}{s}}.$$

*Moreover, for any $x \geq x_0$, we have $x^s \geq c_1 x^q + c_2$.*

**Lemma 6** (Corollary 1 in Lei et al. (2016)). *Let $\mathcal{F}$ be a function class with $\sup_{f \in \mathcal{F}} \|f\|_\infty \leq b$. Assume that there exist three positive numbers $\gamma, d, p$ such that $\log \mathcal{N}(\epsilon, \mathcal{F}, \| \cdot \|_2) \leq d \log^p(\gamma/\epsilon)$ for any $0 < \epsilon \leq \gamma$, then for any $0 < r \leq \gamma^2$ and $n \geq \gamma^{-2}$ there holds that*

$$\mathbb{E}R_n \left\{ f \in \mathcal{F} : Pf^2 \leq r \right\} \leq c(b, p, \gamma) \min \left[ \left( \sqrt{\frac{dr \log^p \left( 2\gamma r^{-1/2} \right)}{n}} + \frac{d \log^p \left( 2\gamma r^{-1/2} \right)}{n} \right) \right.$$
$$\left. \left( \frac{d \log p \left( 2\gamma n^{1/2} \right)}{n} + \sqrt{\frac{rd \log^p \left( 2\gamma n^{1/2} \right)}{n}} \right) \right].$$

*Proof of Theorem 2.* First we define $\widehat{\mathcal{F}}^* := \{ f : (X, Y) \mapsto \ell(h(X), Y) - \ell(\widehat{h}^*(X), Y), h \in \mathcal{H} \}$.

**Step One:** Combining Assumption 2 (a), for any $f \in \widehat{\mathcal{F}}^*$ we derive that

$$\frac{1}{m} \sum_{i=1}^{m} \mathbb{E} \left[ f^2(X_i^1, Y_i^1) \right] = \frac{1}{m} \sum_{i=1}^{m} \mathbb{E} \left[ \ell(h(X_i^1), Y_i^1) - \ell(\widehat{h}^*(X_i^1), Y_i^1) \right]^2$$
$$\leq \frac{L^2}{m} \sum_{i=1}^{m} \mathbb{E} \left[ h(X_i^1) - \widehat{h}^*(X_i^1) \right]^2$$
$$\leq B'L^2 \left( \frac{1}{m} \sum_{i=1}^{m} \mathbb{E} \left[ \ell(h(X_i^1), Y_i^1) - \ell(\widehat{h}^*(X_i^1), Y_i^1) \right] \right)^\beta.$$

Let $V(f) := \frac{1}{m} \sum_{i=1}^{m} \mathbb{E} \left[ \ell(h(X_i^1), Y_i^1) - \ell(\widehat{h}^*(X_i^1), Y_i^1) \right]^2, B = B'L^2$.

Consider the function class $\mathcal{G}_r$ associated with $\widehat{\mathcal{F}}^*$ and $r \geq 0$:

$$\mathcal{G}_r := \left\{ g := \frac{rf}{\max(r, V(f))}, f \in \widehat{\mathcal{F}}^* \right\}.$$

We denote $V_r^+$ by

$$V_r^+ = \sup_{g \in \mathcal{G}_r} \frac{1}{mn} \sum_{i=1}^{m} \sum_{j=1}^{n} \left[ \mathbb{E}[g(Z_i^j)] - g(Z_i^j) \right].$$

Let $K > 1, 0 < \beta \leq 1$ and $B \geq 1$. We first prove that, if $V_r^+ \leq \frac{r^{1/\beta}}{BK}$, then

$$\forall f \in \mathcal{F}, \quad \frac{1}{m} \sum_{i=1}^{m} \mathbb{E}[f(Z_i^1)] \leq \frac{K}{(mn)(K-\beta)} \sum_{i=1}^{m} \sum_{j=1}^{n} \left[ f(Z_i^j) \right] + \frac{r^{1/\beta}}{K}. \tag{4}$$

If $V(f) \leq r$, then $g = f$. It follows that with the assumption $V_r^+ \leq \frac{r^{1/\beta}}{BK}$, we have

$$\frac{1}{m} \sum_{i=1}^{m} \mathbb{E}[f(Z_i^1)] \leq \frac{1}{mn} \sum_{i=1}^{m} \sum_{j=1}^{n} \left[ f(Z_i^j) \right] + \frac{r^{1/\beta}}{BK} \tag{5}$$

$$\leq \frac{K}{(mn)(K-\beta)} \sum_{i=1}^{m} \sum_{j=1}^{n} \left[ f(Z_i^j) \right] + \frac{r^{1/\beta}}{K}. \tag{6}$$

If $V(f) \geq r$, then $g = rf/V(f)$. It follows that with the assumption $V_r^+ \leq \frac{r^{1/\beta}}{BK}$, we have

$$\frac{1}{m} \sum_{i=1}^{m} \mathbb{E}[f(Z_i^1)] \leq \frac{1}{mn} \sum_{i=1}^{m} \sum_{j=1}^{n} \left[ f(Z_i^j) \right] + \frac{r^{\frac{1}{\beta}-1} V(f)}{BK} \tag{7}$$

$$\leq \frac{1}{mn} \sum_{i=1}^{m} \sum_{j=1}^{n} \left[ f(Z_i^j) \right] + \frac{r^{\frac{1}{\beta}-1}}{K} \left( \frac{1}{m} \sum_{i=1}^{m} \mathbb{E}[f(Z_i^1)] \right)^{\beta} \tag{8}$$

$$\leq \frac{1}{mn} \sum_{i=1}^{m} \sum_{j=1}^{n} \left[ f(Z_i^j) \right] + \frac{\beta}{K} \frac{1}{m} \sum_{i=1}^{m} \mathbb{E}[f(Z_i^1)] + \frac{(1-\beta) r^{\frac{1}{\beta}}}{K}, \tag{9}$$

where the second inequality follows from Bernstein Condition and the third inequality follows from Lemma 5.

The obtained inequality can be rewritten as

$$\frac{1}{m} \sum_{i=1}^{m} \mathbb{E}[f(Z_i^1)] \leq \frac{K}{(mn)(K-\beta)} \sum_{i=1}^{m} \sum_{j=1}^{n} \left[ f(Z_i^j) \right] + \frac{(1-\beta) r^{\frac{1}{\beta}}}{(K-\beta)} \tag{10}$$

$$\leq \frac{K}{(mn)(K-\beta)} \sum_{i=1}^{m} \sum_{j=1}^{n} \left[ f(Z_i^j) \right] + \frac{r^{\frac{1}{\beta}}}{K}. \tag{11}$$

where the second inequality is due to $\frac{b-c}{a-c} \leq \frac{b}{a}$ for $c < b \leq a$.

**Step Two:** By the construction of $\mathcal{G}_r$, it can be varified that

$$\frac{1}{mn} \sum_{i=1}^{m} \sum_{j=1}^{n} \mathbb{E}[g(Z_i^j)]^2 \leq r.$$

In details:

If $V(f) \leq r$, we have $g = f$. It follows that

$$\frac{1}{mn} \sum_{i=1}^{m} \sum_{j=1}^{n} \mathbb{E}[g(Z_i^j)]^2 = \frac{1}{m} \sum_{i=1}^{m} \mathbb{E}[g(Z_i^1)]^2 = \frac{1}{m} \sum_{i=1}^{m} \mathbb{E}[f(Z_i^1)]^2 \leq V(f) \leq r. \tag{12}$$

If $V(f) > r$, we have $g = rf/V(f)$. It follows that

$$\frac{1}{mn} \sum_{i=1}^{m} \sum_{j=1}^{n} \mathbb{E}[g(Z_i^j)]^2 = \frac{1}{m} \sum_{i=1}^{m} \mathbb{E}[g(Z_i^1)]^2 = \frac{r^2}{m[V(f)]^2} \sum_{i=1}^{m} \mathbb{E}[f(Z_i^1)]^2 \leq \frac{r^2}{[V(f)]} \leq r. \tag{13}$$

Combining the boundness of $\mathcal{G}_r$ and Lemma 4, with probability at least $1 - \delta, \forall \delta \geq 0, \forall \alpha \geq 0$, we have

$$\sup_{g \in \mathcal{G}_r} \left[ \frac{1}{mn} \sum_{i=1}^{m} \sum_{j=1}^{n} \left[ \mathbb{E}[g(Z_i^j)] - g(Z_i^j) \right] \right] \leq 2(1 + \alpha)\mathbb{E}[\mathcal{R}_{mn}(\mathcal{G}_r)] + \sqrt{\frac{8r \ln(1/\delta)}{mn}} + \left( 1 + \frac{2}{\alpha} \right) \frac{4b \ln(1/\delta)}{mn},$$

where $\mathcal{R}_{mn}(\mathcal{G}_r) = \mathbb{E}_{\sigma} \left[ \sup_{g \in \mathcal{G}_r} \frac{1}{mn} \sum_{i=1}^{m} \sum_{j=1}^{n} \sigma_i^j g(Z_i^j) \right]$.

Next we apply "peeling" technique to bound $\mathbb{E}[\mathcal{R}_{mn}(\mathcal{G}_r)]$. Given $\lambda > 1$, let $\widehat{\mathcal{F}}^*(u, v) := \{ f \in \widehat{\mathcal{F}}^* : u \leq V(f) \leq v \}$ and denote $k$ as the smallest integer such that $r\lambda^{k+1} \geq Bb^\beta$. Then

$$\mathbb{E}[\mathcal{R}_{mn}(\mathcal{G}_r)] = \mathbb{E}_{S,\sigma} \left[ \sup_{g \in \mathcal{G}_r} \frac{1}{mn} \sum_{i=1}^{m} \sum_{j=1}^{n} \sigma_i^j g(Z_i^j) \right] = \mathbb{E}_{S,\sigma} \left[ \sup_{f \in \widehat{\mathcal{F}}^*} \frac{1}{mn} \sum_{i=1}^{m} \sum_{j=1}^{n} \frac{r}{\max(r, V(f))} \sigma_i^j f(Z_i^j) \right]$$

$$\leq \mathbb{E}_{S,\sigma} \left[ \sup_{f \in \widehat{\mathcal{F}}^*(0,r)} \frac{1}{mn} \sum_{i=1}^{m} \sum_{j=1}^{n} \sigma_i^j f(Z_i^j) \right] + \mathbb{E}_{S,\sigma} \left[ \sup_{f \in \widehat{\mathcal{F}}^*(r,Bb^\beta)} \frac{1}{mn} \sum_{i=1}^{m} \sum_{j=1}^{n} \frac{r}{V(f)} \sigma_i^j f(Z_i^j) \right]$$

$$\leq \mathbb{E}_{S,\sigma} \left[ \sup_{f \in \widehat{\mathcal{F}}^*(0,r)} \frac{1}{mn} \sum_{i=1}^{m} \sum_{j=1}^{n} \sigma_i^j f(Z_i^j) \right] + \sum_{j=0}^{k} \lambda^{-j} \mathbb{E}_{S,\sigma} \left[ \sup_{f \in \widehat{\mathcal{F}}^*(r\lambda^j, r\lambda^{j+1})} \frac{1}{mn} \sum_{i=1}^{m} \sum_{j=1}^{n} \sigma_i^j f(Z_i^j) \right]$$

$$\leq \mathbb{E}_{S,\sigma} \left[ \sup_{f \in \widehat{\mathcal{F}}^*(0,r)} \frac{1}{mn} \sum_{i=1}^{m} \sum_{j=1}^{n} \sigma_i^j f(Z_i^j) \right] + \sum_{j=0}^{k} \lambda^{-j} \mathbb{E}_{S,\sigma} \left[ \sup_{f \in \widehat{\mathcal{F}}^*(0, r\lambda^{j+1})} \frac{1}{mn} \sum_{i=1}^{m} \sum_{j=1}^{n} \sigma_i^j f(Z_i^j) \right]$$

$$\leq \frac{\psi(r)}{B} + \sum_{j=0}^{k} \lambda^{-j} \frac{\psi(r\lambda^{j+1})}{B}.$$

By the property of sub-root function it follows that we have $\psi(\theta r) \leq \theta^{\frac{1}{2}} \psi(r)$ for any $\theta \geq 1$. Then,

$$\mathbb{E}[\mathcal{R}_{mn}(\mathcal{G}_r)] \leq \frac{\psi(r)}{B} \left( 1 + \sqrt{\lambda} \sum_{j=0}^{k} \lambda^{-\frac{j}{2}} \right) \leq \frac{\psi(r)}{B} \left( 1 + \frac{\lambda}{\sqrt{\lambda} - 1} \right).$$

Taking $\lambda = 4$ it follows that $\mathbb{E}[\mathcal{R}_{mn}(\mathcal{G}_r)] \leq 5\psi(r)/B$.

Combining with the property $\psi(r) \leq \sqrt{r/r^*}\psi(r^*) = \sqrt{rr^*}$. The following inequality can be obtained with probability at least $1 - \delta, \forall \delta \geq 0$,

$$\sup_{g \in \mathcal{G}_r} \left[ \frac{1}{mn} \sum_{i=1}^{m} \sum_{j=1}^{n} \left[ \mathbb{E}[g(Z_i^j)] - g(Z_i^j) \right] \right] \leq \frac{10(1+\alpha)}{B} \sqrt{rr^*} + \sqrt{\frac{8r \ln(1/\delta)}{mn}} + \left( 1 + \frac{2}{\alpha} \right) \frac{4b \ln(1/\delta)}{mn},$$

where $r^*$ is the fixed point of sub-root function $\psi(r)$.

**Step Three:** Recall that the condition we get inequality (4) is

$$V_r^+ = \sup_{g \in \mathcal{G}_r} \frac{1}{mn} \sum_{i=1}^{m} \sum_{j=1}^{n} \left[ \mathbb{E}[g(Z_i^j)] - g(Z_i^j) \right] \leq \frac{r^{1/\beta}}{BK}.$$

We denote $A$ and $C$ by

$$A = \frac{10(1+\alpha)}{B} \sqrt{r^*} + \sqrt{\frac{8 \ln(1/\delta)}{mn}}, \qquad C = \left( 1 + \frac{2}{\alpha} \right) \frac{4b \ln(1/\delta)}{mn}$$

Next, we need to solve $A\sqrt{r} + C \leq \frac{r^{1/\beta}}{BK}$. Assume $r_0$ is the positive solution of $A\sqrt{r} + C = \frac{r^{1/\beta}}{BK}$, that is

$$r_0^{1/\beta} - ABKr_0^{\frac{1}{2}} - BKC = 0, \tag{14}$$

Then by Lemma 5, we have

$$r_0^{\frac{1}{\beta}} \leq (ABK)^{\frac{2}{2-\beta}} + \frac{2BKC}{2-\beta}$$

$$\leq (BK)^{\frac{2}{2-\beta}} 2^{\frac{\beta}{2-\beta}} \left[ \left( \frac{10(1+\alpha)}{B} \right)^{\frac{2}{2-\beta}} (r^*)^{\frac{1}{2-\beta}} + \left( \frac{8 \ln(1/\delta)}{mn} \right)^{\frac{1}{2-\beta}} \right] + \left( 1 + \frac{2}{\alpha} \right) \frac{8BKb \ln(1/\delta)}{(2-\beta)mn},$$

where the second inequality follows from $(x + y)^p \leq 2^{p-1}(x^p + y^p)$ for $x, y \geq 0, p \geq 1$.

If $r^* \leq r_0$, we can take $r = r_0$. Then we have $V_{r_0}^+ \leq A\sqrt{r_0} + C = \frac{r_0^{1/\beta}}{BK}$. Combining inequality 4, we get

$$\frac{1}{m} \sum_{i=1}^{m} \mathbb{E}[f(Z_i^1)] \leq \frac{K}{(mn)(K - \beta)} \sum_{i=1}^{m} \sum_{j=1}^{n} \left[ f(Z_i^j) \right] + r_0^{\frac{1}{\beta}}/K$$

$$\leq \frac{K}{(mn)(K - \beta)} \sum_{i=1}^{m} \sum_{j=1}^{n} \left[ f(Z_i^j) \right] + (2K)^{\frac{\beta}{2-\beta}} (10(1+\alpha))^{\frac{2}{2-\beta}} (r^*)^{\frac{1}{2-\beta}}$$

$$+ \left( \frac{2^{\beta+3} B^2 K^\beta x}{mn} \right)^{\frac{1}{2-\beta}} + \frac{(1 + (2/\alpha))8Bb\ln(1/\delta)}{(2 - \beta)mn}.$$

If $r^* > r_0$, we can take $r = r^*$. Then we have $V_{r^*}^+ \leq A\sqrt{r^*} + C = \frac{(r^*)^{1/\beta}}{BK}$. Combining inequality 4, we get

$$\frac{1}{m} \sum_{i=1}^{m} \mathbb{E}[f(Z_i^1)] \leq \frac{K}{(mn)(K - \beta)} \sum_{i=1}^{m} \sum_{j=1}^{n} \left[ f(Z_i^j) \right] + (r^*)^{\frac{1}{\beta}}/K.$$

**Step Four:**

Note that $\psi(r)$ is set to satisfy the following conditon,

$$\psi(r) \geq B\mathcal{R}_{mn}(\widehat{\mathcal{F}}^*(0, r)),$$

where $B = B_2 L^2$ and

Clearly $\mathcal{G}_r \subset \{f \in \widehat{\mathcal{F}}^* : T_{mn}(f) \leq r\}$, where $T_{mn}(f) = \frac{1}{m} \sum_{i=1}^{m} \mathbb{E}_{Z \sim D_i} [f(Z)]^2$. Thus,

$$\mathbb{E}[\mathcal{R}_{mn}(\mathcal{G}_r)] = \underset{S_1,\cdots,S_m,\sigma}{\mathbb{E}} \left[ \sup_{g \in \mathcal{G}_r} \frac{1}{mn} \sum_{i=1}^{m} \sum_{j=1}^{n} \sigma_i^j g(Z_i^j) \right]$$

$$\leq \underset{S_1,\cdots,S_m,\sigma}{\mathbb{E}} \left[ \sup_{\substack{T_{mn}(f) \leq r \\ f \in \widehat{\mathcal{F}}^*}} \frac{1}{mn} \sum_{i=1}^{m} \sum_{j=1}^{n} \sigma_i^j f(X_i^j, Y_i^j) \right]$$

$$= \mathcal{R}_{mn}(\widehat{\mathcal{F}}^*, r).$$

Lemma 6 implies that the sub-root function can be chosen as

$$\psi(r) := c \left[ \frac{d \log^p \left(2\gamma(mn)^{1/2}\right)}{mn} + \sqrt{\frac{rd \log^p \left(2\gamma(mn)^{1/2}\right)}{mn}} \right].$$

In a similar way to Lei et al. (2016), let $r_{mn}^*$ be its fixed point then we know that

$$r_{mn}^* = c \left[ \frac{d \log^p \left(2\gamma(mn)^{1/2}\right)}{mn} + \sqrt{\frac{r_{mn}^* d \log^p \left(2\gamma(mn)^{1/2}\right)}{mn}} \right].$$

Solving this equality gives $r_{mn}^* \leq cd(mn)^{-1} \log^p(mn)$.

Combining the fact that $\frac{1}{mn} \sum_{i=1}^{m} \sum_{j=1}^{n} \left[ \ell(\widehat{h}(X_i^j), Y_i^j) - \ell(\widehat{h}^*(X_i^j), Y_i^j) \right] \leq 0$ and the result of step three, we get the following results

$$\mathcal{L}_D(\widehat{h}) - \mathcal{L}_D(\widehat{h}^*) = \frac{1}{m} \sum_{i=1}^{m} \mathbb{E} \left[ \ell(\widehat{h}(X_i^j), Y_i^j) - \ell(\widehat{h}^*(X_i^1), Y_i^1) \right]$$

$$\leq (2K)^{\frac{\beta}{2-\beta}} (10(1+\alpha))^{\frac{2}{2-\beta}} \max((r^*)^{\frac{1}{2-\beta}}, (r^*)^{\frac{1}{\beta}})$$

$$+ \left( \frac{2^{\beta+3} B^2 K^\beta x}{mn} \right)^{\frac{1}{2-\beta}} + \frac{(1 + (2/\alpha))8Bb\ln(1/\delta)}{(2 - \beta)mn}.$$

When $mn \geq cd \log^p(mn)$, we have $\max((r^*)^{\frac{1}{2-\beta}}, (r^*)^{\frac{1}{\beta}}) = (r^*)^{\frac{1}{2-\beta}}$. Thus,

$$\mathcal{L}_D(\widehat{h}) - \mathcal{L}_D(\widehat{h}^*) \leq c_1 \left( \frac{\log^p(mn)}{mn} \right)^{\frac{1}{2-\beta'}} + c_2 \left( \frac{\log(1/\delta)}{mn} \right)^{\frac{1}{2-\beta'}}.$$

$\square$

## C.3  PROOF OF THEOREM 3

*Proof.* First we define $\mathcal{F}^* := \{f : (X, Y) \mapsto \ell(h(X), Y) - \ell(h^*(X), Y), h \in \mathcal{H}\}$.

**Step One:** Combining Assumption 2 (b), for any $f \in \mathcal{F}^*$, we have

$$
\begin{aligned}
\mathbb{E}_{D_i \sim P}[\mathbb{E}_{(X,Y) \sim D_i}[f(X,Y)]]^2 &= \mathbb{E}_{D_i \sim P} \left[ \mathbb{E}_{(X,Y) \sim D_i}[\ell(h(X), Y) - \ell(h^*(X), Y)] \right]^2 \\
&\leq \mathbb{E}_{D_i \sim P} \mathbb{E}_{(X,Y) \sim D_i} \left[ \ell(h(X), Y) - \ell(h^*(X), Y) \right]^2 \\
&= \mathbb{E} \left[ \ell(h(X_1), Y_1) - \ell(h^*(X_1), Y_1) \right]^2 \\
&\leq L^2 \mathbb{E}[h(X_1) - h^*(X_1)]^2 \\
&\leq B'' L^2 (\mathcal{L}_P(h) - \mathcal{L}_P(h^*))^{\beta''}.
\end{aligned}
$$

Let $V(f) := \mathbb{E}_{D_i \sim P} \mathbb{E}_{(X,Y) \sim D_i} \left[ \ell(h(X), Y) - \ell(h^*(X), Y) \right]^2$, $B = B'' L^2$.

Consider the function class $\mathcal{G}_r$ associated with $\mathcal{F}^*$ and $r \geq 0$:

$$\mathcal{G}_r := \left\{ g := \frac{rf}{\max(r, V(f))}, f \in \mathcal{F}^* \right\}.$$

We denote $V_r^+$ by

$$V_r^+ = \sup_{g \in \mathcal{G}_r} \frac{1}{m} \sum_{i=1}^m \left[ \mathbb{E}[g(Z_i)] - \mathbb{E}[g(Z_i^1)] \right].$$

Let $K > 1, 0 < \beta \leq 1$ and $B \geq 1$. We first prove that, if $V_r^+ \leq \frac{r^{1/\beta}}{BK}$, then

$$\forall f \in \mathcal{F}^*, \qquad \mathbb{E}[f(Z_1)] \leq \frac{K}{m(K-\beta)} \frac{1}{m} \sum_{i=1}^m \mathbb{E}\left[f(Z_i^1)\right] + \frac{r^{1/\beta}}{K}. \tag{15}$$

If $V(f) \leq r$, then $g = f$. It follows that with the assumption $V_r^+ \leq \frac{r^{1/\beta}}{BK}$, we have

$$\mathbb{E}[f(Z_1)] \leq \frac{1}{m} \sum_{i=1}^m \mathbb{E}\left[f(Z_i^1)\right] + \frac{r^{1/\beta}}{BK} \tag{16}$$

$$\leq \frac{K}{m(K-\beta)} \sum_{i=1}^m \left[ \mathbb{E}f(Z_i^1) \right] + \frac{r^{1/\beta}}{K}. \tag{17}$$

If $V(f) \geq r$, then $g = rf/V(f)$. It follows that with the assumption $V_r^+ \leq \frac{r^{1/\beta}}{BK}$, we have

$$\mathbb{E}[f(Z_1)] \leq \frac{1}{m} \sum_{i=1}^m \mathbb{E}\left[f(Z_i^1)\right] + \frac{r^{\frac{1}{\beta}-1} V(f)}{BK} \tag{18}$$

$$\leq \frac{1}{m} \sum_{i=1}^m \mathbb{E}\left[f(Z_i^1)\right] + \frac{r^{\frac{1}{\beta}-1}}{K} \left(\mathbb{E}[f(Z_1)]\right)^{\beta} \tag{19}$$

$$\leq \frac{1}{m} \sum_{i=1}^m \mathbb{E}\left[f(Z_i^1)\right] + \frac{\beta}{K} \mathbb{E}[f(Z_1)] + \frac{(1-\beta)r^{\frac{1}{\beta}}}{K}, \tag{20}$$

where the second inequality follows from Bernstein Condition and the third inequality follows from Lemma 5.

The obtained inequality can be rewritten as

$$\mathbb{E}[f(Z_1)] \leq \frac{K}{m(K-\beta)} \sum_{i=1}^{m} \left[ f(Z_i^1) \right] + \frac{(1-\beta)r^{\frac{1}{\beta}}}{(K-\beta)} \tag{21}$$

$$\leq \frac{K}{m(K-\beta)} \sum_{i=1}^{m} \left[ f(Z_i^1) \right] + \frac{r^{\frac{1}{\beta}}}{K}. \tag{22}$$

**Step Two:** By the construction of $\mathcal{G}_r$, it can be varified that

$$\frac{1}{m} \sum_{i=1}^{m} \mathbb{E}_{D_i \sim P}[\mathbb{E}_{Z_i^1 \sim D_i}[g(Z_i^1)]]^2 \leq r.$$

In details:

First, we have

$$\frac{1}{m} \sum_{i=1}^{m} \mathbb{E}_{D_i \sim P}[\mathbb{E}_{Z_i^1 \sim D_i}[g(Z_i^1)]]^2 \leq \frac{1}{m} \sum_{i=1}^{m} \mathbb{E}_{D_i \sim P}[\mathbb{E}_{Z_i^1 \sim D_i}[g(Z_i^1)]^2] = \frac{1}{m} \sum_{i=1}^{m} \mathbb{E}[g(Z_i)]^2. \tag{23}$$

If $V(f) \leq r$, we have $g = f$. It follows that

$$\frac{1}{m} \sum_{i=1}^{m} \mathbb{E}[g(Z_i)]^2 = \mathbb{E}[g(Z_i)]^2 = \mathbb{E}[f(Z_i)]^2 \leq V(f) \leq r. \tag{24}$$

If $V(f) > r$, we have $g = rf/V(f)$. It follows that

$$\frac{1}{m} \sum_{i=1}^{m} \mathbb{E}[g(Z_i)]^2 = \mathbb{E}[g(Z_i)]^2 = \frac{r^2}{[V(f)]^2} \mathbb{E}[f(Z_i)]^2 \leq \frac{r^2}{[V(f)]} \leq r. \tag{25}$$

Combining the boundness of $\mathcal{G}_r$ and Lemma 3, with probability at least $1-\delta, \forall \delta > 0, \forall \alpha > 0$, we have

$$\sup_{g \in \mathcal{G}_r} \frac{1}{m} \sum_{i=1}^{m} \left[ \mathbb{E}[g(Z_i)] - \mathbb{E}[g(Z_i^1)] \right] \leq 2(1+\alpha) \mathop{\mathbb{E}}_{D_i, \cdots, D_m, \sigma} \left[ \sup_{g \in \mathcal{G}_r} \frac{1}{m} \sum_{i=1}^{m} \sigma_i \mathbb{E}_{Z_i^1 \sim D_i}[g(Z_i^1)] \right]$$
$$+ \sqrt{\frac{2r \ln(1/\delta)}{m}} + \left( \frac{1}{3} + \frac{1}{\alpha} \right) \frac{b \ln(1/\delta)}{m}.$$

Thus, we get

$$\sup_{g \in \mathcal{G}_r} \frac{1}{m} \sum_{i=1}^{m} \left[ \mathbb{E}[g(Z_i)] - \mathbb{E}[g(Z_i^1)] \right] \leq 2(1+\alpha)\mathbb{E}[\mathcal{R}_m(\mathcal{G}_r)] + \sqrt{\frac{2r \ln(1/\delta)}{m}} + \left( \frac{1}{3} + \frac{1}{\alpha} \right) \frac{b \ln(1/\delta)}{m}.$$

where $\mathcal{R}_m(\mathcal{G}_r) = \mathop{\mathbb{E}}_{\sigma} \left[ \sup_{g \in \mathcal{G}_r} \frac{1}{m} \sum_{i=1}^{m} \sigma_i g(Z_i) \right]$.

By "peeling" technique and similar following steps in the proof of Theorem 2, it follows with probability at least $1-\delta, \forall \delta \geq 0$

$$\sup_{g \in \mathcal{G}_r} \left[ \frac{1}{mn} \sum_{i=1}^{m} \sum_{j=1}^{n} \left[ \mathbb{E}[g(Z_i^j)] - g(Z_i^j) \right] \right] \leq \frac{10(1+\alpha)}{B} \sqrt{rr^*} + \sqrt{\frac{2r \ln(1/\delta)}{m}} + \left( \frac{1}{3} + \frac{1}{\alpha} \right) \frac{b \ln(1/\delta)}{m},$$

where $r^*$ is the fixed point of sub-root function $\psi(r)$.

**Step Three:** Then by Lemma 5, we have

$$r_0^{\frac{1}{\beta}} \leq (BK)^{\frac{2}{2-\beta}} 2^{\frac{\beta}{2-\beta}} \left[ \left( \frac{10(1+\alpha)}{B} \right)^{\frac{2}{2-\beta}} (r^*)^{\frac{1}{2-\beta}} + \left( \frac{2 \ln(1/\delta)}{mn} \right)^{\frac{1}{2-\beta}} \right] + \left( \frac{1}{3} + \frac{1}{\alpha} \right) \frac{2BKb \ln(1/\delta)}{(2-\beta)mn}.$$

Let $\mathcal{F}$ be the loss function class. We consider the functional $T(f) := Pf^2$ here. The structural result on covering numbers implies that

$$\log \mathcal{N}(\epsilon, \mathcal{F}, \|\cdot\|_2) \leq \log \mathcal{N}(\epsilon/L, \mathcal{H}, \|\cdot\|_2) \leq d \log^p(\gamma L/\epsilon).$$

Lemma 6 implies that the sub-root function can be chosen as

$$\psi(r) := c \left[ \frac{d \log^p \left(2\gamma m^{1/2}\right)}{m} + \sqrt{\frac{rd \log^p \left(2\gamma m^{1/2}\right)}{m}} \right].$$

In a similar way to Lei et al. (2016), let $r_m^*$ be its fixed point then we know that

$$r_m^* = c \left[ \frac{d \log^p \left(2\gamma m^{1/2}\right)}{m} + \sqrt{\frac{r^* d \log^p \left(2\gamma m^{1/2}\right)}{m}} \right].$$

Solving this equality gives $r_m^* \leq c d m^{-1} \log^p(m)$.

In a similar way to the proof of Theorem 2, we have

$$\mathbb{E}[f(Z_1)] \leq \frac{K}{m(K - \beta'')} \sum_{i=1}^{m} \left[ f(Z_i^1) \right] + c_1 \left( \frac{\log^p(m)}{m} \right)^{\frac{1}{2-\beta''}} + c_2 \left( \frac{\log(1/\delta)}{m} \right)^{\frac{1}{2-\beta''}}. \quad (26)$$

That is,

$$\mathcal{L}_P(\widehat{h}) - \mathcal{L}_P(h^*) \leq \frac{K}{K - \beta''} \left( \mathcal{L}_D(\widehat{h}) - \mathcal{L}_D(h^*) \right) + c_1 \left( \frac{\log^p(m)}{m} \right)^{\frac{1}{2-\beta''}} + c_2 \left( \frac{\log(1/\delta)}{m} \right)^{\frac{1}{2-\beta''}}.$$

Using the fact that

$$\mathcal{L}_D(\widehat{h}) - \mathcal{L}_D(h^*) = \mathcal{L}_D(\widehat{h}^*) - \mathcal{L}_D(h^*) + \mathcal{L}_D(\widehat{h}) - \mathcal{L}_D(\widehat{h}^*) \leq \mathcal{L}_D(\widehat{h}) - \mathcal{L}_D(\widehat{h}^*),$$

we get

$$\mathcal{L}_P(\widehat{h}) - \mathcal{L}_P(h^*) \leq \frac{K}{K - \beta''} \left( \mathcal{L}_D(\widehat{h}) - \mathcal{L}_D(\widehat{h}^*) \right) + c_1 \left( \frac{\log^p(m)}{m} \right)^{\frac{1}{2-\beta''}} + c_2 \left( \frac{\log(1/\delta)}{m} \right)^{\frac{1}{2-\beta''}}.$$

Combining with Theorem 2, we complete the proof. $\square$

## D  PROOFS OF THE RESULTS WITH SUB-EXPONENTIAL LOSSES

### D.1  LEARNING RATES FOR SUB-EXPONENTIAL LOSSES

We state our results on the convergnece rate of generalization error for sub-exponential losses. First, we consider the participating clients. Let $S = \{Z_i^j\}_{(i,j=1,1)}^{(m,n)}$ be global data samples whose subsets $S_i = \{Z_i^j\}_{j=1}^n$ include i.i.d random variables at $i$-th client.

**Theorem 6** (Participating error for sub-exponential losses). *Suppose $Z_i^j$ take valued in a Banach space $(\mathcal{Z}, \|\cdot\|)$ and each $\|Z_i^j\|$ is sub-exponential distributed. We denote by $\mathcal{F} = \{Z \mapsto \ell(h, Z) : h \in \mathcal{H}\}$ such that, $\forall f \in \mathcal{F}$ and $\forall z, z' \in \mathcal{Z}$, $|f(z) - f(z')| \leq L\|z - z'\|$. For any $\delta > 0$, if $mn \geq \ln(1/\delta) \geq \ln 2$, then with probability at least $1 - \delta$, we have*

$$\sup_{h \in \mathcal{H}} |\mathcal{L}_D(h) - \mathcal{L}_S(h)| \leq 2\mathbb{E}[\mathcal{R}_{mn}(\mathcal{F})] + \max_{i \in [m]} 16eL \|\|Z_i^1\|\|_{\psi_1} \sqrt{\frac{2\ln(1/\delta)}{mn}},$$

*where $\mathcal{R}_{mn}(\mathcal{F}) = \mathbb{E}_\sigma \left[ \sup_{f \in \mathcal{F}} \frac{1}{mn} \sum_{i=1}^m \sum_{j=1}^n \sigma_i^j f(Z_i^j) \right]$.*

**Remark 9.** *Theorem 6 can be used to bound semi-empirical excess by applying standard uniformly supremum of $h \in \mathcal{H}$. It is worth emphasizing that the bounds derived in Theorem 6 include Rademacher complexity term and $\|\|Z_i^1\|\|_{\psi_1}$ measuring the tails of input data samples $\{Z_i^j\}_{(i,j=1,1)}^{(m,n)}$. Intuitively, in regression problems, as the noise added to the lables increases, it is expected that participating error increase as well. This phenomenon is ignored under the previous bounded assumption on losses.*

**Example 1** (Linear regression with unbounded loss)**.** *Let $\mathcal{Z} = (\mathcal{X}, \mathbb{R})$, where $\mathcal{X}$ is a Hilbert-space with norm $\|\cdot\|_H$. We denote by $(X_i^j, Y_i^j)$ each sub-exponential random variables in $H$ and $\mathbb{R}$ respectively. Let loss function $\ell$ be a 1-Lipschitz function (absolute function or Huber loss) and $\mathcal{F} = \{(x, y) \to f(x, y) = \ell(\langle w, x \rangle - y) : \|w\|_H \leq L\}$. If $mn \geq \ln(1/\delta) \geq \ln 2$, it follows with probability at least $1 - \delta$*

$$\sup_{h \in \mathcal{H}} |\mathcal{L}_D(h) - \mathcal{L}_S(h)| \leq \frac{4}{\sqrt{mn}} \left( \max_{i \in [m]} L \|\|X_i^1\|\|_{\psi_1} + \max_{i \in [m]} \|\|Y_i^1\|\|_{\psi_1} \right) \left( 1 + 6e\sqrt{\ln(\frac{1}{\delta})} \right).$$

**Theorem 7** (Participation gap with unbounded loss)**.** *Under the same conditons as Theorem 6 and Example 1, we have*

$$\sup_{h \in \mathcal{H}} |\mathcal{L}_P(h) - \mathcal{L}_D(h)| \leq \frac{8}{\sqrt{m}} \left( L \|\|X_1\|\|_{\psi_1} + \|\|Y_1\|\|_{\psi_1} \right) \left( 1 + 3e\sqrt{\ln(\frac{1}{\delta})} \right),$$

*where $X_1$ is random vector with expectation across two-level distribution. This is, $\mathbb{E}[\|X_1\|] = \mathbb{E}_{D_i \sim P} \mathbb{E}_{X_1 \sim D_i} \|X_1\|$. Similarly, $\mathbb{E}[\|Y_1\|] = \mathbb{E}_{D_i \sim P} \mathbb{E}_{Y_1 \sim D_i} \|Y_1\|$.*

**Remark 10.** *Combining the results of Theorem 6 and Theorem 7, it can be shown that upper bound of excess risk is of order $\mathcal{O}(\frac{1}{\sqrt{mn}} + \frac{1}{\sqrt{m}})$. Though this bound is derived under the unbounded assumption, its order is comparable with basic results derived in Theorem 1. Note that the upper bounds in Theorem 7 include terms such as $\|X_1\|$ and $\|Y_1\|$, whose underlying distributions are across our two-level framework. This reflects that the participation gap captures the generalization error caused by client sampling.*

**Lemma 7** (Theorem 3.1 in Maurer & Pontil (2021))**.** *Let $f : \mathcal{X}^n \to \mathbb{R}$ and $X = (X_1, \ldots, X_n)$ be a random vector whose elements are independent and take values in a space $\mathcal{X}$. Then for any $t \geq 0$*

$$\mathbb{P}\{f(X) - \mathbb{E}[f(X')] > t\} \leq \exp\left( \frac{-t^2}{4e^2 \left\| \sum_k \|f_k(X)\|_{\psi_1}^2 \right\|_\infty + 2e \max_k \left\| \|f_k(X)\|_{\psi_1} \right\|_\infty t} \right)$$

PROOF OF THEOREM 6

*Proof.* We first define a vector space

$$\mathcal{B} = \left\{ g : \mathcal{F} \to \mathbb{R} : \sup_{f \in \mathcal{F}} |g(f)| \leq \infty \right\}.$$

By definition, $\mathcal{B}$ is a normed space with norm $\|g\|_\mathcal{B} = \sup_{f \in \mathcal{F}} |g(f)|$. For each $Z_i^j \in \mathcal{Z}$, we define $\hat{Z}_i^j(f)$ by $(mn)^{-1}(f(Z_i^j) - \mathbb{E}[f(Z_i'^j)])$. Thus, $\mathbb{E}[\hat{Z}_i^j] \equiv 0$, and

$$\left\| \sum_i \sum_j \hat{Z}_i^j \right\|_\mathcal{B} = \sup_{f \in \mathcal{F}} \left| \frac{1}{mn} \sum_i \sum_j (f(Z_i^j) - \mathbb{E}[f(Z_i'^j)]) \right|.$$

From Lemma 7, we have

$$\left\| \sum_i \sum_j \hat{Z}_i^j \right\|_\mathcal{B} - \mathbb{E}\left[ \left\| \sum_i \sum_j \hat{Z}_i^j \right\|_\mathcal{B} \right] \leq \left\| \sum_i \sum_j \hat{Z}_i^j - \mathbb{E}\left[ \sum_i \sum_j \hat{Z}_i^j \right] \right\|_\mathcal{B}$$
$$\leq \max_{i \in [m]} 8e\sqrt{mn} \|\|\hat{Z}_i^1\|_\mathcal{B}\|_{\psi_1} \sqrt{2\ln(1/\delta)}.$$

Observe that

$$\|\|\hat{Z}_i^j\|_\mathcal{B}\|_{\psi_1} = \frac{1}{mn} \left\| \sup_{f \in \mathcal{F}} \left| \mathbb{E}\left[ f(Z_i^j) - f(Z_i'^j) \mid Z \right] \right| \right\|_{\psi_1} \leq \frac{1}{mn} \left\| \sup_{f \in \mathcal{F}} \mathbb{E}\left[ \left| f(Z_i^j) - f(Z_i'^j) \right| \mid Z \right] \right\|_{\psi_1}$$
$$\leq \frac{L}{mn} \left\| \mathbb{E}\left[ \left\| Z_i^j - Z_i'^j \right\| \mid Z \right] \right\|_{\psi_1} \leq \frac{2L}{mn} \|\|Z_i^1\|\|_{\psi_1}.$$

Therefore, we get $\max_{i\in[m]} \|\|\hat{Z}_i^1\|_{\mathcal{B}}\|_{\psi_1} \leq \max_{i\in[m]} \frac{2L}{mn}\|\|Z_i^1\|\|_{\psi_1}$, and

$$\left\|\sum_i \sum_j \hat{Z}_i^j\right\|_{\mathcal{B}} - \mathbb{E}\left[\left\|\sum_i \sum_j \hat{Z}_i^j\right\|_{\mathcal{B}}\right] \leq \max_{i\in[m]} 16eL\|\|Z_i^1\|\|_{\psi_1}\sqrt{\frac{2\ln(1/\delta)}{mn}}.$$

By symmetrization, we have

$$\mathbb{E}\left[\left\|\sum_i \sum_j \hat{Z}_i^j\right\|_{\mathcal{B}}\right] \leq 2\mathop{\mathbb{E}}_{S_1,\cdots,S_m}[\mathcal{R}_{mn}(\mathcal{F})] = 2\mathop{\mathbb{E}}_{S_1,\cdots,S_m}\left[\mathop{\mathbb{E}}_{\sigma}\left[\sup_{f\in\mathcal{F}}\frac{1}{mn}\sum_{i=1}^n\sum_{j=1}^m \sigma_{ij}f(Z_i^j) \mid Z\right]\right].$$

$$\square$$

PROOF OF EXAMPLE 1

*Proof.*

$$\mathcal{R}_{mn}(\mathcal{F}) = \mathop{\mathbb{E}}_{\sigma}\left[\sup_{f\in\mathcal{F}}\frac{1}{mn}\sum_{i=1}^n\sum_{j=1}^m \sigma_{ij}f(Z_i^j)\right] = \mathop{\mathbb{E}}_{\sigma}\left[\sup_{\|w\|_H\leq L}\frac{1}{mn}\sum_{i=1}^n\sum_{j=1}^m \sigma_{ij}\ell(\langle w, X_i^j\rangle - Y_i^j)\right]$$

$$\leq \mathop{\mathbb{E}}_{\sigma}\left[\sup_{\|w\|_H\leq L}\frac{1}{mn}\sum_{i=1}^n\sum_{j=1}^m \langle w, \sigma_{ij}X_i^j\rangle\right] + \mathop{\mathbb{E}}_{\sigma}\left[\frac{1}{mn}\sum_{i=1}^n\sum_{j=1}^m \sigma_{ij}Y_i^j\right]$$

$$= \mathop{\mathbb{E}}_{\sigma}\left[\sup_{\|w\|_H\leq L}\frac{1}{mn}\langle w, \sum_{i=1}^n\sum_{j=1}^m \sigma_{ij}X_i^j\rangle\right] + \mathop{\mathbb{E}}_{\sigma}\left[\frac{1}{mn}\sum_{i=1}^n\sum_{j=1}^m \sigma_{ij}Y_i^j\right]$$

$$\leq \frac{L}{mn}\mathop{\mathbb{E}}_{\sigma}\left[\|\sum_{i=1}^n\sum_{j=1}^m \sigma_{ij}X_i^j\|_H\right] + \mathop{\mathbb{E}}_{\sigma}\left[\frac{1}{mn}\sum_{i=1}^n\sum_{j=1}^m \sigma_{ij}Y_i^j\right]$$

Next, using Jensen's inequality we can see that

$$\mathop{\mathbb{E}}_{\sigma}\left[\|\sum_{i=1}^n\sum_{j=1}^m \sigma_{ij}X_i^j\|_H\right] = \mathop{\mathbb{E}}_{\sigma}\left[\left(\|\sum_{i=1}^n\sum_{j=1}^m \sigma_{ij}X_i^j\|_H^2\right)^{\frac{1}{2}}\right] \leq \left(\mathop{\mathbb{E}}_{\sigma}\left[\|\sum_{i=1}^n\sum_{j=1}^m \sigma_{ij}X_i^j\|_H^2\right]\right)^{\frac{1}{2}}$$

By the assumption that $\sigma_{1,1},\ldots,\sigma_{m,n}$ are independent, then we have

$$\mathop{\mathbb{E}}_{\sigma}\left[\left\|\sum_{i=1}^n\sum_{j=1}^m \sigma_{ij}X_i^j\right\|_H^2\right] = \mathop{\mathbb{E}}_{\sigma}\left[\sum_{(i,j)=(1,1)}^{(m,n)}\sum_{(s,k)=(1,1)}^{(m,n)} \sigma_{ij}\sigma_{sk}\langle X_i^j, X_s^k\rangle\right]$$

$$= \mathop{\mathbb{E}}_{\sigma}\left[\sum_{(i,j)\neq(s,k)} \sigma_{ij}\sigma_{sk}\langle X_i^j, X_s^k\rangle\right] + \mathop{\mathbb{E}}_{\sigma}\left[\sum_{(i,j)=(1,1)}^{(m,n)} \sigma_{ij}^2\langle X_i^j, X_i^j\rangle\right]$$

$$= \mathop{\mathbb{E}}_{\sigma}\left[\sum_{(i,j)=(1,1)}^{(m,n)} \sigma_{ij}^2\langle X_i^j, X_i^j\rangle\right] = \sum_{(i,j)}\|X_i^j\|_H^2.$$

Thus, we have

$$\mathop{\mathbb{E}}_{\sigma}\left[\|\sum_{i=1}^n\sum_{j=1}^m \sigma_{ij}X_i^j\|_H\right] \leq \sqrt{\sum_{(i,j)}\|X_i^j\|_H^2}.$$

Similarly, we have

$$\mathop{\mathbb{E}}_{\sigma}\left[\sum_{i=1}^n\sum_{j=1}^m \sigma_{ij}Y_i^j\right] \leq \sqrt{\sum_{(i,j)}|Y_i^j|^2}$$

Therefore,

$$\mathcal{R}_{mn}(\mathcal{F}) \le \frac{L}{mn}\sqrt{\sum_{(i,j)}\|X_i^j\|_H^2} + \sqrt{\sum_{(i,j)}|Y_i^j|^2}$$

Since $X_1^1, \cdots, X_m^n$ are independent and $\|\cdot\|_2 \le 2\|\cdot\|_{\psi_1}$, we get

$$\mathbb{E}\left[\mathcal{R}_{mn}(\mathcal{H})\right] \le \max_{i\in[m]}\frac{2L}{\sqrt{mn}}\|\|X_i^1\|\|_{\psi_1} + \max_{i\in[m]}\frac{1}{\sqrt{mn}}\|\|Y_i^1\|\|_{\psi_1}$$

$\square$

PROOF OF THEOREM 7

*Proof.* We first define a vector space

$$\mathcal{B} = \left\{g : \mathcal{F} \to \mathbb{R} : \sup_{f\in\mathcal{F}}|g(f)| \le \infty\right\}.$$

By definition, $\mathcal{B}$ is a normed space with norm $\|g\|_{\mathcal{B}} = \sup_{f\in\mathcal{F}}|g(f)|$. Let $f(D_i) = \mathbb{E}_{Z_i^1\sim D_i}[f(Z_i^1)]$ and $\mathbb{E}[f(D_i)] = \mathbb{E}_{D_i\sim P}[\mathbb{E}_{Z_i^1\sim D_i}[f(Z_i^1)]]$. For each $Z_i^1 \in \mathcal{Z}$, we define $\hat{Z}_i$ by $(1/m)([f(D_i)] - \mathbb{E}[f(D_i)])$. Thus, $\mathbb{E}[\hat{Z}_i] \equiv 0$, and

$$\left\|\sum_i \hat{Z}_i\right\|_{\mathcal{B}} = \sup_{f\in\mathcal{F}}\left|\frac{1}{m}\sum_i(f(D_i) - \mathbb{E}[f(D_i')])\right|.$$

From Lemma 7, we have

$$\left\|\sum_i \hat{Z}_i\right\|_{\mathcal{B}} - \mathbb{E}\left[\left\|\sum_i \hat{Z}_i\right\|_{\mathcal{B}}\right] \le \left\|\sum_i \hat{Z}_i^j - \mathbb{E}\left[\sum_i \hat{Z}_i\right]\right\|_{\mathcal{B}}$$
$$\le 8e\sqrt{m}\|\|\hat{Z}_i\|_{\mathcal{B}}\|_{\psi_1}\sqrt{2\ln(1/\delta)}.$$

Observe that

$$\|\|\hat{Z}_i\|_{\mathcal{B}}\|_{\psi_1} = \frac{1}{m}\left\|\sup_{f\in\mathcal{F}}|\mathbb{E}[f(D_i) - f(D_i') \mid D]|\right\|_{\psi_1} \le \frac{1}{m}\left\|\sup_{f\in\mathcal{F}}\mathbb{E}[|f(D_i) - f(D_i')| \mid D]\right\|_{\psi_1}$$
$$\le \frac{1}{m}\left\|\mathbb{E}\left[\sup_{f\in\mathcal{F}}|f(D_i) - f(D_i')| \mid D\right]\right\|_{\psi_1} \le \frac{1}{m}\left\|\sup_{f\in\mathcal{F}}|f(D_i) - f(D_i')|\right\|_{\psi_1}.$$

$$\mathbb{E}\left[\left|\sup_{f\in\mathcal{F}}|f(D_i) - f(D_i')|\right|^p\right] \le \mathbb{E}\left[\left|\sup_{f\in\mathcal{F}}|f(Z_i) - f(Z_i')|\right|^p\right] \le \mathbb{E}\left[|L\|Z_i - Z_i'\||^p\right].$$

Therefore $\left\|\sup_{f\in\mathcal{F}}|f(D_i) - f(D_i')|\right\|_p \le \|L\|Z_i - Z_i'\|\|_p$ and $\left\|\sup_{f\in\mathcal{F}}|f(D_i) - f(D_i')|\right\|_{\psi_1} \le \|L\|Z_i - Z_i'\|\|_{\psi_1}$. Then we get $\|\|\hat{Z}_i\|_{\mathcal{B}}\|_{\psi_1} \le \frac{2L}{m}\|\|Z_i\|\|_{\psi_1}$, and

$$\left\|\sum_i \hat{Z}_i\right\|_{\mathcal{B}} - \mathbb{E}\left[\left\|\sum_i \hat{Z}_1\right\|_{\mathcal{B}}\right] \le 16eL\|\|Z_i\|\|_{\psi_1}\sqrt{\frac{2\ln(1/\delta)}{m}}.$$

By symmetrization, we have

$$\mathbb{E}\left[\left\|\sum_i \hat{Z}_i\right\|_{\mathcal{B}}\right] \le 2\mathop{\mathbb{E}}_{D_1,\cdots,D_m}\left[\mathop{\mathbb{E}}_{\sigma}\left[\sup_{f\in\mathcal{F}}\frac{1}{mn}\sum_{i=1}^m\sigma_i f(D_i) \mid D\right]\right]$$
$$\le 2\mathop{\mathbb{E}}_{Z_1,\cdots,Z_m}\left[\mathop{\mathbb{E}}_{\sigma}\left[\sup_{f\in\mathcal{F}}\frac{1}{mn}\sum_{i=1}^m\sigma_i f(Z_i) \mid Z\right]\right]$$
$$\le 2\mathop{\mathbb{E}}_{Z_1,\cdots,Z_m}\left[\mathcal{R}_m(\mathcal{F})\right] \le \frac{8}{\sqrt{m}}\left(L\|\|X_1\|\|_{\psi_1} + \|\|Y_1\|\|_{\psi_1}\right).$$

$\square$

# E   PROOFS OF SMALL-BALL BASED METHOD

**Definition 10.** *Let $\mathcal{H} \subset L_2(D)$ be a closed and convex class of functions and $\mathcal{H} - \mathcal{H} := \{h - h' : h, h' \in \mathcal{H}\}$.*

1. *Let $Q_{\mathcal{H}}(\tau) = \inf_{h \in \mathcal{H}} \mathbb{P}(|h(X_i^1)| \geq \tau \|h\|_{L_2(D)})$, where $X_i^1$ represent the random sample at $i$-th participating client.*

2. *Let $Q_{\mathcal{H}}(\tau, P) = \inf_{h \in \mathcal{H}} \mathbb{P}\left(|\mathbb{E}[h(X_i^1)]| \geq \tau \|h\|_{L_2(P)}\right)$, where $X_i^1$ represent the random sample at $i$-th participating client.*

**Definition 11.** *We denote by $B_2^m$ the $L_2(D)$ unit ball entered at $\widehat{h}^*$, that is $B_2^m = \{h \in \mathcal{H} : \|h - \widehat{h}^*\|_{L_2(D)} \leq 1\}$. For every $\eta > 0$, define*

$$\omega_{mn}(\mathcal{H}, \eta) := \inf \left\{ s > 0 : \mathbb{E}\left[ \sup_{h \in \mathcal{H} \cap sB_2^m} \left| \frac{1}{mn} \sum_{i=1}^{m} \sum_{j=1}^{n} \sigma_i^j h(X_i^j) \right| \right] \leq \eta s, \right\}$$

*where $\sigma_i^j$ are Rademacher random variables.*

**Definition 12.** *We denote by $B_2$ the $L_2(P)$ unit ball entered at $h^*$. For every $\eta > 0$, define*

$$\omega_m(\mathcal{H}, \eta) := \inf \left\{ s > 0 : \mathbb{E}\left[ \sup_{h \in \mathcal{H} \cap sB_2} \left| \frac{1}{m} \sum_{i=1}^{m} \sigma_i h(X_i) \right| \right] \leq \eta s, \right\}$$

*where $\sigma_i$ are Rademacher random variables.*

**Lemma 8** (Theorem 1 in (Zhang & Wei, 2022)). *If $\{X_i\}_{i=1}^n$ are independent centralized random variables such that $\|X_i\|_{\psi_\alpha} < \infty$ for all $1 \leq i \leq n$ and some $1 > \alpha > 0$, then for any weight vector $\boldsymbol{w} = (w_1, \ldots, w_n) \in \mathbb{R}^n$, the following bounds holds true:*

$$\mathbb{P}\left( \|\sum_{j=1}^{n} w_i X_i\| \geq t \right) \leq 2 \exp \left\{ -\left( \frac{c_1 t^2}{\sum_{i=1}^{n} w_i^2 \|X_i\|_{\psi_\alpha}^2} \wedge \frac{c_2 n^\alpha}{\max_{1 \leq i \leq n} w_i \|X_i\|_{\psi_\alpha}^\alpha} \right) \right\}.$$

## E.1   PROOF OF THEOREM 4

*Proof.* **Step One:**

$$\mathcal{L}_S(h) - \mathcal{L}_S(\widehat{h}^*) = \frac{1}{mn} \sum_{i=1}^{m} \sum_{j=1}^{n} \left[ (h(X_i^j) - Y_i^j)^2 - (\widehat{h}^*(X_i^j) - Y_i^j)^2 \right] \tag{27}$$

$$= \frac{1}{mn} \sum_{i=1}^{m} \sum_{j=1}^{n} (h - \widehat{h}^*)^2 (X_i^j) + \frac{2}{mn} \sum_{i=1}^{m} \sum_{j=1}^{n} \xi_i^j (h - \widehat{h}^*)(X_i^j), \tag{28}$$

The second term of the RHS of (28) is determined by the underline semi-empirical distribution $D$ and the hypothesis space $\mathcal{H}$, therefore we focus on the first term in the following.

For any $h \in \mathcal{H}$ and $u > 0$,

$$|\{(i,j) : h(X_i^j) > u\}| = \sum_{i=1}^{m} \sum_{j=1}^{n} \mathbb{1}_{\{|h(X_i^j)| \geq u\}}.$$

Also,

$$\frac{1}{mn} \sum_{i=1}^{m} \sum_{j=1}^{n} \mathbb{1}_{\{|h(X_i^j)| \geq u\}} = \frac{1}{m} \sum_{i=1}^{m} \mathbb{E}\left[ \mathbb{1}_{\{|h(X_i^1)| \geq 2u\}} \right]$$

$$+ \frac{1}{mn} \sum_{i=1}^{m} \sum_{j=1}^{n} \mathbb{1}_{\{|h(X_i^j)| \geq u\}} - \frac{1}{m} \sum_{i=1}^{m} \mathbb{E}_{X \sim D_i} \left[ \mathbb{1}_{\{|h(X_i^1)| \geq 2u\}} \right]$$

Let $\phi_u : \mathbb{R} \to [0, 1]$ be the function

$$\phi_u(t) = \begin{cases} 0 & t \leq u, \\ \frac{t}{u} - 1 & u \leq t \leq 2u, \\ 1 & t \geq 2u. \end{cases} \quad (29)$$

Note that for every $t \in \mathbb{R}$, $\phi_u(t) \geq \mathbb{1}_{\{t \geq 2u\}}$ and $\phi_u(t) \leq \mathbb{1}_{\{t \geq u\}}$. Thus,

$$
\begin{aligned}
\frac{1}{mn} \sum_{i=1}^m \sum_{j=1}^n \mathbb{1}_{\{|h(X_i^j)| \geq u\}} &\geq \frac{1}{m} \sum_{i=1}^m \mathbb{E}\left[\mathbb{1}_{\{|h(X_i^1)| \geq 2u\}}\right] \\
&\quad + \frac{1}{mn} \sum_{i=1}^m \sum_{j=1}^n \phi_u(|h(X_i^j)|) - \frac{1}{m} \sum_{i=1}^m \mathbb{E}\left[\phi_u(|h(X_i^1)|)\right] \\
&\geq \inf_{h \in \mathcal{H}} \mathbb{P}(|h(X_i^1)| \geq 2u) \\
&\quad - \sup_{h \in \mathcal{H}} \left| \frac{1}{mn} \sum_{i=1}^m \sum_{j=1}^n \phi_u(|h(X_i^j)|) - \frac{1}{m} \sum_{i=1}^m \mathbb{E}_{X \sim D_i}\left[\phi_u(|h(X)|)\right] \right|.
\end{aligned}
$$

Since function $\phi_u(t)$ is bounded by 1, using Mcdiarmid's inequality, we get that, for every $\delta > 0$, with probability at least $1 - 2\exp(-2\delta^2)$,

$$
\begin{aligned}
\sup_{h \in \mathcal{H}} &\left| \frac{1}{mn} \sum_{i=1}^m \sum_{j=1}^n \phi_u(|h(X_i^j)|) - \frac{1}{m} \sum_{i=1}^m \mathbb{E}\left[\phi_u(|h(X_i^1)|)\right] \right| \\
&\leq \mathbb{E} \sup_{h \in \mathcal{H}} \left| \frac{1}{mn} \sum_{i=1}^m \sum_{j=1}^n \phi_u(|h(X_i^j)|) - \frac{1}{m} \sum_{i=1}^m \mathbb{E}\left[\phi_u(|h(X_i^1)|)\right] \right| + \frac{\delta}{\sqrt{mn}}.
\end{aligned}
$$

By the Lipschitz property of $\phi_u(|t|)$ and the symmetrization theorem, we have

$$
\mathbb{E} \sup_{h \in \mathcal{H}} \left| \frac{1}{mn} \sum_{i=1}^m \sum_{j=1}^n \phi_u(|h(X_i^j)|) - \frac{1}{m} \sum_{i=1}^m \mathbb{E}\left[\phi_u(|h(X_i^1)|)\right] \right| \leq \frac{4}{u} \mathbb{E} \sup_{h \in \mathcal{H}} \left| \frac{1}{mn} \sum_{i=1}^m \sum_{j=1}^n \sigma_i^j h(X_i^j) \right|.
$$

Therefore, for every $h \in \mathcal{H}$, it follows that with probability at least $1 - 2\exp(-2\delta^2)$, we have

$$
\frac{1}{mn} \sum_{i=1}^m \sum_{j=1}^n \mathbb{1}_{\{|h(X_i^j)| \geq u\}} \geq \inf_{h \in \mathcal{H}} \mathbb{P}(|h(X_i^1)| \geq 2u) - \frac{4}{u} \mathbb{E} \sup_{h \in \mathcal{H}} \left| \frac{1}{mn} \sum_{i=1}^m \sum_{j=1}^n \sigma_i^j h(X_i^j) \right| - \frac{\delta}{\sqrt{mn}}. \quad (30)
$$

**Step Two:**

The first term on the RHS can be bounded by small ball condtion. Let $\mathcal{H}^* = \mathcal{H} - \widehat{h}^*$. We first prove that $\mathcal{H}^*$ is star-shaped around 0. For every $h - \widehat{h}^* \in \mathcal{H}^*$ and $0 \leq \lambda \leq 1$, we have $\lambda(h - \widehat{h}^*) = \lambda h + (1 - \lambda)\widehat{h}^* - \widehat{h}^*$. Since $\mathcal{H}$ is convex, it follows that $\lambda h + (1 - \lambda)\widehat{h}^* \in \mathcal{H}$. Then the claim follows because $\lambda(h - \widehat{h}^*) \in \mathcal{H} - \widehat{h}^*$.

Assume that these exsits $\tau > 0$ for which $Q_{\mathcal{H}^*}(2\tau) > 0$. The for every $s \geq \omega(\mathcal{H}^*, \tau Q_{\mathcal{H}^*}(2\tau)/16)$, we have

$$
\mathbb{E} \sup_{h \in \mathcal{H}^* \cap sB_2^m} \left| \frac{1}{mn} \sum_{i=1}^m \sum_{j=1}^n \sigma_i^j h(X_i^j) \right| \leq \frac{\tau Q_{\mathcal{H}^*}(2\tau)}{16} s.
$$

Let $\mathcal{G}$ be a function class associated with $\mathcal{H}^*$

$$
\mathcal{G} = \left\{ \frac{h}{s} : h \in \mathcal{H}^* \cap sB_2^m \right\} \subset B_2^m,
$$

where $B_2^m$ is the unit ball with respect with $L_2(D)$ and $D$ is the semi-empirical distribution.

$$\mathbb{E}\sup_{g\in\mathcal{G}}\left|\frac{1}{mn}\sum_{i=1}^{m}\sum_{j=1}^{n}\sigma_i^j g(X_i^j)\right| = \mathbb{E}\sup_{h\in\mathcal{H}^*\cap sB_2^m}\left|\frac{1}{mn}\sum_{i=1}^{m}\sum_{j=1}^{n}\sigma_i^j\frac{h(X_i^j)}{s}\right| \leq \frac{\tau Q_{\mathcal{H}^*}(2\tau)}{16} \leq \frac{\tau Q_{\mathcal{G}}(2\tau)}{16},$$

where the last inequality follows from $Q_{\mathcal{G}}(2\tau) \geq Q_{\mathcal{H}^*}(2\tau)$.

By equation (30) applied to the function class $\mathcal{G}$, it follows that with probability at least $1 - 2\exp(-2\delta^2)$

$$\frac{1}{mn}\sum_{i=1}^{m}\sum_{j=1}^{n}\mathbb{1}_{\{|g(X_i^j)|\geq u\}} \geq \inf_{g\in\mathcal{G}}\mathbb{P}(|g(X_i^1)|\geq 2u) - \frac{4}{u}\mathbb{E}\sup_{g\in\mathcal{G}}\left|\frac{1}{mn}\sum_{i=1}^{m}\sum_{j=1}^{n}\sigma_i^j g(X_i^j)\right| - \frac{\delta}{\sqrt{mn}}$$

$$\geq Q_G(2u) - \frac{4}{u}\mathbb{E}\sup_{g\in\mathcal{G}}\left|\frac{1}{mn}\sum_{i=1}^{m}\sum_{j=1}^{n}\sigma_i^j g(X_i^j)\right| - \frac{\delta}{\sqrt{mn}}$$

$$\geq Q_G(2u) - \frac{4}{u}\frac{\tau Q_{\mathcal{G}}(2\tau)}{16} - \frac{\delta}{\sqrt{mn}}.$$

Now, setting

$$u = \tau \quad \delta = \frac{\sqrt{mn}Q_{\mathcal{G}}(2\tau)}{2},$$

it follows that with probability at least $1 - 2\exp(-mnQ_{\mathcal{G}}(2\tau)/2)$

$$\frac{1}{mn}\sum_{i=1}^{m}\sum_{j=1}^{n}\mathbb{1}_{\{|g(X_i^j)|\geq\tau\}} \geq Q_G(2\tau) - \frac{Q_{\mathcal{G}}(2\tau)}{4} - \frac{Q_{\mathcal{G}}(2\tau)}{2} = \frac{Q_{\mathcal{G}}(2\tau)}{4}.$$

Using the condition $Q_{\mathcal{G}}(2\tau) \geq Q_{\mathcal{H}^*}(2\tau)$, for every $s \geq \omega(\mathcal{H}^*, \tau Q_{\mathcal{H}^*}(2\tau)/16)$, it follows that with probability at least $1 - 2\exp(-mnQ_{\mathcal{H}^*}(2\tau)/2)$, we get

$$\inf_{g\in\mathcal{G}}|\{(i,j): |g(X_i^j)| > \tau\}| = \sum_{i=1}^{m}\sum_{j=1}^{n}\mathbb{1}_{\{|g(X_i^j)|\geq\tau\}} \geq \frac{mnQ_{\mathcal{G}}(2\tau)}{4} \geq \frac{mnQ_{\mathcal{H}^*}(2\tau)}{4}. \tag{31}$$

For every $h - \widehat{h}^* \in \mathcal{H}^*$ that satisfies $\|h - \widehat{h}^*\|_{L_2(D)} \geq s$, since $\mathcal{H} - \widehat{h}^*$ is star-shaped around 0, we have $\left(s/\|h - \widehat{h}^*\|_{L_2(D)}\right)(h - \widehat{h}^*) \in \mathcal{H}^* \cap sB_2^m$. Thus, $(h - \widehat{h}^*)/\|h - \widehat{h}^*\|_{L_2(D)} \in \mathcal{G}$.

Combining equation (31), if $s \geq \omega(\mathcal{H}^*, \tau Q_{\mathcal{H}^*}(2\tau)/16)$, then for every $h \in \mathcal{H}$ that satisfies $\|h - \widehat{h}^*\|_{L_2(D)} \geq s$, if follows that with probability at least $1 - 2\exp(-mnQ_{\mathcal{H}^*}(2\tau)/2)$, one has

$$\left|\{(i,j): |(h - \widehat{h}^*)(X_i^j)| > \tau\|h - \widehat{h}^*\|_{L_2(D)}\}\right| \geq \frac{mnQ_{\mathcal{H}^*}(2\tau)}{4}$$

Therefore, on that event, we get

$$\frac{1}{mn}\sum_{i=1}^{m}\sum_{j=1}^{n}(h - \widehat{h}^*)^2(X_i^j) > \frac{\tau^2}{4}Q_{\mathcal{H}^*}(2\tau)\|h - \widehat{h}^*\|_{L_2(D)}^2. \tag{32}$$

Note that $\mathcal{H}^* \subset \mathcal{H} - \mathcal{H}$, the same conclusion holds with $Q_{\mathcal{H}-\mathcal{H}}(2\tau)$ replacing the larger $Q_{\mathcal{H}^*}$. That is, if $s \geq \omega(\mathcal{H}^*, \tau Q_{\mathcal{H}-\mathcal{H}}(2\tau)/16)$, then for every $h \in \mathcal{H}$ that satisfies $\|h - \widehat{h}^*\|_{L_2(D)} \geq s$, if follows that with probability at least $1 - 2\exp(-mnQ_{\mathcal{H}-\mathcal{H}}(2\tau)/2)$, one has

$$\frac{1}{mn}\sum_{i=1}^{m}\sum_{j=1}^{n}(h - \widehat{h}^*)^2(X_i^j) > \frac{\tau^2}{4}Q_{\mathcal{H}-\mathcal{H}}(2\tau)\|h - \widehat{h}^*\|_{L_2(D)}^2.$$

**Step Three:**

Combining equation (2), with the same conditions we get

$$\mathcal{L}_S(h) - \mathcal{L}_S(\widehat{h}^*) \geq \frac{2}{mn}\sum_{i=1}^{m}\sum_{j=1}^{n}\xi_i^j(h - \widehat{h}^*)(X_i^j) + \frac{\tau^2}{4}Q_{\mathcal{H}-\mathcal{H}}(2\tau)\|h - \widehat{h}^*\|_{L_2(D)}^2. \tag{33}$$

Since $\frac{1}{m}\sum_{i=1}^m \mathbb{E}\left[\xi(h-\widehat{h}^*)(X_i^1)\right] \geq 0$, then we have

$$\frac{1}{mn}\sum_{i=1}^m\sum_{j=i}^n \xi_i^j(h-\widehat{h}^*)(X_i^j) \geq \frac{1}{mn}\sum_{i=1}^m\sum_{j=1}^n \xi_i^j(h-\widehat{h}^*)(X_i^j) - \frac{1}{m}\sum_{i=1}^m \mathbb{E}_{X\sim D_i}\left[\xi(h-\widehat{h}^*)(X)\right].$$

According to Lemma 7, we have

$$\mathbb{P}\left(\left|\frac{1}{mn}\sum_{i=1}^m\sum_{j=1}^n \xi_i^j(h-\widehat{h}^*)(X_i^j) - \frac{1}{m}\sum_{i=1}^m \mathbb{E}_{X\sim D_i}\left[\xi(h-\widehat{h}^*)(X)\right]\right| \geq \eta\|h-\widehat{h}^*\|_{L_2(D)}^2\right)$$

$$\leq 2\exp\left\{-\left(\frac{c_1\eta_0^2(mn)\|h-\widehat{h}^*\|_{L_2(D)}^4}{\frac{1}{mn}\sum_{i=1}^m\sum_{j=1}^n 2\left\|V_i^j\right\|_{\psi_\alpha}^2} \wedge \frac{c_2\eta_0^\alpha(mn)^\alpha\|h-\widehat{h}^*\|_{L_2(D)}^{2\alpha}}{\max_{(1,1)\leq(i,j)\leq(m,n)}\left\|V_i^j\right\|_{\psi_\alpha}^\alpha}\right)\right\},$$

where

$$V_i^j = \xi_i^j(h-\widehat{h}^*)(X_i^j) - \mathbb{E}\left[\xi_i^j(h-\widehat{h}^*)(X_i^j)\right].$$

We denote $C_1, C_2$ by

$$C_1 = \frac{\frac{1}{mn}\sum_{i=1}^m\sum_{j=1}^n 2\left\|V_i^j\right\|_{\psi_\alpha}^2}{c_1}, \qquad C_2 = \frac{\max_{(1,1)\leq(i,j)\leq(m,n)}\left\|V_i^j\right\|_{\psi_\alpha}^\alpha}{c_2}.$$

Then we have

$$\mathbb{P}\left(\left|\frac{1}{mn}\sum_{i=1}^m\sum_{j=1}^n \xi_i^j(h-\widehat{h}^*)(X_i^j) - \frac{1}{m}\sum_{i=1}^m \mathbb{E}_{X\sim D_i}\left[\xi(h-\widehat{h}^*)(X)\right]\right| \geq \eta\|h-\widehat{h}^*\|_{L_2(D)}^2\right)$$

$$\leq 2\exp\left\{-\left(\frac{\eta^2(mn)\|h-\widehat{h}^*\|_{L_2(D)}^4}{C_1} \wedge \frac{\eta^\alpha(mn)^\alpha\|h-\widehat{h}^*\|_{L_2(D)}^{2\alpha}}{C_2}\right)\right\}.$$

To make sure that the probability tends to zero as $mn$ increase, we could chose $\|h-\widehat{h}^*\|_{L_2(D)} \geq \kappa = (mn)^{-\frac{1}{4}+\iota}$, where $0 < \iota < \frac{1}{4}$.

Then with probability at least

$$\delta = 1 - 2\exp\left\{-\left(\frac{\eta^2(mn)^{4\iota}}{C_1} \wedge \frac{\eta^\alpha(mn)^{\frac{\alpha(1+4\iota)}{2}}}{C_2}\right)\right\},$$

we have

$$\left|\frac{1}{mn}\sum_{i=1}^m\sum_{j=1}^n \xi_i^j(h-\widehat{h}^*)(X_i^j) - \frac{1}{m}\sum_{i=1}^m \mathbb{E}_{X\sim D_i}\left[\xi(h-\widehat{h}^*)(X)\right]\right| \leq \eta\|h-\widehat{h}^*\|_{L_2(D)}^2. \tag{34}$$

Combining (2) and (34), if $\|h-\widehat{h}^*\|_{L_2(D)} \geq \max(\kappa, s)$, it follows that with probability at least $1 - \delta - 2\exp(-mnQ_{\mathcal{H}^*}(2\tau)/2)$,

$$\mathcal{L}_S(h) - \mathcal{L}_S(\widehat{h}^*) \geq \|h-\widehat{h}^*\|_{L_2(D)}^2 \left(\frac{\tau^2}{4}Q_{\mathcal{H}-\mathcal{H}}(2\tau) - 4\eta\right).$$

Consider $\eta < \tau^2 Q_{\mathcal{H}-\mathcal{H}}(2\tau)/16$, we get

$$\mathcal{L}_S(h) - \mathcal{L}_S(\widehat{h}^*) \geq 0.$$

On the same event, we have

$$\|\widehat{h}-\widehat{h}^*\|_{L_2(D)} \leq \max(\kappa, s) = \max((mn)^{-\frac{1}{4}+\iota}, \omega(\mathcal{H}-\mathcal{H}, \tau Q_{\mathcal{H}-\mathcal{H}}(2\tau)/16)).$$

$\square$

PROOF OF COROLLARY 1

*Proof.* Let $s = \omega(\mathcal{H} - \mathcal{H}, \tau Q_{\mathcal{H}-\mathcal{H}}(2\tau)/16)$ and $\kappa = (mn)^{-\frac{1}{4}+\iota}$, where $0 < \iota < \frac{1}{4}$. By theorem 4, we have

$$\|\widehat{h} - \widehat{h}^*\|_{L_2(D)} \leq \max(\kappa, s).$$

Therefore, it suffices to show that for quadratic loss we have

$$\mathcal{L}_D(h) - \mathcal{L}_D(\widehat{h}^*) = \frac{1}{m} \sum_{i=1}^{m} \sum_{j=1}^{n} \mathbb{E}\left[(h(X_i^j) - Y_i^j)^2 - (\widehat{h}^*(X_i^j) - Y_i^j)^2\right] \tag{35}$$

$$= \frac{1}{mn} \sum_{i=1}^{m} \sum_{j=1}^{n} \mathbb{E}\left[(h - \widehat{h}^*)^2(X_i^j)\right] + \frac{2}{mn} \sum_{i=1}^{m} \sum_{j=1}^{n} \mathbb{E}\left[\xi_i^j(h - \widehat{h}^*)(X_i^j)\right], \tag{36}$$

Note that for $h \in \mathcal{H}$ either

$$\mathcal{L}_D(h) - \mathcal{L}_D(\widehat{h}^*) \leq \frac{2}{mn} \sum_{i=1}^{m} \sum_{j=1}^{n} \mathbb{E}\left[(h - \widehat{h}^*)^2(X_i^j)\right],$$

or

$$\mathcal{L}_D(h) - \mathcal{L}_D(\widehat{h}^*) \leq \frac{4}{mn} \sum_{i=1}^{m} \sum_{j=1}^{n} \mathbb{E}\left[\xi_i^j(h - \widehat{h}^*)(X_i^j)\right].$$

For the first case, we have

$$\mathcal{L}_D(\widehat{h}) - \mathcal{L}_D(\widehat{h}^*) \leq 2\|\widehat{h} - \widehat{h}^*\|_{L_2(D)}^2 \leq 2\max(\kappa^2, s^2).$$

For the second case,

$$\mathcal{L}_S(h) - \mathcal{L}_S(\widehat{h}^*) = \frac{1}{mn} \sum_{i=1}^{m} \sum_{j=1}^{n} \left[(h(X_i^j) - Y_i^j)^2 - (\widehat{h}^*(X_i^j) - Y_i^j)^2\right]$$

$$= \frac{1}{mn} \sum_{i=1}^{m} \sum_{j=1}^{n} (h - \widehat{h}^*)^2(X_i^j) + \frac{2}{mn} \sum_{i=1}^{m} \sum_{j=1}^{n} \xi_i^j(h - \widehat{h}^*)(X_i^j)$$

$$\geq \frac{2}{mn} \sum_{i=1}^{m} \sum_{j=1}^{n} \mathbb{E}\left[\xi_i^j(h - \widehat{h}^*)(X_i^j)\right]$$

$$+ \frac{2}{mn} \sum_{i=1}^{m} \sum_{j=1}^{n} \xi_i^j(h - \widehat{h}^*)(X_i^j) - \frac{2}{mn} \sum_{i=1}^{m} \sum_{j=1}^{n} \mathbb{E}\left[\xi_i^j(h - \widehat{h}^*)(X_i^j)\right].$$

For convex function class $\mathcal{H}$, we have $\mathbb{E}[\xi_i^j(h - \widehat{h}^*)(X_i^j)] \geq 0$. On the same condition that inequality (34) holds, we get

$$\mathcal{L}_S(h) - \mathcal{L}_S(\widehat{h}^*) \geq \frac{2}{mn} \sum_{i=1}^{m} \sum_{j=1}^{n} \mathbb{E}\left[\xi_i^j(h - \widehat{h}^*)(X_i^j)\right]$$

$$- \left|\frac{2}{mn} \sum_{i=1}^{m} \sum_{j=1}^{n} \xi_i^j(h - \widehat{h}^*)(X_i^j) - \frac{2}{mn} \sum_{i=1}^{m} \sum_{j=1}^{n} \mathbb{E}\left[\xi_i^j(h - \widehat{h}^*)(X_i^j)\right]\right|$$

$$\geq \frac{1}{2}(\mathcal{L}_D(h) - \mathcal{L}_D(\widehat{h}^*)) - \frac{\tau^2 Q_{\mathcal{H}-\mathcal{H}}(2\tau)}{8}\|\widehat{h} - \widehat{h}^*\|_{L_2(D)}^2$$

$$\geq \frac{1}{2}(\mathcal{L}_D(h) - \mathcal{L}_D(\widehat{h}^*)) - \frac{\tau^2 Q_{\mathcal{H}-\mathcal{H}}(2\tau)}{8}\max(\kappa^2, s^2)$$

Since $\mathcal{L}_S(h) - \mathcal{L}_S(\widehat{h}^*) \leq 0$, it follows that

$$(\mathcal{L}_D(h) - \mathcal{L}_D(\widehat{h}^*)) \leq \frac{\tau^2 Q_{\mathcal{H}-\mathcal{H}}(2\tau)}{4}\max(\kappa^2, s^2).$$

$\square$

### E.2 PROOF OF THEOREM 5

*Proof.* For quadratic loss, we have

$$\mathcal{L}_D(h) - \mathcal{L}_D(\widehat{h}^*) = \frac{1}{m} \sum_{i=1}^{m} \mathbb{E}\left[(h(X_i^1) - Y_i^1)^2 - (\widehat{h}^*(X_i^1) - Y_i^1)^2\right] \tag{37}$$

$$= \frac{1}{m} \sum_{i=1}^{m} \mathbb{E}[(h - \widehat{h}^*)^2(X_i^1)] + \frac{2}{m} \sum_{i=1}^{m} \mathbb{E}[\xi_i^1(h - \widehat{h}^*)(X_i^1)], \tag{38}$$

**Step One:**

For any $h \in \mathcal{H}$ and $u > 0$,

$$|\{i : |\mathbb{E}[h(X_i^1)]| > u\}| = \sum_{i=1}^{m} \mathbb{1}_{\{|\mathbb{E}[h(X_i^1)]| \geq u\}}$$

Also,

$$\frac{1}{m} \sum_{i=1}^{m} \mathbb{1}_{\{|\mathbb{E}[h(X_i^1)]| \geq u\}} = \mathop{\mathbb{E}}_{D_i \sim P}\left[\mathbb{1}_{\{|\mathbb{E}[h(X_i^1)]| \geq 2u\}}\right]$$

$$+ \frac{1}{m} \sum_{i=1}^{m} \mathbb{1}_{\{|\mathbb{E}[h(X_i^1)]| \geq u\}} - \mathop{\mathbb{E}}_{D_i \sim P}\left[\mathbb{1}_{\{|\mathbb{E}[h(X_i^1)]| \geq 2u\}}\right]$$

Let $\phi_u : \mathbb{R} \to [0, 1]$ be the function

$$\phi_u(t) = \begin{cases} 0 & t \leq u, \\ \frac{t}{u} - 1 & u \leq t \leq 2u, \\ 1 & t \geq 2u. \end{cases} \tag{39}$$

Note that for every $t \in \mathbb{R}$, $\phi_u(t) \geq \mathbb{1}_{\{t \geq 2u\}}$ and $\phi_u(t) \leq \mathbb{1}_{\{t \geq u\}}$. Thus,

$$\frac{1}{m} \sum_{i=1}^{m} \mathbb{1}_{\{|\mathbb{E}[h(X_i^1)]| \geq u\}} \geq \mathop{\mathbb{E}}_{D_i \sim P}\left[\mathbb{1}_{\{|\mathbb{E}[h(X_i^1)]| \geq 2u\}}\right]$$

$$+ \frac{1}{m} \sum_{i=1}^{m} \phi_u(|\mathbb{E}[h(X_i^1)]|) - \mathop{\mathbb{E}}_{D_i \sim P}\left[\phi_u(|\mathbb{E}[h(X_i^1)]|)\right]$$

$$\geq \inf_{h \in \mathcal{H}} \mathbb{P}(|\mathbb{E}[h(X_i^1)]| \geq 2u)$$

$$- \sup_{h \in \mathcal{H}} \left| \frac{1}{m} \sum_{i=1}^{m} \phi_u(|\mathbb{E}[h(X_i^1)]|) - \mathop{\mathbb{E}}_{D_i \sim P}\left[\phi_u(|\mathbb{E}[h(X_i^1)]|)\right] \right|.$$

Since function $\phi_u(t)$ is bounded by 1, using Mcdiarmid's inequality, we get that, for every $\delta > 0$, with probability at least $1 - 2\exp(-2\delta^2)$,

$$\sup_{h \in \mathcal{H}} \left| \frac{1}{m} \sum_{i=1}^{m} \phi_u(|\mathbb{E}[h(X_i^1)]|) - \mathop{\mathbb{E}}_{D_i \sim P}\left[\phi_u(|\mathbb{E}[h(X_i^1)]|)\right] \right|$$

$$\leq \mathbb{E} \sup_{h \in \mathcal{H}} \left| \frac{1}{m} \sum_{i=1}^{m} \phi_u(|\mathbb{E}[h(X_i^1)]|) - \mathop{\mathbb{E}}_{D_i \sim P}\left[\phi_u(|\mathbb{E}[h(X_i^1)]|)\right] \right| + \frac{\delta}{\sqrt{m}}.$$

By the Lipschitz property of $\phi_u(|t|)$ and the symmetrization theorem, we have

$$\mathbb{E} \sup_{h \in \mathcal{H}} \left| \frac{1}{m} \sum_{i=1}^{m} \phi_u(|\mathbb{E}[h(X_i^1)]|) - \mathop{\mathbb{E}}_{D_i \sim P}\left[\phi_u(|\mathbb{E}[h(X_i^1)]|)\right] \right| \leq \frac{4}{u} \mathbb{E} \sup_{h \in \mathcal{H}} \left| \frac{1}{m} \sum_{i=1}^{m} \sigma_i h(X_i) \right|,$$

where $X_i$ is the random sampled across two-level distribution framework.

Therefore, for every $h \in \mathcal{H}$, it follows that with probability at least $1 - 2\exp(-2\delta^2)$, we have

$$\frac{1}{m}\sum_{i=1}^{m}\mathbb{1}_{\{|\mathbb{E}[h(X_i^1)]|\geq u\}} \geq \inf_{h\in\mathcal{H}}\mathbb{P}(|\mathbb{E}[h(X_i^1)]|\geq 2u) - \frac{4}{u}\mathbb{E}\sup_{h\in\mathcal{H}}\left|\frac{1}{m}\sum_{i=1}^{m}\sigma_i h(X_i)\right| - \frac{\delta}{\sqrt{m}}. \quad (40)$$

The first term on the RHS can be bounded by small ball condtion. Let $\mathcal{H}^* = \mathcal{H} - h^*$, we have proved that $\mathcal{H}^*$ is star-shaped around 0.

Assume that these exsits $\tau > 0$ for which $Q_{\mathcal{H}^*}(2\tau, P) > 0$. Then for every $s \geq \omega_m(\mathcal{H}^*, \tau Q_{\mathcal{H}^*}(2\tau, P)/16)$, we have

$$\mathbb{E}\sup_{h\in\mathcal{H}^*\cap sB_m}\left|\frac{1}{m}\sum_{i=1}^{m}\sigma_i h(X_i)\right| \leq \frac{\tau Q_{\mathcal{H}^*}(2\tau, P)}{16}s.$$

Let $\mathcal{G}$ be a function class associated with $\mathcal{H}^*$

$$\mathcal{G} = \left\{\frac{h}{s} : h \in \mathcal{H}^* \cap sB_m\right\} \subset B_m,$$

where $B_m$ is the unit ball with respect with $L_2(P)$ and $P$ is the population distribution.

$$\mathbb{E}\sup_{g\in\mathcal{G}}\left|\frac{1}{m}\sum_{i=1}^{m}\sigma_i g(X_i)\right| = \mathbb{E}\sup_{h\in\mathcal{H}^*\cap sB_m}\left|\frac{1}{m}\sum_{i=1}^{m}\sigma_i \frac{h(X_i)}{s}\right| \leq \frac{\tau Q_{\mathcal{H}^*}(2\tau, P)}{16} \leq \frac{\tau Q_{\mathcal{G}}(2\tau, P)}{16},$$

where the last inequality follows from $Q_{\mathcal{G}}(2\tau, P) \geq Q_{\mathcal{H}^*}(2\tau, P)$. By applying inequality (40) to the function class $\mathcal{G}$, it follows that with probability at least $1 - 2\exp(-2\delta^2)$

$$\frac{1}{m}\sum_{i=1}^{m}\mathbb{1}_{\{|\mathbb{E}[g(X_i^1)]|\geq u\}} \geq \inf_{g\in\mathcal{G}}\mathbb{P}(|\mathbb{E}[g(X_i^1)]|\geq 2u) - \frac{4}{u}\mathbb{E}\sup_{g\in\mathcal{G}}\left|\frac{1}{m}\sum_{i=1}^{m}\sigma_i g(X_i)\right| - \frac{\delta}{\sqrt{m}}.$$

$$\geq Q_{\mathcal{G}}(2u, P) - \frac{4}{u}\mathbb{E}\sup_{g\in\mathcal{G}}\left|\frac{1}{m}\sum_{i=1}^{m}\sigma_i g(X_i)\right| - \frac{\delta}{\sqrt{m}}$$

$$\geq Q_{\mathcal{G}}(2u, P) - \frac{4}{u}\frac{\tau Q_{\mathcal{G}}(2\tau, P)}{16} - \frac{\delta}{\sqrt{m}}.$$

Now, setting

$$u = \tau \quad \delta = \frac{\sqrt{m}Q_{\mathcal{G}}(2\tau, P)}{2},$$

it follows that with probability at least $1 - 2\exp(-mQ_{\mathcal{G}}(2\tau, P)/2)$

$$\frac{1}{m}\sum_{i=1}^{m}\mathbb{1}_{\{|\mathbb{E}[g(X_i^1)]|\geq u\}} \geq Q_G(2\tau, P) - \frac{Q_{\mathcal{G}}(2\tau, P)}{4} - \frac{Q_{\mathcal{G}}(2\tau, P)}{2} = \frac{Q_{\mathcal{G}}(2\tau, P)}{4}.$$

Using the condition $Q_{\mathcal{G}}(2\tau, P) \geq Q_{\mathcal{H}^*}(2\tau, P)$, for every $s \geq \omega_m(\mathcal{H}^*, \tau Q_{\mathcal{H}^*}(2\tau, P)/16)$, it follows that with probability at least $1 - 2\exp(-mQ_{\mathcal{H}^*}(2\tau, P)/2)$, we get

$$\inf_{g\in\mathcal{G}}|\{i : |\mathbb{E}[g(X_i^1)]| > \tau\}| = \sum_{i=1}^{m}\mathbb{1}_{\{|\mathbb{E}[g(X_i^1)]|\geq\tau\}} \geq \frac{mQ_{\mathcal{G}}(2\tau, P)}{4} \geq \frac{mQ_{\mathcal{H}^*}(2\tau, P)}{4}. \quad (41)$$

For every $h - h^* \in \mathcal{H}^*$ that satisfies $\|h - h^*\|_{L_2(P)} \geq s$, since $\mathcal{H} - h^*$ is star-shaped around 0, we have $\left(s/\|h - h^*\|_{L_2(P)}\right)(h - h^*) \in \mathcal{H}^* \cap sB_m$. Thus, $(h - h^*)/\|h - h^*\|_{L_2(P)} \in \mathcal{G}$.

Combining equation (41), if $s \geq \omega_m(\mathcal{H}^*, \tau Q_{\mathcal{H}^*}(2\tau, P)/16)$, then for every $h \in \mathcal{H}$ that satisfies $\|h - h^*\|_{L_2(P)} \geq s$, if follows that with probability at least $1 - 2\exp(-mQ_{\mathcal{H}^*}(2\tau, P)/2)$, one has

$$\left|\{i : |\mathbb{E}[(h - h^*)(X_i^1)]| > \tau\|h - h^*\|_{L_2(P)}\}\right| \geq \frac{mQ_{\mathcal{H}^*}(2\tau, P)}{4}$$

Therefore, on that event, we get

$$\frac{1}{m}\sum_{i=1}^{m}\mathbb{E}\left[(h - h^*)^2(X_i^1)\right] \geq \frac{1}{m}\sum_{i=1}^{m}\left|\mathbb{E}[(h - h^*)(X_i^1)]\right|^2 \geq \frac{\tau^2}{4}Q_{\mathcal{H}^*}(2\tau, P)\|h - h^*\|_{L_2(P)}^2. \quad (42)$$

Note that $\mathcal{H}^* \subset \mathcal{H} - \mathcal{H}$, the same conclusion holds with $Q_{\mathcal{H}-\mathcal{H}}(2\tau, P)$ replacing the larger $Q_{\mathcal{H}^*}(2\tau, P)$. That is, if $s \geq \omega_m(\mathcal{H}^*, \tau Q_{\mathcal{H}-\mathcal{H}}(2\tau, P)/16)$, then for every $h \in \mathcal{H}$ that satisfies $\|h - h^*\|_{L_2(P)} \geq s$, if follows that with probability at least $1 - 2\exp(-mQ_{\mathcal{H}-\mathcal{H}}(2\tau, P)/2)$, one has

$$\frac{1}{m}\sum_{i=1}^{m} \mathbb{E}\left[(h - h^*)^2(X_i^1)\right] > \frac{\tau^2}{4}Q_{\mathcal{H}-\mathcal{H}}(2\tau, P)\|h - h^*\|_{L_2(P)}^2.$$

With the same conditions we get

$$\mathcal{L}_P(h) - \mathcal{L}_P(h^*) \geq \frac{2}{m}\sum_{i=1}^{m}\mathbb{E}\left[\xi_i^1(h - h^*)(X_i^1)\right] + \frac{\tau^2}{4}Q_{\mathcal{H}-\mathcal{H}}(2\tau, P)\|h - h^*\|_{L_2(P)}^2. \tag{43}$$

Since $\mathbb{E}_{D_i \sim P}\mathbb{E}_{(X_i^1, Y_i^1) \sim D_i}\left[\xi_i^1(h - h^*)(X_i^1)\right] \geq 0$, then we have

$$\frac{1}{m}\sum_{i=1}^{m}\mathbb{E}_{(X_i^1, Y_i^1) \sim D_i}\left[\xi_i^1(h - h^*)(X_i^1)\right] \geq \frac{1}{m}\sum_{i=1}^{m}\mathbb{E}_{(X_i^1, Y_i^1) \sim D_i}\left[\xi_i^1(h - h^*)(X_i^1)\right]$$
$$- \mathbb{E}_{D_i \sim P}\mathbb{E}_{(X_i, Y_i) \sim D_i}\left[\xi_i^1(h - h^*)(X_i^1)\right].$$

According to Lemma 7, we have

$$\mathbb{P}\left(\left|\frac{1}{m}\sum_{i=1}^{m}\mathbb{E}_{(X_i^1, Y_i^1) \sim D_i}\left[\xi_i^1(h - h^*)(X_i^1)\right] - \mathbb{E}_{D_i \sim P}\mathbb{E}_{(X_i, Y_i) \sim D_i}\left[\xi(h - h^*)(X_i)\right]\right| \geq \eta\|h - h^*\|_{L_2(P)}^2\right)$$
$$\leq 2\exp\left\{-\left(\frac{c_1\eta^2 m\|h - h^*\|_{L_2(P)}^4}{\frac{1}{m}\sum_{i=1}^{m}\|V_i\|_{\psi_\alpha}^2} \wedge \frac{c_2\eta^\alpha m^\alpha\|h - h^*\|_{L_2(P)}^{2\alpha}}{\max_{1 \leq i \leq m}\|V_i\|_{\psi_\alpha}^\alpha}\right)\right\},$$

where

$$V_i = \mathbb{E}_{(X_i^1, Y_i^1) \sim D_i}\left[\xi_i^1(h - h^*)(X_i^1)\right] - \mathbb{E}_{D_i \sim P}\mathbb{E}_{(X_i, Y_i) \sim D_i}\left[\xi(h - h^*)(X_i)\right].$$

We denote $C_1, C_2$ by

$$C_1 = \frac{\frac{1}{m}\sum_{i=1}^{m}\|V_i\|_{\psi_\alpha}^2}{c_1}, \qquad C_2 = \frac{\max_{1 \leq i \leq m}\|V_i\|_{\psi_\alpha}^\alpha}{c_2}.$$

Then we have

$$\mathbb{P}\left(\left|\frac{1}{m}\sum_{i=1}^{m}\mathbb{E}_{(X_i^1, Y_i^1) \sim D_i}\left[\xi_i^1(h - h^*)(X_i^1)\right] - \mathbb{E}_{D_i \sim P}\mathbb{E}_{(X_i, Y_i) \sim D_i}\left[\xi(h - h^*)(X_i)\right]\right| \geq \eta\|h - h^*\|_{L_2(P)}^2\right)$$
$$\leq 2\exp\left\{-\left(\frac{\eta^2 m\|h - h^*\|_{L_2(P)}^4}{C_1} \wedge \frac{\eta^\alpha m^\alpha\|h - h^*\|_{L_2(P)}^{2\alpha}}{C_2}\right)\right\}.$$

To make sure that the probability tends to zero as $m$ increase, we could chose $\|h - h^*\|_{L_2(P)} \geq \kappa = m^{-\frac{1}{4}+\iota}$, where $0 < \iota < \frac{1}{4}$.

If $\|h - h^*\|_{L_2(P)} \geq \max(\kappa, s)$, it follows that with probability at least $1 - \delta_m - 2\exp(-mQ_{\mathcal{H}^*}(2\tau, P)/2)$,

$$\mathcal{L}_D(h) - \mathcal{L}_D(h^*) \geq \|h - h^*\|_{L_2(P)}^2\left(\frac{\tau^2}{4}Q_{\mathcal{H}-\mathcal{H}}(2\tau, P) - 4\eta\right),$$

where $\delta_m = \exp\{-(\frac{c_1\eta^2 m^{4\iota}}{\frac{1}{m}\sum_{i=1}^{m}\|V_i\|_{\psi_\alpha}^2} \wedge \frac{c_2\eta^\alpha m^{\alpha(1/2+2\iota)}}{\max_{1 \leq i \leq m}\|V_i\|_{\psi_\alpha}^\alpha})\}$ Consider $\eta < \tau^2 Q_{\mathcal{H}-\mathcal{H}}(2\tau, P)/16$, we get

$$\mathcal{L}_D(h) - \mathcal{L}_D(h^*) \geq 0.$$

On the same event, we have

$$\|h - h^*\|_{L_2(P)} \leq \max(\kappa, s) = \max(m^{-\frac{1}{4}+\iota}, \omega_m(\mathcal{H} - \mathcal{H}, \tau Q_{\mathcal{H}-\mathcal{H}}(2\tau)/16)). \tag{44}$$

$\square$

PROOF OF COROLLARY 2

*Proof.* Since excess risk is defined across our two-level framework, the steps in the proof of Corollary 1 can not be applied directly to derive Corollary 2. The key step to derive Corollary 2 is to bound $\|\widehat{h} - h^*\|^2_{L_2(P)}$. First, this term can be decomposed as

$$\|\widehat{h} - h^*\|^2_{L_2(P)} \leq 2\|\widehat{h} - \widehat{h}^*\|^2_{L_2(P)} + 2\|\widehat{h}^* - h^*\|^2_{L_2(P)}.$$

Note that $\|\widehat{h}^* - h^*\|^2_{L_2(P)}$ has been bounded by Theorem 5. To bound $\|\widehat{h} - \widehat{h}^*\|^2_{L_2(P)}$, we use the following decomposition:

$$\|\widehat{h} - \widehat{h}^*\|^2_{L_2(P)} = \|\widehat{h} - \widehat{h}^*\|^2_{L_2(P)} - \|\widehat{h} - \widehat{h}^*\|^2_{L_2(D)} + \|\widehat{h} - \widehat{h}^*\|^2_{L_2(D)}.$$

Note that $\|\widehat{h} - \widehat{h}^*\|^2_{L_2(D)}$ has been bounded by Theorem 4.

According to Lemma 7, we have

$$\mathbb{P}\left(\left|\frac{1}{m}\sum_{i=1}^{m}\mathop{\mathbb{E}}_{(X_i,Y_i)\sim D_i}\left[(h-\widehat{h}^*)^2(X_i)\right] - \mathbb{E}_{D_i\sim P}\mathbb{E}_{(X_i,Y_i)\sim D_i}\left[(h-\widehat{h}^*)^2(X_i)\right]\right| \geq \eta\|h-\widehat{h}^*\|^2_{L_2(P)}\right)$$

$$\leq 2\exp\left\{-\left(\frac{c_1\eta^2 m\|h-\widehat{h}^*\|^4_{L_2(P)}}{\frac{1}{m}\sum_{i=1}^{m}\|V_i\|^2_{\psi_\alpha}} \wedge \frac{c_2\eta^\alpha m^\alpha\|h-\widehat{h}^*\|^{2\alpha}_{L_2(P)}}{\max_{1\leq i\leq m}\|V_i\|^\alpha_{\psi_\alpha}}\right)\right\},$$

where

$$V_i = \mathop{\mathbb{E}}_{(X_i,Y_i)\sim D_i}\left[(h-\widehat{h}^*)^2(X_i)\right] - \mathbb{E}_{D_i\sim P}\mathbb{E}_{(X_i,Y_i)\sim D_i}\left[(h-\widehat{h}^*)^2(X_i)\right].$$

We denote $C_1, C_2$ by

$$C_1 = \frac{\frac{1}{m}\sum_{i=1}^{m}\|V_i\|^2_{\psi_\alpha}}{c_1}, \qquad C_2 = \frac{\max_{1\leq i\leq m}\|V_i\|^\alpha_{\psi_\alpha}}{c_2}.$$

Then we have

$$\mathbb{P}\left(\left|\frac{1}{m}\sum_{i=1}^{m}\mathop{\mathbb{E}}_{(X_i,Y_i)\sim D_i}\left[(h-\widehat{h}^*)^2(X_i)\right] - \mathbb{E}_{D_i\sim P}\mathbb{E}_{(X_i,Y_i)\sim D_i}\left[(h-\widehat{h}^*)^2(X_i)\right]\right| \geq \eta\|h-\widehat{h}^*\|^2_{L_2(P)}\right)$$

$$\leq 2\exp\left\{-\left(\frac{\eta^2 m\|h-\widehat{h}^*\|^4_{L_2(P)}}{C_1} \wedge \frac{\eta^\alpha m^\alpha\|h-\widehat{h}^*\|^{2\alpha}_{L_2(P)}}{C_2}\right)\right\}.$$

To make sure that the probability tends to zero as $m$ increase, we could chose $\|h - \widehat{h}^*\|_{L_2(P)} \geq m^{-\frac{1}{4}+\iota}$, where $0 < \iota < \frac{1}{4}$.

By following the similar steps of proof of Theorem 5, it can be proved that with probability at least

$$1 - \exp\left\{-\left(\frac{c_1\eta^2 m^{4\iota}}{\frac{1}{m}\sum_{i=1}^{m}\|V_i\|^2_{\psi_\alpha}} \wedge \frac{c_2\eta^\alpha m^{\frac{\alpha(1+4\iota)}{2}}}{\max_{1\leq i\leq m}\|V_i\|^\alpha_{\psi_\alpha}}\right)\right\},$$

one has

$$\|\widehat{h} - \widehat{h}^*\|^2_{L_2(P)} - \|\widehat{h} - \widehat{h}^*\|^2_{L_2(D)} \leq \eta\|\widehat{h} - \widehat{h}^*\|^2_{L_2(P)}.$$

Thus,

$$\|\widehat{h} - \widehat{h}^*\|^2_{L_2(P)} \leq \frac{1}{1-\eta}\|\widehat{h} - \widehat{h}^*\|^2_{L_2(D)}.$$

Moreover,

$$\|\widehat{h} - h^*\|^2_{L_2(P)} \leq \frac{2}{1-\eta}\|\widehat{h} - \widehat{h}^*\|^2_{L_2(D)} + 2\|\widehat{h}^* - h^*\|^2_{L_2(P)}.$$

If $\xi_i = h^*(X_i) - Y_i$ is independent of $X$, then $\mathbb{E}[\xi_i(h-h^*)(X_i)] = 0$.

$$\mathcal{L}_P(h) - \mathcal{L}_P(h^*) = \frac{1}{m} \sum_{i=1}^{m} \mathbb{E}\left[ (h(X_i) - Y_i)^2 - (\widehat{h}^*(X_i) - Y_i)^2 \right] \tag{45}$$

$$= \frac{1}{m} \sum_{i=1}^{m} \mathbb{E}\left[ (h - \widehat{h}^*)^2(X_i) \right] + \frac{2}{m} \sum_{i=1}^{m} \mathbb{E}\left[ \xi_i (h - \widehat{h}^*)(X_i) \right] \tag{46}$$

$$= \frac{1}{m} \sum_{i=1}^{m} \mathbb{E}\left[ (h - \widehat{h}^*)^2(X_i) \right]. \tag{47}$$

Thus,

$$\mathcal{L}_P(h) - \mathcal{L}_P(h^*) = \frac{1}{m} \sum_{i=1}^{m} \mathbb{E}[(h - \widehat{h}^*)^2(X_i)] \tag{48}$$

$$= \|\widehat{h} - h^*\|_{L_2(P)}^2 \tag{49}$$

$$\leq \frac{2}{1-\eta} \|\widehat{h} - \widehat{h}^*\|_{L_2(D)}^2 + 2\|\widehat{h}^* - h^*\|_{L_2(P)}^2. \tag{50}$$

Combining inequality (50) with Theorem 4 and Theorem 5, we complete the proof.

$\square$

## F  EXPERIMENTAL RESULTS

### F.1  CONVOLUTIONAL NEURAL NETWORKS FOR EMNIST TASK

To check the validity of our theory for over-parameterized models, we train convolutional neural network for EMINIST task (Cohen et al., 2017). In particular, we use FedAdam (Reddi et al., 2020) with server momentum $= 0.9$. The participating and unparticipating clients are split based on the methods proposed in (Yuan et al., 2021). We set the unparticipating rate as 0.2. Our experiments are based on Tensorflow Federated (TFF) (Alex & Krzys, 2019).

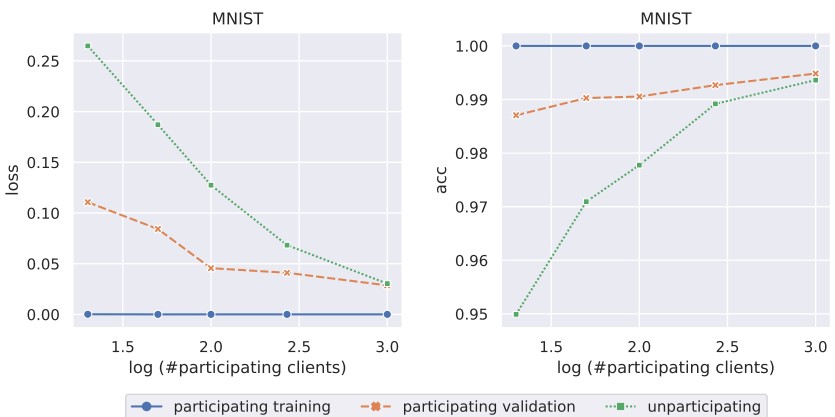

Figure 2: Generalization error versus the number of participating clients.

In Figure 2, we show how generalization errors for unparticipating clients and participating clients convergence when we increase the number of participating clients $m$. Here we fix $n = 100$. It can be seen that the convergence rate of participating error is slower than that of unparticipating error. This phenomenon matches our theretical results in Theorem 1 well. In our results, the convergence rate of unparticipating error is of order $\mathcal{O}(\frac{1}{\sqrt{mn}} + \frac{1}{\sqrt{m}})$. Compared to the convergence rate of participating error, which is of order $\mathcal{O}(\frac{1}{\sqrt{mn}})$, unparticipating error is expected to have faster convergence rate.

In Figure 3, we show how generalization errors for unparticipating clients and unparticipating clients convergence when we increase $n$. Here we fix $m = 20$. It can be seen that the convergence rates

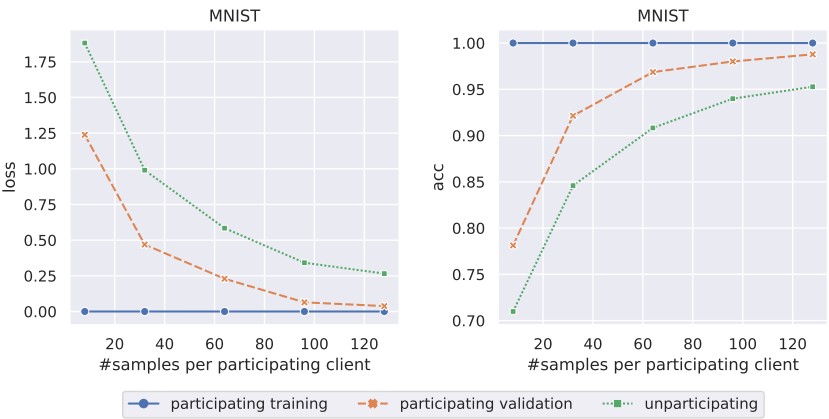

Figure 3: Generalization error versus the number of samples per participating client.

of participating error and unparticipating error are similar. This phenomenon matches our theretical results in theorem 1 as well. When $m$ is fixed, the order of $\mathcal{O}(\frac{1}{\sqrt{mn}} + \frac{1}{\sqrt{m}})$ is same as the order of $\mathcal{O}(\frac{1}{\sqrt{mn}})$.

### F.2 FEDERATED LINEAR REGRESSION WITH SYNTHETIC DATA

In Figure 4, we show the numerical experiments results based on the linear regression model. We first describe our linear regression setting as follows. For client $i \in [m]$, the dataset is given as $S_i = \{X_i^j, Y_i^j\}$ with $n$ samples. Let $d$ be the dimensionality of the input space. We focus on the setting:

$$Y_i^j \mid X_i^j, \theta_i \sim \mathcal{N}\left(X_i^{j\top}\theta_i, \sigma_i^2\right), \quad \forall j = 1, \ldots, n,$$

where $\sigma_k^2$ is a noise parameter. In our experiments, we set $\sigma_i = 0.05$. For excess risk, we fix $n = 20$. For semi-excess risk we fix $m = 40$.

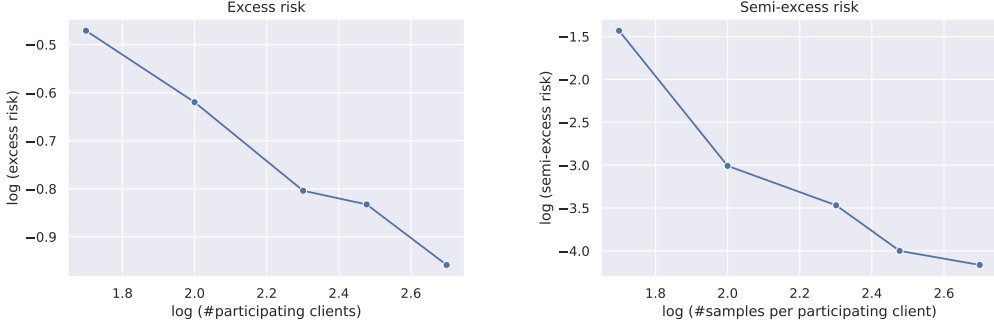

Figure 4: Excess risk versus $m$ and Semi-excess risk versus $n$.

