# OpenReview forum: "Generalization Bounds for Federated Learning: Fast Rates, Unparticipating Clients and Unbounded Losses"
_ICLR.cc/2023/Conference — ICLR 2023 poster_

### Official Review · Reviewer_8Tvp · 2022-10-21

**Confidence:** 4
**Correctness:** 2
**Technical Novelty And Significance:** 2
**Empirical Novelty And Significance:** Not applicable
**Recommendation:** 3

**Clarity, Quality, Novelty And Reproducibility:**

The paper is mostly well-written but there were issues in the proofs, which should be properly addressed to improve clarity of the paper.
I think the paper's quality can be improved. The techniques based on Rademacher complexity- and local Rademacher complexity to develop generalisation bounds are pretty standard. The reproducibility is not applicable as there are not experimental results.



**Strength And Weaknesses:**

Strength: Studying federated learning in terms of learning is certainly an important direction, which has not been addressed in the current literature. I think the paper is well-written and the authors  did a good job in terms of going beyond convergence analyses and using known Rademacher complexity-based generalization bounds to obtain (semi) excess risk bounds.

-----

Weakness: Although the authors emphasized much about the relevanece of their bounds to FL in practical settings, in particular cross-device FL with a large pool of unparticipating clients, there are major concerns regarding the considered model, the correctness of results, and conditions under which main theorems hold, which will be elaborated in the following. Also, it is unfortunate that the paper lacks convincing experimental results to validate that the developed bounds indeed explain whether "unparticipating clients benefit from the model trained by participating clients?"

-----
Detailed comments:

 "We argue that this assumption is reasonable in practice. For instance, in cross-device federated learning, the number of total clients is generally large and it is natural to assume that there exists a meta-distribution"

This is not really convincing. What if some clients have data distributions that are quite different from others. Also what if the distribution of clients changes over time.  Another major issue is  how should such meta-distribution be modelled? Will you use parametric or non-parameteric models? Given that the authors emphasize much about practical scenarios and settings where several clients may not even participate in training, how can one model/estimate the meta distribution? Imagine that a practitioner does not have even a sample drawn from distributions for the major of clients (unparticipating clients).

-----

The symmetrization step in the proof of Lemma 1 should be expanded. BTW, the first line after "By symmetrization, we have" cannot be an equality.

More importantly, the definitions for ${\cal R}_{mn}({\cal F})$ in pages 14 and 15 are different, which makes the proof hard to read. The same issue exists in Lemma 2. In addition, $Z_1$,..., $Z_m$ in the proof of Lemma 2 are undefined. So I am not confident about the correctness of the results.

-----

Theorem 2 requires $n=\Omega(d)$, which does not hold for typical overparameterized models whose number of parameters far exceed the number of training data.  This is even worse in Theorem 4 where $m$, which is claimed to be relatively small is required to be larger than $d$.

-----

The techniques used to prove Theorem 2 are pretty standard based on (Bartlett et al., 2002, Sridharan et al., 2009, Yousefi et a.., 2018). Is there any technical difficulty in the problem of bounding semi-excess risk compared to previous work?


Karthik  Sridharan, Shai Shalev-Shwartz, and Nathan Srebro. "Fast rates for regularized objectives." Advances in neural information processing systems (NeurIPS),  2008.


Peter L. Bartlett, Olivier Bousquet, and Shahar Mendelson. "Localized rademacher complexities." International Conference on Computational Learning Theory (COLT), 2002.

Niloofar Yousefi, Yunwen Lei, Marius Kloft, Mansooreh Mollaghasemi, and Georgios C Anagnostopoulos. Local rademacher complexity-based learning guarantees for multi-task learning. The Journal of Machine Learning Research, 19(1):1385–1431, 2018.



-----

Why not considering the excess risk instead of semi-excess risk for participating clients? It seems that the bounds will not be better than those for unparticipating clients.

-----

The bound in Theorem 1 will be vacuous when using overparameterized models and a small $m$. The authors emphasized that $m$ is a moderate value in their settings.  which does not lead to a meaningful generalization results.

-----

Assumption 2: $X_i^1$ in Assumption 2 is drawn from $D_i$, which is not $D$. Assumption 2 (a) does not seem to be compatible with Definition 1.

-----


It will be nice if the authors elaborate on what specific problems in ML lead to sub-Weibull loss  distributions with $\alpha\neq 1,2$ for $h$ as in Theorems 5 and 6. Also it is a bit confusing since in the introduction the authors claim that they consider sub-Weibull loss functions while they indeed consider sub-Weibull classifiers in Theorem 5,6.

-----

The definition of  VC major class, which is the main building block of Theorem 1 should be provided in the main paper.

**Summary Of The Paper:**

The authors consider  data distributions of participating and unparticipating clients are drawn
from a meta-distribution and use classic Rademacher complexity- and local Rademacher complexity-based generalisation bounds to develop excess risk bounds for unparticipating clients and fast semi-excess risk bounds for participating clients.


**Summary Of The Review:**

The authors  did a good job in terms of going beyond convergence analyses and developing (semi) excess risk bounds for participating and unparticipating clients. However, there are major concerns regarding the considered model, the correctness of results, and conditions under which main theorems hold.

---

> ### Author Response · Authors · 2022-11-19
> **Responses to Reviewer 8Tvp (1/3)**
>
> Dear Reviewer 8Tvp, we sincerely appreciate your invaluable and constructive comments. We respond to your concerns one by one.
>
> **R: It is unfortunate that the paper lacks convincing experimental results to validate that the developed bounds indeed explain whether "unparticipating clients benefit from the model trained by participating clients?**
>
> **A:** We add experimental results on the MNIST dataset and synthetic data (see appendix E). The experimental details can be found in the appendix. Here we provide a discussion of our results.
> - The experimental results show that the generalization error of unparticipating clients decreases along with the increase in the number of participating clients. This phenomenon matches our theory well.
> It can be seen that
> - There indeed exists a gap between participating error and unparticipating error. Moreover, the convergence rate of participating error is slower than that of unparticipating error when we only increase the number of participating clients. This phenomenon matches our theoretical results in theorem 1 well. Let's explain this in more detail. In our results, the convergence rate of unparticipating error is of order $\mathcal{O}(\frac{1}{\sqrt{mn}} + \frac{1}{\sqrt{m}} )$. Compared to the convergence rate of participating error, which is of order $\mathcal{O}(\frac{1}{\sqrt{mn}})$, unparticipating error is expected to have a faster convergence rate.
>
> **R: What if some clients have data distributions that are quite different from others? Also what if the distribution of clients changes over time.**
>
> **A:** We'd like to highlight that assumption of the meta-distribution is more general than other assumptions on the clients' heterogeneity. Even if the distributions differ greatly, we can still consider them to come from a meta-distribution. If the difference between the distributions is relatively large, this corresponds to the heavy-tailed phenomenon of the meta-distribution.
>
> **R: What if the distribution of clients changes over time.**
>
> **A:** This is a great question. Our theoretical results are derived under the empirical risk minimization (ERM) setting. If the participating clients change over time, then the minimizers $\widehat{h} = \underset{h \in \mathcal{H}}{\operatorname{argmin}} \mathcal{L}_{S}({h})$
>
> and $\widehat{h}^* = \underset{h \in \mathcal{H}}{\operatorname{argmin}} \mathcal{L}_{D}({h})$ will change over time.  This brings more challenges to finding the optimal minimizer $\widehat{h}$, which is an optimization problem. Therefore, our results can still be applied at every single time. We believe our results can be extended to the time-varying scenario by modeling the mixture distributions $D$ at different times as dependent random variables(i.e., Markov chain). However, this is beyond the scope of this paper and we leave this problem for further work.
>
> **R: Another major issue is how should such meta-distribution be modelled? Imagine that a practitioner does not have even a sample drawn from distributions for the major of clients (unparticipating clients).**
>
> **A:** This is a very good question. A natural way to estimate the meta-distribution is to choose some clients (distribution) that never participated in the training for validation. This works when the local distributions are similar. However, it is difficult or even impossible to model the meta-distribution when customers are very different. According to our experimental results on MNIST(see Appendix F), this method works well.
>
> **R: Will you use parametric or non-parameteric models?**
>
> **A:** This is a good question. It depends on the properties of practical problems. Our experimental results show that the neural network works well on the MNIST dataset. However, in applications such as federated healthcare, a spare model like $\ell_1$ regression may be better.
>
> **R: The symmetrization step in the proof of Lemma 1 should be expanded. BTW, the first line after "By symmetrization, we have" cannot be an equality.**
>
> **A:** Thanks for pointing this out. We have developed the symmetrization steps in Lemma 1 and Lemma 2 in detail, and further provided annotations. The first line after "By symmetrizing, we have" in Lemma 1 should be an inequality because Jensen's inequality is used. In addition, we have carefully checked the proof in the attachment to prevent similar typos. Thanks for pointing out the mismatch in definitions on pages 14 and 15. The definition on page 14 is the expected version of the definition (empirical Rademacher complexity) on page 15. These two terms appear in the proof at different steps. The final results only involve empirical Rademacher complexity. We have fixed this and only use empirical Rademacher complexity in the revised version. We believe the revised version is clearer and easier to read. Thank you very much for your careful review, which greatly helps us improve the quality of the draft.

---

> > ### Author Response · Authors · 2022-11-19
> > **Responses to Reviewer 8Tvp (2/3)**
> >
> > **R: Theorem 2 requires $n=\Omega(d)$, which does not hold for typical overparameterized models whose number of parameters far exceed the number of training data. This is even worse in Theorem 4 where $m$, which is claimed to be relatively small is required to be larger than $d$.**
> >
> > **A:** The requirement here is used to get fast rates. Let's explain this in more detail. Compared to the total number of clients $M$, the number of participating clients $m$ is relatively small. However, $m$ can still be large compared to $d$. For example, for the kernel method, $d$ represents the rank of the kernel matrix. Also, we added experiments based on neural networks. The experimental results show that in the cases where neural networks are used, the convergence trend of generalization error matches with the prediction of Theorem 1.
> >
> > **R: The bound in Theorem 1 will be vacuous when using overparameterized models and a small m. The authors emphasized that m is a moderate value in their settings. which does not lead to a meaningful generalization results.**
> >
> > **A:** This is a good question. Before going further, we would like to emphasize that the main contribution of this paper is a systematic theoretical analysis of the generalization of federated learning under the framework of the two-layer framework. We believe that the theoretical results of this paper will bring useful inspiration to the federated learning community. The reason why the traditional complexity method cannot give non-vacuous bounds is that the complexity method measures the size of the entire hypothesis space and is independent of algorithms. However, for over-parameterized models like neural networks, it has been shown that algorithms like SGD will not search the entire hypothesis space. In other words, the hypothesis space of a neural network is compressible, and the compressed hypothesis space is potentially small.
> > We notice that several recent works combined the complexity method with other tools to give non-vacuous bounds. For example, literature [1] and [2]  combined Rademacher complexity and the trajectory of SGD to give a non-vacuous bound. We believe that our results can be extended to a non-vacuous version based on similar techniques. Also, we added experiments based on neural networks. The experimental results show that in the cases where neural networks are used, the convergence trend of generalization error matches with the prediction of Theorem 1.
> >
> > **R: The techniques used to prove Theorem 2 are pretty standard based on (Bartlett et al., 2002, Sridharan et al., 2009, Yousefi et a.., 2018). Is there any technical difficulty in the problem of bounding semi-excess risk compared to previous work?**
> >
> > **A:** Please refer to the public response to all Reviewers.
> >
> > **R: Why not considering the excess risk instead of semi-excess risk for participating clients? It seems that the bounds will not be better than those for unparticipating clients.**
> >
> > Please correct us if we misunderstood the question. According to the definition, semi-excess risk represents the performance of the trained model with respect to seen distributions (participating clients). The excess risk represents the performance of the trained model with respect to unseen distributions (unparticipating clients).
> >
> > **R: Assumption 2: Xi1 in Assumption 2 is drawn from Di, which is not D. Assumption 2 (a) does not seem to be compatible with Definition 1.**
> >
> > **A:** We recall that distribution D is defined as a mixture of the participating client distributions $\{D_i\}_{i=1}^m$. Therefore, the expression in Assumption 2(a) is the equivalent form of the standard definition 1, which can be seen by expanding the mixture distribution$D$. To avoid this confusion, we have switched to the standard form in the updated version.
> >
> > **R: It will be nice if the authors elaborate on what specific problems in ML lead to sub-Weibull loss distributions with $\alpha \neq 1,2$ for h as in Theorems 5 and 6.**
> >
> > **A:** Taking the application of federated learning in medical health as an example, the cost and hospitalization time of patients with serious illnesses are extreme values. Moreover, the probability of appearing these extrema values is higher than the gaussian distribution and may be heavy-tailed.
> >
> > **R: Also it is a bit confusing since in the introduction the authors claim that they consider sub-Weibull loss functions while they indeed consider sub-Weibull classifiers in Theorem 5,6.**
> >
> > **A:** Both the unbounded property of the classifier $h(X)$ and the label $Y$ will lead to unbounded loss. In the sub-exponential section, we consider the case where both the classifier $h(X)$ and the label $Y$ are sub-exponential. In the small-ball-based analysis, the unbounded property of the label is hidden in the noise we defined. Then We carefully made a sub-Weibull assumption on the product of noise and classifier.

---

> > > ### Author Response · Authors · 2022-11-19
> > > **Responses to Reviewer 8Tvp (3/3)**
> > >
> > > **R: The definition of VC major class, which is the main building block of Theorem 1 should be provided in the main paper.**
> > >
> > > **A:** Thank you for pointing out this, we have moved the definition of VC major class from the appendix to the main paper in the updated version.
> > >
> > > [1] Park, Sejun, Umut Şimşekli, and Murat A. Erdogdu. "Generalization Bounds for Stochastic Gradient Descent via Localized $\varepsilon $-Covers." arXiv preprint arXiv:2209.08951  (2022).
> > >
> > > [2] Wang, Mingze, and Chao Ma. "Generalization Error Bounds for Deep Neural Networks Trained by SGD." arXiv preprint arXiv:2206.03299  (2022).

---

> > > > ### Author Response · Authors · 2022-12-06
> > > > **Happy to provide further clarification**
> > > >
> > > > Thank you for the comments and questions. We hope our response clarified your initial concerns/questions. We would be happy to provide further clarifications where necessary.

---

> > > > > ### Comment · Reviewer_8Tvp · 2022-12-10
> > > > > **After Rebuttal**
> > > > >
> > > > > I would like to thank the authors for their detailed response.
> > > > >
> > > > > Considering the issues in the proofs that I could check while reviewing the paper, I am not confident that all proofs are correct.
> > > > >
> > > > > In addition, I think the experimental results in Appendix F are not sufficient to validate the theory. The authors use FedAdam with server momentum on MNIST, which is not clear how it is related to the theoretical results. Momentum itself may impact generalization error, which is not analyzed in this paper. Also the experiments have nothing to do with the meta distribution. The authors wrote "A natural way to estimate the meta-distribution is to choose some clients (distribution) that never participated in the training for validation." This is very vague. Who will sample from those clients? Who does have access to such samples when those clients are not participating?
> > > > >
> > > > > I read all comments and responses carefully. Unfortunately, I think this paper is far from being ready and recommend rejection.

---

> > > > > > ### Author Response · Authors · 2022-12-11
> > > > > > **Response to Reviewer 8Tvp | Clarify Major Misunderstandings**
> > > > > >
> > > > > > Dear reviewer 8Tvp,
> > > > > >
> > > > > > In the first phase of rebuttal discussion, we answered your initial questions one by one. For your concerns on the standard "symmetrization trick" in the proof, we have provided expansion steps in the revision paper (see Page 16).  If you have further concerns about the proof, we hope you point out the specific steps. We are happy to provide further explanations. Without focusing on specific steps, we have no idea how we're going to address your concern. More importantly, expressing further concerns about our proofs without responding to our previous detailed explanations and pointing out specific steps is unacceptable.
> > > > > >
> > > > > > **R: The experimental results are not sufficient to validate the theory since Momentum is not analyzed in this paper.**
> > > > > >
> > > > > > **A:** This claim is not correct. All of our theoretical results are derived under Empirical Risk Minimization (ERM) setting and are independent from specific learning algorithms. We clarified this in the initial submitted version and emphasized again in our responses. Specifically, all of our generalization bounds reflect the performance of the ERM solution $\widehat{h} = \underset{h \in \mathcal{H}}{\operatorname{argmin}} \mathcal{L}_{S}({h}),
> > > > > > $ and have nothing to do with specific optimizer. We choose FedAdam since the model trained by FedAdam is optimal and closest to ERM solution $\widehat{h}.$
> > > > > >
> > > > > > **R: The experiments have nothing to do with the meta distribution.**
> > > > > >
> > > > > > **A:** This claim is not correct and is caused by the misunderstanding of the definition of generallization error for unparticipationg clients. The generalization error for unparticipating clients is defined as $ \mathcal{L}_P(\widehat{h})=
> > > > > > \underset{D_i \sim P}{\operatorname{ \mathbb{E}}}   \underset{Z \sim D_i}{\operatorname{ \mathbb{E}}} [\ell(\widehat{h}, Z)]] ,
> > > > > > $ which is associated with meta distribution $P.$
> > > > > > For the experimental results on MNIST (see F.1), the green line corresponds to the $\mathcal{L}_P(\widehat{h}).$
> > > > > > For the numerical results, the blue line corresponds to excess risk $\mathcal{L}_P(\widehat{h}) - \mathcal{L}_P(h^*),
> > > > > > $ which is associated with meta distribution.
> > > > > >
> > > > > > **R: The estimation of the meta-distribution.**
> > > > > >
> > > > > > **A:** In our two-level framework, if we want to estimate the meta distribution, we have to keep some testing clients as "unparticipating". These testing clients will not participate in the training process, but will evaluate the trained model with their local data. This is like using testing data to estimate generalization error, which is common in the classic Expected Risk Minimization framework.

---

> > > > ### Author Response · Authors · 2022-12-10
> > > > **Please let us know if you have more questions / concerns.**
> > > >
> > > > Dear Reviewer 8Tvp,
> > > >
> > > > Thank you once again for your detailed review and valuable suggestions. We hope that our response satisfactorily addressed your questions / concerns. Given that the discussion deadline is coming, please let us know if you have more questions / concerns.

---

### Official Review · Reviewer_zSzo · 2022-10-24

**Confidence:** 3
**Correctness:** 4
**Technical Novelty And Significance:** 3
**Empirical Novelty And Significance:** Not applicable
**Recommendation:** 6

**Clarity, Quality, Novelty And Reproducibility:**

Clarity: The paper may be mathematically heavy for most people working on federated learning nowadays. Further discussion on new insights obtained from the theoretical results would be very useful, especially those that are not known in the field of federated learning as of today.

Quality: The paper appears to be technically solid, although it is beyond my capacity to check all the details.

Novelty: The work seems sufficiently novel.

Reproducibility: This is a theoretical paper with no experiments, so reproducibility is out of scope.

**Strength And Weaknesses:**

Strength:
- The analysis of generalization in federated learning is interesting and not often seen in the literature.
- This paper is a solid piece of theoretical work which may inspire future work in this direction, although checking all the details is beyond my capacity.
- The consideration of unbounded and heavy-tailed losses is an interesting theoretical contribution.

Weakness:
- It is not clear how clients are selected to participate or not participate. My conjecture is that the participating clients are selected independently from the underlying data distribution. If so, this setup does not fully consider some practical scenarios, such as where certain populations may generally have low-end phones or poor connections so that it is difficult for them to participate in federated learning. It would be more realistic if participation can be modeled as correlated to the underlying dataset, but I also understand the difficulty here.
- The results depend on VC dimension, which is a way of quantifying the complexity of classical models but it generally does not capture the empirical behaviors of deep neural networks. Therefore, the theoretical bounds may not reflect what will really happen when training deep neural networks.
- In Section 3, the assumptions are much stronger than what is usually used in the analysis of SGD-based federated learning algorithms. So the paper should not emphasize too much on obtaining 'fast' rates, as it is not fair to compare the rates obtained under different assumptions.
- The paper considers clients that either participate or do not participate. For the participating clients, it assumes that the minimum empirical risk can be obtained by $\hat{h}$. In reality, some clients may participate in more rounds and other clients may participate in less rounds, which may result in an inaccurate $\hat{h}$ that does not give the true minimum of empirical risk. Such varying degrees of participation has not been considered in this paper.
- The paper is mathematically heavy and lacks discussions on new findings and insights that can guide the understanding and design of practical federated learning systems. More discussions on what the bounds tell us intuitively (in plain words) about how federated learning algorithms will behave would be very useful.


**Summary Of The Paper:**

The paper studies generalization bounds in federated learning with unparticipating clients. A two-level distribution framework is proposed, where the data distribution of each client is sampled according to a meta distribution, then the data samples at each client is sampled from the client's data distribution. Afterwards, upper bounds on the generalization error are shown for two classes of loss functions.

**Summary Of The Review:**

Nice theoretical work, but it would be helpful if connections to practical federated learning systems and new insights can be discussed.

---

> ### Author Response · Authors · 2022-11-19
> **Responses to Reviewer zSzo**
>
> Dear Reviewer zSzo, we sincerely appreciate your invaluable and constructive comments. We respond to your concerns one by one.
>
> **R: It is not clear how clients are selected to participate or not participate. It would be more realistic if participation can be modeled as correlated to the underlying dataset, but I also understand the difficulty here**
>
> **A:** This is a great question. Your conjecture is right. We assume that participating clients are selected independently from meta-distribution. We agree that modeling the relationship between participation and the underlying dataset is important. To me, this sounds like saying that the meta distribution in the training phase is different from the test phase. This reminds me of the paper "Agnostic Federated Learning" by Mohri et al. which addresses a similar question. They argue that federated models can be biased towards different clients when we ignore the dependence between participation and the underlying dataset. However, they only focus on the mixture of seen clients during training. We believe that our results in two-level framework can be extended to their "Agnostic“ setting, which reflects the relationship between participation and the underlying dataset.
>
> **R: In Section 3, the assumptions are much stronger than what is usually used in the analysis of SGD-based federated learning algorithms. So the paper should not emphasize too much on obtaining 'fast' rates, as it is not fair to compare the rates obtained under different assumptions.**
>
> **A:** Thank you for raising this question. We'd like to argue that the assumptions in Section 3 are mild instead of strong. We are happy to discuss and clarify this in the following. First, Bernstein condition is usually implied by the convexity of the hypothesis class and the convexity of the loss function, just as in the examples we provide in Appendix C. Moreover, Strong convexity and Lipschitz condition imply Bernstein condition immediately. Therefore, Bernstein condition is more general than Strong convexity. We have added a discussion on the connection to practice in the appendix. Thanks again for raising this question, which helps us improve the quality of the paper.
>
> **The paper considers clients that either participate or do not participate. For the participating clients, it assumes that the minimum empirical risk can be obtained by h^. In reality, some clients may participate in more rounds and other clients may participate in less rounds, which may result in an inaccurate h^ that does not give the true minimum of empirical risk. Such varying degrees of participation has not been considered in this paper.**
>
> **A:** Good question. Our theoretical results are derived under the empirical risk minimization (ERM) setting. If the participating clients change over time, then the minimizers $\widehat{h} = \underset{h \in \mathcal{H}}{\operatorname{argmin}}\mathcal{L}_{S}({h})$
>
> and $\widehat{h}^* =  \underset{h \in \mathcal{H}}{\operatorname{argmin}} \mathcal{L}_{D}({h})$ will change over time. This brings more challenges to finding the optimal minimizer $\widehat{h}$, which is an optimization problem. Therefore, our results can still be applied at every single time. Moreover, we believe our results can be extended to the time-varying scenario by modeling the mixture distributions $D$ at different times as dependent random variables(i.e., Markov chain).
>
> **R: The paper is mathematically heavy and lacks discussions on new findings and insights that can guide the understanding and design of practical federated learning systems. More discussions on what the bounds tell us intuitively (in plain words) about how federated learning algorithms will behave would be very useful.**
>
> **A:** Please refer to the public response to all Reviewers.
>
> [1] Mohri, Mehryar, Gary Sivek, and Ananda Theertha Suresh. "Agnostic federated learning." International Conference on Machine Learning. PMLR, 2019.

---

> > ### Comment · Reviewer_zSzo · 2022-12-07
> > **Thank you**
> >
> > Thank you for the explanation, which has helped me understand the paper better. Since this is a growing field, I believe the contributions in this paper can be useful for advancing our understanding of generalization in federated learning, so the paper can be accepted if there is room. Meanwhile, some parts of the theory are quite different from the practical setting of federated learning systems (e.g., clients selected independently from meta-distribution, etc.), so I keep my original score of 6.
> >
> > Regarding novelty of the proof as pointed out by some other reviewers, it would be nice if the authors could point out some specific steps in the analysis that differ from existing analyses.

---

> > > ### Author Response · Authors · 2022-12-09
> > > **Thanks for the reply**
> > >
> > > Dear Reviewer zSzo,
> > >
> > > We sincerely thank you for your reply and the recognition of this paper. We feel very encouraged.
> > >
> > > Thank you again for your suggestion. We will further carefully revise this paper and point out the specific steps that differ from existing analyses.

---

### Official Review · Reviewer_yZwY · 2022-11-03

**Confidence:** 2
**Correctness:** 3
**Technical Novelty And Significance:** 3
**Empirical Novelty And Significance:** 3
**Recommendation:** 6

**Clarity, Quality, Novelty And Reproducibility:**

**Clarity**

The following parts of this paper are not clear to me.
- Theorems 5 and 6 look weird to me. If $\iota \in (0,\frac{1}{4})$ (say $\iota = \frac{1}{8})$, the second part of the rate is of $\left(\frac{1}{m}\right)^{-\frac{1}{4} + \iota}$, which diverges as $m\rightarrow \infty$. It does not recover the $\frac{1}{m^{\frac{1}{4}}}$ rate shown in Table 1.
- Assumption 3 states that $\mathcal{H}$ is a family of bounded functions while the first sentence below that assumption claims that ``Assumption 3 is a mild assumption is the function classes are bounded.'' Could you please explain this?
- Both column 2 and column 3 in Table 1 are some assumptions of the loss function (e.g. "bounded" in column 2 and "Smoothness, SC" in column 3). Why list them separately?
- In several places in the appendix, I see both $(X,Y)\sim D_i$ and $X\sim D_i$.

**Quality**

This paper is well-motivated and the research goal is easy to understand. However, there are a couple of typos and minor mistakes:
- The last terms in Lemma 1 and Lemma 2 should be $3M\sqrt{\frac{\ln(1/\delta)}{2mn}}$ and $3M\sqrt{\frac{\ln(1/\delta)}{2m}}$?
- "state of art guarantees"
- "Convering number"
- "Uniformly entropy number"
- "produc"
- Several missing references to equations in the appendix

**Reproducibility**
N/A


**Strength And Weaknesses:**


**Strength**
- The problem studied in this paper is interesting and challenging.
- Proof techniques seem to be new in the context of the generalization property of FL. Some open problems seem to be answered (e.g. unbounded loss, high-probability bounds).
- The assumptions come with examples to back them up (e.g. Appendix B.1 for Assumption 2). Since this paper uses different assumptions compared to previous work, the examples make me better understand the contributions.

**Weakness**
- Some statements and claims in this paper are confusing to me (see the "Clarity" section below).
- No empirical verifications (although this is not a big issue for a theory paper).









**Summary Of The Paper:**

After the rebuttal, my main concerns have been resolved.
-------------
This paper studies the generalization bounds for federated learning (FL) with heterogeneous local data. Similar to recent works (e.g. Yuan et al. 2021), a two-level distribution framework is considered, i.e. the local distribution is drawn from the meta-distribution, and the local data is drawn from the local distribution. Compared to previous works on the generalization of FL, this work can decouple the generalization bounds for participating and nonparticipating clients while showing fast rates under different sets of assumptions.

**Summary Of The Review:**

In summary, this paper presents some interesting results on the generalization properties of FL. However, some technical claims seem to be problematic to me (especially the first bullet in the "Clarity" section). Disclaimer: This is probably due to my limited time for reviewing (It was assigned to me too late) and my lack of expertise on this topic. I am more than happy to raise my score if concerns are resolved.

---

> ### Author Response · Authors · 2022-11-19
> **Responses to Reviewer yZwY (1/2)**
>
> Dear Reviewer yZwY, we sincerely appreciate your invaluable and constructive comments. We respond to your concerns one by one.
>
> **R: No empirical verifications (although this is not a big issue for a theory paper).**
>
> **A:** We add experimental results on the MNIST dataset and synthetic data. We use CNN for MINIST task(see Figure 2 and Figure 3), which allows us to check the validity of our theory for over-parameterized models. Moreover, we add numerical experiments based on the linear regression model(see Figure 4). In details:
>
> - In Figure 2, we show how generalization errors for unparticipating clients and participating clients converge when we increase the number of participating clients $m$. It can be seen that the convergence rate of participating error is slower than that of unparticipating error. This phenomenon matches our theoretical results in theorem 1 well. In our results, the convergence rate of unparticipating error is of order $\mathcal{O}(\frac{1}{\sqrt{mn}} + \frac{1}{\sqrt{m}} )$. Compared to the convergence rate of participating error, which is of order $\mathcal{O}(\frac{1}{\sqrt{mn}})$, unparticipating error is expected to have a faster convergence rate.
>
> - In Figure 3, we show how generalization errors for unparticipating clients and unparticipating clients converge when we increase $n$. It can be seen that the convergence rates of participating error and unparticipating error are similar. This phenomenon matches our theoretical results in theorem 1 as well. When $m$ is fixed, the order of $\mathcal{O}(\frac{1}{\sqrt{mn}} + \frac{1}{\sqrt{m}} )$ is same as the order of $\mathcal{O}(\frac{1}{\sqrt{mn}})$.
>
> - In Figure 2, it can be seen that the participating gap decreases along with the increase of $m$. In Figure 3, it can be seen that the participating gap always exists even we increase $n$. This phenomenon matches our theoretical results as well.
>
> - In Figure 4, for excess risk, we show how it convergence when we increase $m$. For semi-excess risk, we show how it convergence when we increase $n$.
>
> **R: Theorems 5 and 6 look weird to me. If $\iota \in\left(0, \frac{1}{4}\right)$ (say $\iota=\frac{1}{8}$), the second part of the rate is of $\left(\frac{1}{m}\right)^{-\frac{1}{4}+\iota}$, which diverges as $m \rightarrow \infty$. It does not recover the $\frac{1}{m^{\frac{1}{4}}}$ rate shown in Table 1.**
>
> **A:** Thank you for pointing this out. In the corresponding Section of the appendix, it can be seen that our results are $(mn)^{-\frac{1}{4} + \iota }$ and $m^{-\frac{1}{4} + \iota }$. The expression of forms $(\frac{1}{mn})^{-\frac{1}{4} + \iota }$ and $(\frac{1}{m})^{-\frac{1}{4} + \iota }$ are typos and have been fixed in our new version. We are really sorry for the confusion. In addition, we have carefully checked the proof in the attachment to prevent similar typos. We hope that we have addressed your concerns.
>
> **R: Assumption 3 states that H is a family of bounded functions while the first sentence below that assumption claims that ""Assumption 3 is a mild assumption is the function classes are bounded.'' Could you please explain this?**
>
> **A:** Uniform entropy number can be seen as a type of "size" of the hypothesis classes. Let $\mathcal{H}$ be the hypothesis class and $S=\{ X_1, X_2, \cdots, X_n \}$ be the dataset. Entropy number represents the minimal size of quantified version $ \widehat{\mathcal{H}} $ of $\mathcal{H}$. The condition of this quantification is that $ \widehat{\mathcal{H}} $ needs to be able to cover ${\mathcal{H}}$ with given tolerance. The measure of tolerance is defined on the distance between projections of $ \widehat{\mathcal{H}} $ and ${\mathcal{H}}$ on dataset $ S $. To bound this type of tolerance, it is commonly required that the hypothesis(function) in the hypothesis(function) class is bounded. We hope that we have addressed your concerns.

---

> > ### Author Response · Authors · 2022-11-19
> > **Responses to Reviewer yZwY (2/2)**
> >
> > **R: Both column 2 and column 3 in Table 1 are some assumptions of the loss function (e.g. "bounded" in column 2 and "Smoothness, SC" in column 3). Why list them separately?**
> >
> > **A:** This is a good question. First, we list them separately to highlight our results in different unbounded settings(sub-exponential and Sub-Weibull). Second, let's discuss this in more detail. If we make assumptions on losses evaluated at one single data point, these assumptions are orthogonal to the underlying data distribution. For the analysis of unbounded losses, the assumptions are related to the underlying data distribution instead of one data point. Therefore, we list them separately. We hope that we have addressed your concerns.
> >
> > **R: In several places in the appendix, I see both $(X,Y)\sim D_i$ and $X \sim D_i.$**
> >
> > **A:** Thank you for pointing this out. We are sorry for abusing $X \sim D_i$ as the marginal distribution of $X$. We have fixed the abuse cases like this in the revised version. In addition, We carefully checked the proof in the attachment to prevent similar abuse cases.
> >
> > **R: The constants of last terms in Lemma 1 and Lemma 2.**
> >
> > **A:** Thank you for pointing this out. We have fixed this in the revised version.

---

### Official Review · Reviewer_zYTD · 2022-11-03

**Confidence:** 2
**Clarity, Quality, Novelty And Reproducibility:** The paper is well written.
**Correctness:** 4
**Technical Novelty And Significance:** 4
**Empirical Novelty And Significance:** Not applicable
**Recommendation:** 6

**Strength And Weaknesses:**

The paper provides interesting theoretical insights about heterogeneity and the important challenge of partial participation in federated learning (FL).

I am an optimizer, not a statistician. I develop optimization algorithms for FL and I am not familiar with these aspects. With this disclaimer in mind, I have the following questions:
1) Is there a connection between your setting and the literature on personalized FL? Indeed, from the point of view of the nonparticipating clients, the obtained model is not personalized enough. I am thinking for instance of the paper Hanzely et al. "Lower Bounds and Optimal Algorithms for Personalized Federated Learning"
2) I have come across the paper "With a Little Help from My Friend: Server-Aided Federated Learning with Partial Client Participation" of Yang et al. (https://openreview.net/forum?id=xT6d2Ghtkv) which also studies the generalization error of heterogeneous FL with partial participation, and study server-aided FL as a possible remedy. Could you comment on their findings?
3) Characterizing heterogeneity is important, as well as the study of the convergence speed of learning algorithms, depending on the amount of heterogeneity: intuitively, convergence is faster in the homogeneous setting, since there is redundancy, than in the heterogeneous setting. For instance in the paper "Permutation Compressors for Provably Faster Distributed Nonconvex Optimization"
by Szlendak et al., they introduce the notion of "Hessian variance", which characterizes the similarity between the individual loss functions of the clients. Since the losses are date-fit terms with respect to the individual datasets, this is directly related to the amount of heterogeneity. Could you comment on this aspect?

**Summary Of The Paper:**

This is a theoretical work on the analysis of the generalization error in federated learning with heterogeneity of the losses and partial participation of the clients in the training process. The authors address the question: "Would the unparticipating clients benefit from the model trained by participating clients?". They provide a statistical analysis of the generalization error. The main assumption behind is that  the local distributions are sampled from a meta-distribution.

**Summary Of The Review:**

The paper provides some new insights on partial participation, which is a timely and important problem in modern distributed learning settings. The paper studies the impact of partial participation from the mathematical/statistical point of view. These findings are interesting but there is no contribution from an algorithmic point of view, on how to mitigate the effects of partial participation.

---

> ### Author Response · Authors · 2022-11-19
> **Responses to Reviewer zYTD**
>
> Dear Reviewer zYTD, we sincerely appreciate your invaluable and constructive comments. We respond to your concerns one by one.
>
> **R: Is there a connection between your setting and the literature on personalized FL? Indeed, from the point of view of the nonparticipating clients, the obtained model is not personalized enough. I am thinking for instance of the paper Hanzely et al. "Lower Bounds and Optimal Algorithms for Personalized Federated Learning"**
>
> **A:** Please refer to the public response to all Reviewers. In this response, we have clarified the effect of heterogeneity on the unparticipating error and discussed how our results relate to personal Federated Learning. Our theoretical results are derived under the empirical risk minimization (ERM) setting, where we focus on the performance of empirical minimizer $\widehat{h} := \underset{h \in \mathcal{H}}{\operatorname{argmin}} \mathcal{L}_{S}({h})$. This paper(Hanzely et al.) provides lower bounds on the communication complexity and
> the local oracle complexity for finding $\widehat{h}$. Combining our generalization analysis with their optimization analysis is an interesting future direction.
>
> **R: I have come across the paper "With a Little Help from My Friend: Server-Aided Federated Learning with Partial Client Participation" of Yang et al., which also studies the generalization error of heterogeneous FL with partial participation, and study server-aided FL as a possible remedy. Could you comment on their findings?**
>
> **A:** We really appreciate you for giving us this related work. This paper derived a generalization error lower bound, which means that Federated Learning is not PAC-learnable under incomplete participation. Moreover, they proposed Server-Aided Federated Learning algorithms to overcome this challenge. I like this paper, to some extent this paper answered one major question we mentioned in the public response. That is the lower bound of $\mathcal{L}_P(h^*)$. It is reasonable to expect that, in the cases when every client is completely different, $\mathcal{L}_P(h^*) $ will be large. This paper proved this by providing lower bounds in the PAC-learnable framework. Also, combining our results with their lower bound is an interesting future direction.
>
> **R: Characterizing heterogeneity is important, as well as the study of the convergence speed of learning algorithms, depending on the amount of heterogeneity: intuitively, convergence is faster in the homogeneous setting, since there is redundancy, than in the heterogeneous setting. For instance in the paper "Permutation Compressors for Provably Faster Distributed Nonconvex Optimization" by Szlendak et al., they introduce the notion of "Hessian variance", which characterizes the similarity between the individual loss functions of the clients. Since the losses are date-fit terms with respect to the individual datasets, this is directly related to the amount of heterogeneity. Could you comment on this aspect?**
>
> **A:** We really appreciate you for giving us this related work. First, to use "Hessian variance" to reflect the heterogeneity is reasonable. Second, in this paper, they extend the MARINA method to the non-convex setting. MARINA is a second-order method, which relies on the Hessian matrix. However, in federated learning, it is unaffordable to compute the exact Hessian matrix. Therefore, they used a compressed method to overcome this challenge. I think it is interesting (also challenging) to extend the "Hessian variance" idea to models like neural networks. The extension to non-smooth problems is also a very important direction.

---

### Official Review · Reviewer_Uj7L · 2022-11-04

**Confidence:** 2
**Correctness:** 4
**Technical Novelty And Significance:** 3
**Empirical Novelty And Significance:** Not applicable
**Recommendation:** 8

**Clarity, Quality, Novelty And Reproducibility:**

See comments on strengths and weaknesses.


**Strength And Weaknesses:**

The paper is generally well written and clear. I appreciate the summary in Table 1. Though I did not check the details of the proof, the results look reasonable.

Generalization in federated learning is a timely topic, and the two-level distribution analysis makes a lot of sense.

I have a few “cliche” comments for a theoretical paper.

Maybe I missed it, could the authors comment more on the connection to practice. Specifically, in table 1, compared to assumptions in previous papers, do new assumptions make more sense in practice? I understand that the results themselves are interesting: Lipschitz is probably more general than bi-classifiation; Bernstein can be more general than smoothness+SC (maybe?) and a probability form rather than expectation form is shown; sub-Weibull/heavy tailed losses are analyzed for the first time for FL.

Another bonus point: what are the main insights for federated learning? – it is nice to know that the risk gaps close when clients m and samples n are increased, but anything more? For example, what is the role of heterogeneity for seen and unseen clients? For semi-excess risk, why m and n seem to play a similar role? How does the assumptions interact with two-level distribution – any common or difference compared to centralized setting?


**Summary Of The Paper:**

This paper studies the generalization property of federated learning under the two-level distribution framework: client D_i is sampled from a meta-distribution P, and sample Z_i^j is sampled from D_i. Excess risk for unseen clients and semi-excess risk for seen data are analyzed under various conditions in high probability form. For VC major class, O(1/\sqrt{mn} + 1/\sqrt{m}) rate is  established; for stronger Bernstein condition, O(1/{mn} + 1/m) is achieved; for sub-Weibull/heavy tailed losses, O(1/{mn}^{1/4} + 1/m^{1/4}) rate is achieved.


**Summary Of The Review:**

I like the topic of the paper, and the results seem to be reasonable. I would encourage authors to provide stronger motivation for analyzing generalization in FL under the proposed conditions, and the insights from these analysis. I will try to read in more detail for technical correctness and understand the difficulty of the analysis.

---

> ### Author Response · Authors · 2022-11-19
> **Responses to Reviewer Uj7L**
>
> Dear Reviewer Uj7L, we sincerely appreciate your invaluable and constructive comments. Thanks for your encouraging comments on the summary in Table 1. We respond to your concerns one by one.
>
> **R: Could the authors comment more on the connection to practice. Specifically, in table 1, compared to assumptions in previous papers, do new assumptions make more sense in practice?**
>
> **A:** This is a great question. We are happy to discuss how the assumptions made in our paper relate to practice. First, your understanding of the conditions listed in table 1 is correct. It can be verified that the commonly used binary classification losses satisfy the Lipschitz condition, such as logistic regression and SVM. In addition to classification losses, the commonly used regression losses satisfy the Lipschitz condition as well. For instance, $L_p$-loss $(p \geq 1)$ in regression problem satisfies the Lipschitz condition when the loss is bounded. Therefore, Lipschitz is more general than binary classification. Next, we discuss the relationship between Bernstein condition and Strong convexity. Bernstein condition is usually implied by the convexity of the hypothesis class and the convexity of the loss function, just as the examples we provide in Appendix. Moreover, Strong convexity and Lipschitz condition imply Bernstein condition immediately. Therefore, Bernstein condition is more general than Strong convexity. We have added a discussion on the connection to practice in the appendix. Thanks again for raising this question, which helps us improve the quality of the paper.
>
> **R: What are the main insights for federated learning? – it is nice to know that the risk gaps close when clients m and samples n are increased, but anything more?What is the role of heterogeneity for seen and unseen clients?**
>
> **A:** Please refer to the public response to all Reviewers.
>
> **R: For semi-excess risk, why m and n seem to play a similar role?**
>
> **A:** The theoretical analysis for semi-excess risk is under the assumption that the participating clients are determined. In this case, the number of participating clients $m$ is fixed. This is like multi-task learning involved with $m$ tasks. Therefore, $m \times n$ represents the total samples drawn from $m$ distributions. We believe our results can be extended to the scenario where participating clients are randomly sampled. In this time-varying participating scenario, the mixture distributions $D$ at different times can be modeled as dependent random variables(i.e., Markov chain). However, this is beyond the scope of this paper and we leave this problem for further work.
>
> **R: How does the assumptions interact with two-level distribution – any common or difference compared to centralized setting?**
>
> **A:** This is a good question. The assumptions made only on losses are orthogonal to the two-level framework(i.e., Lipschitz condition). The assumptions related to the underlying data distribution need to be carefully made in our two-level framework. In other words, this two-level framework brings an extra challenge. We tackle this challenge by carefully making different types of Bernstein condition and Small-ball condition. Moreover, we provide examples to show that the assumptions we made are mild.
>
> In a centralized setting, the participating gap and the effect of heterogeneity still persist. The challenges that we discussed in the first question(the role of heterogeneity) still persist. The difference is that, In a centralized setting, the convergence to optimal model $\widehat{h}$ will be more efficient. In other words, it is easier to find a good approximation of $ \widehat{h} $ in a centralized setting.

---

### Official Review · Reviewer_QQ79 · 2022-11-04

**Confidence:** 4
**Clarity, Quality, Novelty And Reproducibility:** Overall, the paper is well-written an…
**Correctness:** 3
**Technical Novelty And Significance:** 2
**Empirical Novelty And Significance:** Not applicable
**Recommendation:** 5

**Strength And Weaknesses:**

Strength

1. This paper used a bi-level distribution framework to model the federated learning, and assume the nonparticipating client's distribution comes from meta-distribution $P$. Then, they treat the generalization risk on nonparticipating distribution as the expected risk on a distribution sampled from $P$.  This formulation looks novel to me, even though I still have some concern about this setting which I will discuss later.

2. To get sharper risk bound, they employed seminal local Rademacher complexity based generalization analysis. To my best knowledge, in federated learning community, these techniques are new (analysis on Bernstein class, local Rademacher complexity, small ball method etc), and no one before use them to analyze FL's generalization bound.

Weakness

1. Assuming all clients' distributions obeys the same meta distribution, and assuming clients are sampled i.i.d. from this meta distribution are strong. Hence, in the risk bound for nonparticipating clients, we cannot see the impact of discrepancy between distributions used in training and target client distribution. To show the dependency of distribution discrepancy in nonparticipating client's risk bound is important, since based on that we can design domain adaptation algorithms to make nonparticipating client benefits more from collaborative learning.


2. From a technical perspective, the novelty in the proof is limited. To me this paper is more like applying existing generalization analysis of Bernstein class, and small ball condition class on federated learning. The only difference is that here we are not drawing samples from one distribution but $m$ of them. However, this does not change the game too much since it is still an unbiased sampling: we draw a multi-sample from the product of distributions, and we examine the population risk also on the product of these distributions. For examples, when the authors prove Theorem 2, they borrow the proof procedure of Theorem B.3, Theorem 9 of [1], controlling the variance of the loss function i.e., $Pf^2$, bounding the local Rademacher complexity, and relating the local class to full function class. The proof without a bounded loss setting is also similar to seminal work [2][3]. I suggest the authors can elaborate more on their technical contributions beyond these existing works [1][2][3].

Questions:

 - In Theorem 2, should not the first term in the risk bound depend on VC dimension?

- At the beginning of Page 19, the definition of $V(f) = \frac{L^2}{m}\sum_{i=1}^m \mathbb{E}(h(X) -\hat{h}^*(X) )^2$ , it seems that the index of $X$ is missing? I guess the expectation is taken over $X \sim D_i$.

- The typo right after the computation of $R(F^*,r)$: Condiser $\mapsto$ Consider

- At the beginning of Page 25, definition of $B$ norm, should not it be $\|g\|_{B}=\sup_{f\in F} |g(f)|$?

[1] Yousefi, Niloofar, et al. "Local rademacher complexity-based learning guarantees for multi-task learning." The Journal of Machine Learning Research 19.1 (2018): 1385-1431.

[2] Mendelson, Shahar. "Learning without concentration." Journal of the ACM (JACM) 62.3 (2015): 1-25.

[3] Mendelson, Shahar. "Learning without concentration for general loss functions." arXiv preprint arXiv:1410.3192 (2014).


**Summary Of The Paper:**

This paper studied generalization of federated learning. They first consider Bernstein function class, and achieve faster rate in risk bound: $O(1/mn)$ for participating clients and $O(1/mn)+O(1/m)$ for nonparticipating clients. They further proceed to the unbounded loss setting, with small-ball condition assumptions. To prove generalization in the bounded loss setting, they borrow the standard local Rademacher complexity based analysis from [1], where they controlled the variation of the loss function i.e., $Pf^2$ and relating localized class to full class. For unbounded loss setting, they borrow the technique from seminal work Learning without concentration~[2][3], where they assume a small ball condition on hypothesis class.


**Summary Of The Review:**

In summary, this paper introduces classic finer generalization analysis technique to federated learning, and does achieve better rates, but I am not sure how much the technical contribution are novel in the context of  existing works as discussed above.

---

> ### Author Response · Authors · 2022-11-19
> **Responses to Reviewer QQ79**
>
> Dear Reviewer QQ79, we sincerely appreciate your invaluable and constructive comments. We respond to your concerns one by one.
>
> **R: Assuming all clients' distributions obeys the same meta distribution, and assuming clients are sampled i.i.d. from this meta distribution are strong. Hence, in the risk bound for nonparticipating clients, we cannot see the impact of the discrepancy between distributions used in training and target client distribution. To show the dependency of distribution discrepancy in nonparticipating client's risk bound is important since based on that we can design domain adaptation algorithms to make nonparticipating client benefits more from collaborative learning.**
>
> **A:** Thank you for this question. Please refer to the public response to all Reviewers. Next, let's discuss this in more detail. We'd like to highlight that assuming all clients obey the one meta distribution is not strong. The reason is that we do not assume any property on the meta distribution, which means that meta distribution can be arbitrarily complex. In other words, meta distribution covers the case where every client is completely different from the other. We agree that having a discrepancy in unparticipating risk bound is important. We hope our public responses address your concern. Also, we add a discussion on this in Remark 1.
>
> Next, we'd like to highlight that high discrepancy in clients' distributions can be modeled as a type of "heavy-tailed" phenomenon on the level of clients(distributions). From this perspective, our analysis with heavy-tailed assumptions is related to the cases where the discrepancy of clients is high. Our results show that the convergence rate of generalization error becomes slow when a heavy-tailed phenomenon exists.
>
> **R: From a technical perspective, the novelty in the proof is limited. To me, this paper is more like applying existing generalization analysis of Bernstein class, and small ball condition class on federated learning.**
>
> **A:** Please refer to the public response to all Reviewers.
>
> **R: In Theorem 2, should not the first term in the risk bound depend on VC dimension?**
>
> **A:** Thank you for this question. The first term in the risk bound indeed depends on VC dimension. We have fixed this in the new version.
>
> **R: At the beginning of Page 19, the definition of $V(f)$, it seems that the index of X is missing? I guess the expectation is taken over $X \sim D_i.$**
>
> **A:** Thank you for raising this question. The answer to this question is yes. We have carefully checked the proofs in Appendix B and fixed typos like this.

---

> > ### Author Response · Authors · 2022-12-06
> > **Happy to provide further clarification**
> >
> > Thank you for the comments and questions. We hope that our response satisfactorily addressed all of your questions and concerns. Particularly with regards to "weakness1", we hope that our explanation of meta distribution clarified your initial concerns. We would be happy to provide further clarifications where necessary.

---

> > ### Author Response · Authors · 2022-12-10
> > **Please let us know if you have more questions / concerns.**
> >
> > Dear Reviewer QQ79,
> >
> > Thank you once again for your detailed review and valuable suggestions. We hope that our response satisfactorily addressed your questions / concerns. Given that the discussion deadline is coming, please let us know if you have more questions / concerns.

---

### Official Review · Reviewer_KoJF · 2022-11-14

**Confidence:** 3
**Correctness:** 3
**Technical Novelty And Significance:** 3
**Empirical Novelty And Significance:** Not applicable
**Recommendation:** 6

**Clarity, Quality, Novelty And Reproducibility:**

# Clarity

There are several issues with clarity.
- There seems to be an implicit assumption that $\mathcal{Y} \subset \mathbb{R}$. Does the framework handle multi-output regression or multiclass/multilabel classification?
- **Bernstein condition**: The Bernstein condition is not very interpretable. That it holds for linear regression is clear from Appendix B. Does it hold for other settings such as binary logistic regression? If yes, under what assumptions? What are the values of $B$ and $\beta$ in this case? I would suggest that these details, if proved in previous papers, can be recalled in the appendix for the reader's convenience.
- What is $\mathcal{F}$ in theorems 2, 3, 4? It is not defined.
- *Sub-root function* in Theorem 3 is not defined. It might be easier for the exposition to combine Theorems 3 and 4 into one.
- Small-ball condition: In Definition 4, the quantity $\mathbb{P}( | h| > \tau \| h\|_{L_2})$ appears. What is the probability over? I'm assuming you mean $h(X)$. But what distribution is $X$ drawn from?
- The bounds in section 4 are on $\| h_1 - h_2\|_{L_2}$. How do these translate to vanilla generalization bounds, such as the ones developed in the preceding sections?
- The definitions of $\omega$ and $\kappa$ are not interpretable. What do these terms scale like? What do you they look like for simple special cases?
- The readability of the proofs would be greatly helped with a description of the high level idea and refactoring the proofs into smaller logical units.
- There are far too many (??) in the appendix and the manuscript looks incomplete.
- Definition 7 in Appendix A is not a real definition.
- Appendix B.1 is not rigorous and can be improved. First, $X$ and $Y$ are dropped everywhere and it is hard to read. Next, the form of the Bernstein condition used here is different from the one in the main paper (and more like the one in Yousefi et al.). Equalities and inequalities in the first display of Section B.1.1 seem to be interchanged. For Section B.1.2, $\mathcal{F}$ is unclear. Further, "$\hat h^*$ is the projection of $Y$ in the space $\mathcal{F}$" is too casual. The inner product in $\langle Y - \hat h^*(X), h(X) - \hat h^*(X) \rangle$ is not defined. The $L_2$ norm in this space has not been defined either.

# Novelty
- The results appear to be somewhat straight-forward extensions of the existing literature (please correct me if I'm wrong). This in and of itself is not necessarily a negative, especially if the results give a novel perspective on the federated learning literature. However, there are one or two more steps required in translating these results into ones that make sense from a federated learning context.

# Reproducibility
- I have not had the time to carefully check the proofs. Furthermore, the fact that I cannot interpret the definitions (as mentioned previously) means that I really do not know what exactly the statements of Section 4 convey.

# Minor
- There might be some technical difficulties in defining the two-level framework in full rigor, for instance, measurability, compactness, etc.
- There are several instances of comparing bounds in expectation versus high probability, e.g. in Remark 3: "... in high probability form, which is more emergent and challenging". Could you explain this further? As far as I can tell, both convey the same information, except for technical differences.

**Strength And Weaknesses:**

I should start with the disclaimer that I only have a passing familiarity in the techniques used in this work.

**Strengths**:
- First work to consider generalization of a federated learning to clients that do not participate in learning. Assuming a meta-distribution over clients is a natural idea for cross-device federated learning that has been suggested in the literature yet a precise analysis of this generalization has eluded us.
- Studying unbounded losses in this setting is technically challenging. It is possible to sweep under the rug subtle differences across clients with bounded loss/gradient assumptions in the federated setting. Avoiding such assumptions is a plus.

**Weaknesses**:

(W1) Interpretability: The biggest weakness for me is the lack of interpretability of the results, especially in Section 4.
- The unparticipating generalization bound must show some measure of the *spread* of $P$ as a bound on the heterogeniety. In the degenerate case that every client is completely different from each other, it should not be possible to bound the unparticipating gap. Likewise, if also the clients are identical, this gap should be 0.
- My best guess is that these details are hidden in the $\kappa_m$ and $\omega_m$ objects (or in the previous sections, in the bound on the loss). It would be extremely helpful if the authors could elaborate on this dependence.
- More details on this below.

(W2) Lack of clarity: it is nearly impossible to understand this paper without a deep familiarity with a fairly advanced and niche area in learning theory. To make the paper more accessible to the federated learning community, it would be helpful to precisely define all of the mathematical objects used (at least in the appendix), point to textbooks for a condensed background (if applicable), and discuss several simple special cases such as linear regression and binary logistic regression.


**Summary Of The Paper:**

The authors study the generalization behavior of federated learning in a meta-distribution framework by characterizing two error terms: the empirical-population gap on clients that participate in training as well as the gap between participating-population at the clients level. They establish such bounds for bounded losses, but also unbounded losses for the case of least squares regression with sub-Weibull noise condition and a small ball inequality.

**EDIT**: I have raised my score from 5 to 6 after the rebuttal.

**Summary Of The Review:**

While the setting of the paper is very timely and there are many technical difficulties in the process, I feel like this paper is still a work in progress. These results look very much like intermediate results to me and are missing a final level or two of translation into statements that are more directly applicable in the federated learning setting. I would urge the authors to take steps to make the paper more accessible and specialize the results to interpretable special cases.

---

> ### Author Response · Authors · 2022-11-19
> **Responses to Reviewer KoJF (1/2)**
>
> Dear Reviewer KoJF, we sincerely appreciate your invaluable and constructive comments. We respond to your concerns one by one.
>
> **R: The unparticipating generalization bound must show some measure of the spread of P as a bound on the heterogeneity. In the degenerate case that every client is completely different from the other, it should not be possible to bound the unparticipating gap. Likewise, if also the clients are identical, this gap should be 0.**
>
> **A:** Please refer to the public response to all Reviewers. In this response, we have clarified the effect of heterogeneity on the unparticipating error and discussed how our results relate to personal Federated Learning.
>
> **R: My best guess is that these details are hidden in the $\kappa_m$ and $\omega_m$ objects (or in the previous sections, in the bound on the loss). It would be extremely helpful if the authors could elaborate on this dependence.**
>
> **A:** Thank you for raising this question. Both $\kappa_m$ and $\omega_m$ are dependent on samples drawn from meta-distribution $P$, which means that they reflect the spread of $P$ to some extent. For random variables, the bounded assumption leads to the sub-gaussian property. In our two-level distribution framework, the bounded assumption can be seen as a type of control on client heterogeneity.
>
> **R: To make the paper more accessible to the federated learning community, it would be helpful to precisely define all of the mathematical objects used (at least in the appendix), point to textbooks for a condensed background (if applicable), and discuss several simple special cases.**
>
> **A:** We thank the reviewer for suggesting this work. We have added precise definitions at the beginning of the appendix. Also, we add textbook references and discussions to ease the understanding of these definitions.
>
> **R: There seems to be an implicit assumption that $\mathcal{Y} \subset \mathbb{R}$. Does the framework handle multi-output regression or multiclass/multilabel classification?**
>
> **A:** Yes, in this paper we focus on assumption that $\mathcal{Y} \subset \mathbb{R}$. We'd like to highlight that our analysis in two-level framework is orthogonal to the setting of multi-output. Therefore, we believe that our results can be extended to multi-output problems. However, they are beyond the scope of our current paper and we leave this for further work. We have clarified this in the Introduction.
>
> **R: The Bernstein condition is not very interpretable. That it holds for linear regression is clear from Appendix B. Does it hold for other settings such as binary logistic regression? If yes, under what assumptions? What are the values of B and β in this case? I would suggest that these details if proved in previous papers, can be recalled in the appendix for the reader's convenience.**
>
> **A:** Thank you for your suggestion. In short, the Bernstein condition indicates that the variance of the loss function can be bounded by its expectation. In the analysis of generalization, the Bernstein condition is widely used to get the fast rate. Bernstein condition is usually implied by the convexity of the hypothesis class and the convexity of the loss function, just as the examples we provide in Appendix. Moreover, Strong convexity and Lipschitz condition imply Bernstein condition immediately. It can be verified that logistic regression satisfies Lipschitz condition. Therefore, $L_2$ regulized logistic regression satisfies Bernstein condition. In this case, we have $\beta=1, B = \lambda/2$, where $\lambda$ is the regularization coefficient. Also, we add references on the Bernstein condition in Appendix.
>
> **R: Sub-root function in Theorem 3 is not defined. It might be easier for the exposition to combine Theorems 3 and 4 into one.**
>
> **A:** We add the definition of the sub-root function before presenting Theorem 3. The generalization bound in Theorem 3 involves the fixed point of the sub-root function. It is non-trivial to construct a computational sub-root function. In theorem 4, we tackle this challenge using the tool of uniform entropy number. We update the proof of theorem 4 and provide more details to address this challenge.
>
> **R: The definitions of $\omega$ and $\kappa$ are not interpretable. What do these terms scale like? What do you they look like for simple special cases?**
>
> **A**: The quantity $\omega$ measures the 'size' of the localized hypothesis class based on the Rademacher complexity of the localized function set. In addition to hypothesis class $ \mathcal{H} $, $\omega$ depends on the global input samples drawn from the distribution $P$ as well. Intuitively, $ \kappa$ reflects the noise level in learning problems. For $\ell_1$ regression with boundness assumption, $\omega$ is of order at least $ \mathcal{O}(\frac{1}{n}) $, $\kappa$ is of order at least $ \mathcal{O}(\frac{1}{\sqrt{n}}) $, where $n$ is the sample size. We add an interpretation of this in Section 4.

---

> > ### Author Response · Authors · 2022-11-19
> > **Responses to Reviewer KoJF (2/2)**
> >
> > **R: Small-ball condition: In Definition 4, the quantity $\mathbb{P}\left(|h|>\tau|h| L_2\right)$ appears. What is the probability over? I'm assuming you mean $h(X)$. But what distribution is $X$ drawn from?**
> >
> > **A:** Thank you for the question. Let $X_i$ be the sample drawn from local distribution $D_i$, the probability is over $h(X_i)$.
> >
> > **R: The bounds in section 4 are on $\left|h_1-h_2\right|_{L_2}$. How do these translate to vanilla generalization bounds, such as the ones developed in the preceding sections?**
> >
> > **A:** This is a good question. For convex loss functions that satisfy the smooth condition, the bounds in the preceding sections can be bounded by the bounds in section 4. In the updated version, we add a discussion on this in Section 4.
> >
> > **R: Appendix B.1 is not rigorous and can be improved. First, $X$ and $Y$ are dropped everywhere and it is hard to read. Next, the form of the Bernstein condition used here is different from the one in the main paper (and more like the one in Yousefi et al.).**
> >
> > **A:** We thank the reviewer for this comment. We reformulate the contents in Appendix B. We thank the reviewer for pointing out the mismatch in the definition of the Bernstein condition. Our definition leads to the definition in [Yousefi et al.] if the loss function satisfies the Lipschitz condition. We update this subsection in the new version to follow our definition.
> >
> > **R: Equalities and inequalities in the first display of Section B.1.1 seem to be interchanged. For Section B.1.2, F is unclear.**
> >
> > **A:** Yes, Equalities and inequalities in the first display of Section B.1.1 are interchanged. Keep your comments on the clarity of Section B.1.2 in mind, we reformulate the contents in Appendix B.1. We add references to the expression like "projection in the space $\mathcal{F}$". Also, we add the missing definition in the new version.
> >
> > **R: What is F in theorems 2, 3, and 4? It is not defined. Definition 7 in Appendix A is not a real definition. There are (??) in the appendix.**
> >
> > **A:** Throughout the paper we use $\mathcal{F}$ to denote the family of loss functions associated with hypothesis function $\mathcal{H}$, that is, $\mathcal{F} = \{z \mapsto \ell(h, z): h \in \mathcal{H} \}$. We have added this definition at the beginning of Section 2. Also, we rewrite Definition 7 and move it to the main paper. We carefully checked the appendix and fix the typos like (??).
> >
> > **R: The readability of the proofs would be greatly helped with a description of the high-level idea and refactoring the proofs into smaller logical units.**
> >
> > **A:** We sincerely appreciate your invaluable and constructive comments, which allow us to improve the clarity of the paper. We update the appendix according to these suggestions.
> >
> > **R: The results appear to be somewhat straightforward extensions of the existing literature.**
> >
> > **A:** Please refer to the public response to all Reviewers.
> >
> > **R: There are one or two more steps required in translating these results into ones that make sense from a federated learning context.**
> >
> > **A:** Please refer to the public response to all Reviewers.
> >
> > **R: There might be some technical difficulties in defining the two-level framework in full rigor, for instance, measurability, compactness, etc.**
> >
> > **A:** Thanks for asking this question. We agree that it is important to provide a more rigorous definition based on measure theory. As a first attempt at analysis within this two-level framework, in this paper, we avoid relying on measure-theoretic definitions. For the convenience of analysis and presentation, we assume that the problems we deal with are measurable. We hope that our work can draw community attention to the generalization theory of federated learning.
> >
> > **R: There are several instances of comparing bounds in expectation versus high probability, e.g. in Remark 3: "... in high probability form, which is more emergent and challenging". Could you explain this further? As far as I can tell, both convey the same information, except for technical differences.**
> >
> > **A:** First, just as you said, extending from expectation bounds to high probability bounds is nontrivial. In particular, we provide bounds with a fast rate and results for unbounded losses. Next, let's discuss the difference between expectation and high probability. In our setting, generalization error is dependent on the data distribution. Since we only have access to finite samples drawn from the underlying distribution, the randomness of sampling will affect the generalization. The expectation bound represents the average generalization error over multiple experiments(sampling), while the probability bound represents the upper bound on the generalization error corresponding to a single experiment.

---

> > > ### Author Response · Authors · 2022-12-06
> > > **Happy to provide further clarification**
> > >
> > > Thank you for the comments and questions. We hope that our response satisfactorily addressed all of your questions and concerns. Particularly with regards to "weakness1", we hope that our explanation of the effect of heterogeneity clarified your initial concerns. We would be happy to provide more clarifications if needed.

---

> > > > ### Comment · Reviewer_KoJF · 2022-12-06
> > > > **Thanks for the authors for the detailed response | some concerns remain**
> > > >
> > > > Huge thanks to the authors for their detailed response. Several issues with the paper have been tackled but some concerns remain. I increase my score from 5 to 6 to reflect the authors' response.
> > > >
> > > > **Spread of $P$**:
> > > > Thank you for the response. The authors argue that $L_p(h^*)$ is large if the heterogeneity is large. While this is true, I do not think this fully addresses the issue. Consider mean estimation for Gaussians. The estimation error is $\sigma / \sqrt{n}$ where $\sigma$ is the standard deviation. Here, $\sigma$ measures the spread of the data. I think the un-participating generalization bounds in the federated setting should reflect the "extent" of the heterogeneity as a rough equivalent of $\sigma$.
> > > >
> > > > What do you think? Is my intuition correct? If yes, can your bounds be interpreted this way? If not, can you tell me why this intuition is wrong?
> > > >
> > > > **Logistic regression + Bernstein condition**:
> > > > Thank you for the explanation. Why don't you include this in the paper?
> > > >
> > > > **Comparing the bounds of Section 4 to the previous bounds**:
> > > > Where exactly are these comparisons? Could you write corollaries of Theorems 5/6 that are directly comparable to Theorems 2/4?
> > > >
> > > > **The restriction that $\mathcal{Y} \subset \mathbb{R}$**:
> > > > That's fine, please be upfront about it in the paper.
> > > >
> > > > **Experiments**:
> > > > Thank you for adding the experiments, this is super useful! Please proof-read the captions of the figures and the paper in general.
> > > >
> > > > **Suggestions to improve clarity/readability**:
> > > > The paper is still extremely hard to read and is dense. The technical depth of the paper is a strength but the onus to make the paper widely readable is on the authors. Personally, I think a more readable paper would have a greater reach/impact.
> > > >
> > > >
> > > > The authors can take several steps to make the main paper read like an extended summary of the key results and focus on the interpretation of the results for the federated learning setting. It would be nice to keep 2 or 3 running examples that are
> > > > - Right now, definitions 1 and 2 appear as an unhelpful blob of technical details. In general, technical definitions need to be introduced smoothly and explained or they can be given in the appendix. My opinion is that the latter is better for this case. The main paper can have a pointer to the appendix, a quick intuition behind the definition, and concrete values for the running examples.
> > > > - The remarks following the theorems can be given as an in-text description. Remark environments are helpful for small but significant pointers but it does not help for full discussions.
> > > > - Definition 3 and Assumption 2: explain the intuition behind it and instantiate them for concrete running examples introduced earlier.
> > > > - Without further details, definition 4 is not helpful. In fact, Definition 4, Theorem 3, and Remark 4 can be moved into the appendix. They do not add much useful information to the discussion and are hard to parse. Instead, it would be helpful to discuss the resulting rates in the running examples and compare them with the VC-based rates.
> > > > - Assumption 4 needs to be instantiated for the running examples. Same for definitions 6 and 7, which are currently not interpretable in terms of concrete applications. Theorems 5 and 6 can be stated in a form that is directly comparable to the previous sections (an end user might not be interested in the $L_2$ bounds presented but only cares about the loss). The $L_2$ bounds feel like intermediate results and can be given in the appendix.

---

> > > > > ### Author Response · Authors · 2022-12-09
> > > > > **Reply to Remaining Concerns (1/3)**
> > > > >
> > > > > Dear Reviewer KoJF,
> > > > >
> > > > > We sincerely appreciate your insightful comments and detailed guidance on the writing of this paper. We are deeply impressed by your comment "the onus to make the paper widely readable is on the authors". We have truly learned a lot from these helpful comments and got the importance of the wide readability of the theoretical paper.
> > > > >
> > > > > Thank you for raising your score. We feel very encouraged. Next, we respond to your remaining concerns one by one.
> > > > >
> > > > > **R1: Spread of $P.$**
> > > > >
> > > > > **A1:** This is a good question. Your intuition is correct, and the generalization bounds in our paper do reflect the "extent" of heterogeneity as a rough equivalent of $\sigma$. In short, when the loss function $\ell$ is bounded by $b$, the extent of heterogeneity is reflected by $b$. For random variables, the bounded assumption leads to the sub-gaussian property. Consider the mean estimation problem. Suppose that the variables $X_i, i=1,\cdots,n$ are independent, and $X_i$ has mean $\mu$ and $X_i \in [0,b]$. Then, for any $\delta > 0$, with probability at least
> > > > > $1-\delta,$ $$\frac{1}{n}\sum_{i=1}^n(X_i-\mu) \leq b\sqrt{\frac{\log(1/\delta)}{2n}}.$$ In our paper, Theorem 1, Theorem 2, and Theorem 4 are presented using a similar form. However, Theorem 5 and Theorem 6 are presented using a different form. Next, we explain Theorem 6 in detail to clarify this.
> > > > >
> > > > > In the analysis of unparticipating error with unbounded losses, we assume that $\xi_i h(X_i), i=1,\cdots,m$ satisfy sub-Weibull condition, where $\xi_i$ represent the noise and $X_i$ represent the random variable across the two-level framework
> > > > > $\left( \operatorname{ \mathbb{E}}[X_i] = \underset{D_i \sim P}{\operatorname{ \mathbb{E}}} \left[ \underset{X_i \sim D_i}{\operatorname{ \mathbb{E}}}
> > > > > \left[X_i\right]\right] \right).$
> > > > >
> > > > > In this case, the sub-Weibull norm
> > > > > $ || \xi_i h(X_i) ||_{\psi(\alpha)} $ (see Definition 5) reflects the spread of $P$.
> > > > >
> > > > > Theorem 6 states that, for $ 0< \iota < \frac{1}{4}$, with probability at least
> > > > > $1-\delta-\exp(-mQ_{\mathcal{H}-\mathcal{H}}^2/2) $, we have
> > > > >
> > > > > $|| \widehat{h}^*-h^* ||_{L_2(P)} \leq 2 \max \left[ \omega_m(\cdot), m^{-\frac{1}{4}+\iota} \right] ,$
> > > > > where $\delta=\exp \left[-\left(c_1 m^{4 \iota} \wedge c_2 m^{\frac{\alpha(1+4 \iota)}{2}}\right)\right].$
> > > > >
> > > > > In the current version of the paper, the terms
> > > > > $|| \xi_i h(X_i) ||_{\psi(\alpha)}$ are hidden in $\delta$.
> > > > >
> > > > > More specifically, the constents $c_1$ and $c_2$ appearing in $\delta$ depend on
> > > > > $ || \xi_i h(X_i) ||_{ \psi(\alpha) }$
> > > > > and reflect the spread of $P$.
> > > > >
> > > > > In Appendix E, the two expressions at the end of page 30 explicitly show the dependency between $\delta$ and sub-Weibull random variables
> > > > > $||\xi_i h(X_i)||_{\psi(\alpha)}.$
> > > > > Specifically, the heavier the tail of $\xi_i h(X_i)$,
> > > > > the larger $\delta$ will be.
> > > > >
> > > > > Note that in addition to $||\xi_i h(X_i)||_{\psi(\alpha)}$, $\delta$ also depends on $m^{4 \iota}$. That is, the larger $m$ and $\iota$ are, the smaller $\delta$ will be.
> > > > >
> > > > > To ensure that Theorem 6 holds with high probability, we must ensure that $\delta$ is small enough. Therefore, when $m$ is fixed,  the heavier the tail of $\xi_i h(X_i)$, the larger $\iota$ should be. It can be seen from Theorem 6 that the large $\iota$ is, the slower the convergence rate of $||\widehat{h}^*-h^*||_{L_2(P)}$ is.
> > > > >
> > > > > We sincerely thank you for raising this question. We agree that it is important to explicitly show the effect of the "extent" of heterogeneity in the main paper. In the next revision, we will show this "extent" of heterogeneity explicitly in Theorem 5 and Theorem 6. Moreover, we will add more explanations to clarify the effect of the spread of $P$.
> > > > >
> > > > > **R2: Logistic regression + Bernstein condition.**
> > > > >
> > > > > **A2:** Thank you for your suggestion. For the reader's convenience, we will include our explanations on "Logistic regression + Bernstein condition" in the next revision.

---

> > > > > > ### Author Response · Authors · 2022-12-09
> > > > > > **Reply to Remaining Concerns (2/3)**
> > > > > >
> > > > > > **R3: Comparing the bounds of Section 4 to the previous bounds.**
> > > > > >
> > > > > > **A3:** Thank you for your suggestion. We will include two corollaries of Theorems 5/6 that are directly comparable to Theorems 2/4. For quadratic loss function, Theorem 5 implies the following corollary.
> > > > > >
> > > > > > **Corollary 1:** Suppose all the assumptions of Theorem 5 hold. For $0<\iota<\frac{1}{4}$, with probability at least $1-\delta-\exp(-mnQ_{\mathcal{H}-\mathcal{H}}^2/2) $, we have
> > > > > >
> > > > > > $$ \mathcal{L}_D (\widehat{h}) - \mathcal{L}_D (\widehat{h}^*)
> > > > > > \leq c_0 \max \left[ \omega^2, (mn)^{-\frac{1-4\iota}{2}}
> > > > > > \right]$$
> > > > > >
> > > > > > where $ \delta = \exp \left[-\left(c_1 (mn)^{4 \iota} \wedge c_2 (mn)^{\frac{\alpha(1+4 \iota)}{2}}\right)\right],$ $c_0$ depends on the constants $\tau$ and $Q_{\mathcal{H}}(\tau)$ defined in Assumption 4.
> > > > > >
> > > > > > Corollary 1 can be derived by following similar steps in the proof of Theorem 2.3 of [1]. Theorem 2.3 of [1] states that for convex loss functions that satisfy the smooth condition, if on one event,
> > > > > > $ || \widehat{h} - \widehat{h}^* ||_{L_2(D)} \leq \max [ \omega, \kappa ]$,
> > > > > > then on the same event one has $\mathcal{L}_D(\widehat{h})-\mathcal{L}_D\left(\widehat{h}^*\right) \lesssim \max[ \omega^2, \kappa^2 ].$
> > > > > >
> > > > > > For sub-Weibull losses, Corollary 1 shows that the convergence rate of semi-empirical excess risk is slower than $\mathcal{O}\left(\frac{1}{\sqrt{m n}}\right)$.
> > > > > >
> > > > > > **Corollary 2:** Suppose all the assumptions of Theorem 5 and Theorem 6 hold. For $0<\iota<\frac{1}{4}$, with probability at least $1-\delta-\delta^\prime- \exp(-mQ_{\mathcal{H}-\mathcal{H}}^2(\tau, P)/2) - \exp(-mQ_{\mathcal{H}-\mathcal{H}}^2(\tau^\prime, P)/2) $, we have
> > > > > >
> > > > > > $$\mathcal{L}_P (\widehat{h})-\mathcal{L}_P ({h}^*) \leq c_0 \max [\omega^2(\cdot), (mn)^{-\frac{1-4\iota}{2}}] + c_0^\prime \max [ \omega_m^2(\cdot), m^{-\frac{1-4\iota}{2}}], $$
> > > > > >
> > > > > > where $\delta=\exp \left[-\left(c_1 (mn)^{4 \iota} \wedge c_2 (mn)^{\frac{\alpha(1+4 \iota)}{2}}\right) \right], $ $\delta^\prime=\exp \left[-\left(c_1^\prime (m)^{4 \iota} \wedge c_2^\prime (m)^{\frac{\alpha(1+4 \iota)}{2}}\right) \right], $ $c_0$ and $c_0^\prime$ are constants depending on $\tau,$ $Q_{\mathcal{H}}(\tau),$ and $Q_{\mathcal{H}}(\tau,P)$, $\tau^\prime$ is a well-chosen value depending on the small-ball constants defined in Assumption 4.
> > > > > >
> > > > > > Since excess risk is defined across our two-level framework, the steps in the proof of Corollary 1 can not be applied directly to derive Corollary 2. The key step to derive Corollary 2 is to bound $||\widehat{h}-h^*||_{L_2(P)}^2. $
> > > > > >
> > > > > > First, this term can be decomposed as $
> > > > > > || \widehat{h} - h^* ||_{L_2(P)}^2  \leq 2T_1 + 2T_2,
> > > > > > $
> > > > > >
> > > > > > where $T_1 = ||\widehat{h}-\widehat{h}^*||_{L_2(P)}^2,$
> > > > > >
> > > > > > $T_2 = ||\widehat{h}^* - {h}^*||_{L_2(P)}^2.$
> > > > > >
> > > > > > Note that $T_2$ has been bounded by Theorem 6. To bound $T_1$, we use the following decomposition:
> > > > > >
> > > > > > $$T_1 = T_1 - c_0 T_3 + c_0 T_3 .$$
> > > > > >
> > > > > > where $T_3 = ||\widehat{h}-\widehat{h}^*||_{L_2(D)}^2.$
> > > > > >
> > > > > > Note that $T_3$ has been bounded by Theorem 5. By following the similar steps of proof of Theorem 6, it can be proved that with probability at least $1- \exp(-mQ_{\mathcal{H}-\mathcal{H}}^2(\tau^\prime, P)/2),$ one has
> > > > > > $T_1 - c_0 T_3 \leq 0,$ where $\tau^\prime$ is a well-chosen value depending on the small-ball constants defined in Assumption 4. The remaining steps are similar to those of Corollary 1. Compared to Theorem 4, Corollary 2 shows that the convergence rate of excess risk is slower than $\mathcal{O}\left(\frac{1}{\sqrt{m n}}+\frac{1}{\sqrt{m}}\right)$.
> > > > > >
> > > > > > Corollary 1 and Corollary 2 provide the first results on the generalization error of heterogeneous federated learning with heavy-tailed losses. Compared to Theorem 2/4, the convergence rates in Corollary 1/2 are slow. This is reasonable since the spread of heavy-tailed data is large.
> > > > > >
> > > > > > Thanks again for your suggestion. We agree that it is important to include corollaries of Theorems 5/6 that are directly comparable to previous sections. In the next revision, we will include these two corollaries.
> > > > > >
> > > > > > **R4: The restriction that $Y\subset \mathbb{R}.$**
> > > > > >
> > > > > > **A4:** Thank you for your suggestion. We will make sure to clarify this where necessary.
> > > > > >
> > > > > > **R5: Experiments.**
> > > > > >
> > > > > > **A5:** Thank you for your positive feedback. We feel very encouraged. In the next reversion, we will further carefully proof-read and revise this paper.

---

> > > > > > > ### Author Response · Authors · 2022-12-09
> > > > > > > **Reply to Remaining Concerns (3/3)**
> > > > > > >
> > > > > > > **R6: Suggestions to improve clarity/readability.**
> > > > > > >
> > > > > > > **A6:** Thank you for your suggestion. To improve the readability of the paper, we will add several running examples (i.e., linear regression) in our next revision. Moreover, we will provide more interpretations of our theoretical results in the federated learning setting. In details:
> > > > > > >
> > > > > > > - We will move definition 1, definition 2, and definition 4 into the appendix. In the main paper, we will use linear regression to provide an intuitive explanation of definitions 1-3 and Assumption 2. Moreover, we will add a discussion to compare the fast rates and slow rates.
> > > > > > > - Thank you for your detailed guidance. We will turn the full discussions following the theorems into in-text descriptions.
> > > > > > > - Thank you for your suggestion. We will use Lasso regression to explicitly show the concrete values defined in Assumption 4, definitions 6 and 7.
> > > > > > >
> > > > > > > [1] Mendelson, Shahar. "Learning without Concentration for General Loss Functions[J]." Probability Theory & Related Fields, 2018, 171(4):1-44.

---

### Author Response · Authors · 2022-11-19
**General Response to All Reviewers (1/3)**

We thank the reviewers for the time spent reviewing our work, their thorough review, and their interest in our work. We'll answer some common questions here.

## Insights for federated learning

- In Section 3, we derive fast learning bounds of order $\mathcal{O}(\frac{1}{mn} + \frac{1}{m})$ for unparticipating clients under the assumption of Bernstein condition. We recall that in Section 2, slow learning bounds of order $ \mathcal{O}(\frac{1}{\sqrt{mn}} + \frac{1}{\sqrt{m}}) $ are obtained without using Bernstein condition. When we fix $ m $ and $n$, the fast rates indicate that the corresponding generalization error is smaller. In other words, using losses satisfying Bernstein condition will lead to lower generalization error for the same $ m $ and $n$.

- In Section 4, we analyze the generalization error for participating and unparticipating clients with unbounded losses. The results show that the learning bounds will become slow when the losses are heavy-tailed. We highlight that assuming the data are unbounded or heavy-tailed is more practical. We believe that our theoretical results will promote an understanding of a generalization of federated learning.

- In the first public response, we clarify the effect of heterogeneity on the unparticipating error and discussed how our results relate to personal Federated Learning. These observations indicate that we can not expect one common model works well when heterogeneity is high. Thus, personal federated learning is an emerging and important paradigm.

## Novelty of the proof

First, we'd like to highlight that the objective of this paper is to provide theoretical results in the two-level framework. We believe that our theoretical results will inspire both theoretical and experimental researchers in the federated learning community. We elaborate on our technical contributions beyond these existing works in the following.

- It is the first time that small-ball methods are used in the analysis of federated learning. Moreover, the extension of small-ball methods in our two-level framework is non-trivial. To overcome the extra challenges caused by our framework, we propose different definitions of small-ball conditions. Moreover, we derived generalization bounds for participating and unparticipating clients.

- Most existing theoretical results based on local Rademacher complexity methods are under i.i.d assumption. Our results are derived in the two-level framework and non-i.i.d setting. The results under the non-i.i.d setting in [1] are derived for kernel-based methods. Our results can be applied to more interesting hypothesis classes that have uniform entropy numbers.

- We analyze heavy-tailed problems in our framework, which are not covered by [2] and [3]. Moreover, this is the first analysis of the generalization of federated learning with heavy-tailed losses.

[1] Yousefi, Niloofar, et al. "Local rademacher complexity-based learning guarantees for multi-task learning." The Journal of Machine Learning Research 19.1 (2018): 1385-1431.

[2] Mendelson, Shahar. "Learning without concentration." Journal of the ACM (JACM) 62.3 (2015): 1-25.

[3] Mendelson, Shahar. "Learning without concentration for general loss functions." arXiv preprint arXiv:1410.3192   Add to Citavi project by ArXiv ID(2014).

---

> ### Author Response · Authors · 2022-11-19
> **General Response to All Reviewers (2/3)**
>
> ## The effect of heterogeneity of clients on the generalization error (1/2)
> In Section 2 and Section 3, we focus on the excess risk defined as
> $$\mathcal{L}_P (\widehat{h})-\mathcal{L}_P(h^*) =  \underset{D_i \sim P}{\operatorname{ \mathbb{E}}}   \underset{Z \sim D_i}{\operatorname{ \mathbb{E}}} [\ell(\widehat{h}, Z)]] - \underset{h \in \mathcal{H}}{\operatorname{ \inf}} \underset{D_i \sim P}{\operatorname{ \mathbb{E}}}   \underset{Z \sim D_i}{\operatorname{ \mathbb{E}}} [\ell(h, Z)]] .$$
> Our theoretical results show that under some assumptions we are able to bound the excess risk $\mathcal{L}_P(\widehat{h})-\mathcal{L}_P(h^*)$. Note that the generalization error for unparticipating clients is $\mathcal{L}_P(\widehat{h})$. Thus, the results like $\mathcal{L}_P(\widehat{h})-\mathcal{L}_P(h^*) \leq \mathcal{O}(\frac{1}{mn} + \frac{1}{m})$ only indicate that
> $\mathcal{L}_P(\widehat{h}) \leq \mathcal{L}_P(h^*) + \mathcal{O}(\frac{1}{mn} + \frac{1}{m}).$
> One natural question is how the heterogeneity of clients affects the generalization error bounds. In short, for bounded losses, the heterogeneity of clients will affect $\mathcal{L}_P(h^*)$. In the cases when every client is completely different, $\mathcal{L}_P(h^*) $ will be large.
>
> This can be better understood by comparing it with personal federated learning. Instead of considering the common optimal model $h^*$ defined as $ h^* = \underset{h \in \mathcal{H}}{\operatorname{argmin}} \mathcal{L}_{P}({h}).$
>
> Personal federated learning focus on the personal optimal models that vary with distributions(clients). Let $\mathcal{L}_P(\mathcal{H}) := \underset{D_i \sim P}{\operatorname{ \mathbb{E}}} \left[\underset{h \in \mathcal{H}}{\operatorname{ \inf}}  \underset{Z \sim D_i}{\operatorname{ \mathbb{E}}}
> \left[
> \ell(h, Z)\right]\right].$
> The personal excess risk in personal federated learning is defined as:
> $$\mathcal{L}_P(\widehat{h}) - \mathcal{L}_P(\mathcal{H}) = \underset{D_i \sim P}{\operatorname{ \mathbb{E}}} [\underset{Z \sim D_i}{\operatorname{ \mathbb{E}}}[\ell(\widehat{h}, Z)]] - \underset{D_i \sim P}{\operatorname{ \mathbb{E}}} \left[[\underset{h \in \mathcal{H}}{\operatorname{ \inf}} \underset{Z \sim D_i}{\operatorname{ \mathbb{E}}}\left[
> \ell(h, Z)\right]\right].$$
> The relationship between personal excess risk and the excess risk defined in this paper can be shown by the following decomposition:
> $\mathcal{L}_P(\widehat{h})- \mathcal{L}_P(\mathcal{H}) = \mathcal{L}_P(\widehat{h}) -\mathcal{L}_P(h^*) +\mathcal{L}_P(h^*) - \mathcal{L}_P(\mathcal{H})$. In this paper, we focus on the term $\mathcal{L}_P(\widehat{h}) -\mathcal{L}_P(h^*).$ The term $$\mathcal{L}_P(h^*) - \mathcal{L}_P(\mathcal{H})=\underset{h \in \mathcal{H}}{\operatorname{ \inf}} \underset{D_i \sim P}{\operatorname{ \mathbb{E}}}[\underset{Z \sim D_i}{\operatorname{ \mathbb{E}}}[\ell({h}, Z)]] - \underset{D_i \sim P}{\operatorname{ \mathbb{E}}}\left[\underset{h \in \mathcal{H}}{\operatorname{ \inf}}\underset{Z \sim D_i}{\operatorname{ \mathbb{E}}}\left[\ell(h, Z)\right]\right]$$ measures the spread of meta-measure $P$. If all clients are identical, $\mathcal{L}_P(h^*) - \mathcal{L}_P(\mathcal{H})$ will be zero. When the clients are quite different, $\mathcal{L}_P(h^*) - \mathcal{L}_P(\mathcal{H})$ will be large. These observations indicate that we can not expect one common model works well when heterogeneity is high. Thus, personal federated learning is an emerging and important paradigm.

---

> > ### Author Response · Authors · 2022-11-19
> > **General Response to All Reviewers (3/3)**
> >
> > ## The effect of heterogeneity of clients on the generalization error (2/2)
> >
> > For participating clients, we focus on the semi-excess risk defined as
> > $$\mathcal{L}_D(\widehat{h})-\mathcal{L}_D(\widehat{h}^*)= \frac{1}{m} \underset{i \in [m]}{\operatorname{ \sum}} \underset{Z \sim D_i}{\operatorname{ \mathbb{E}}}[\ell(\widehat{h}, Z)] - \underset{h \in \mathcal{H}}{\operatorname{ \inf}}\frac{1}{m} \underset{i \in [m]}{\operatorname{ \sum}}\underset{Z \sim D_i}{\operatorname{ \mathbb{E}}}[\ell({h}, Z)].$$
> > Our theoretical results show that under some assumptions we can bound the excess risk $\mathcal{L}_D(\widehat{h})-\mathcal{L}_D(\widehat{h}^*)$. That is
> > $$\mathcal{L}_D(\widehat{h}) \leq \mathcal{L}_D(\widehat{h}^*) + \mathcal{O}(\frac{1}{mn}).$$
> > Thus, the heterogeneity of clients will affect $\mathcal{L}_D(\widehat{h}^*)$. In the cases when participating clients are quite different, $\mathcal{L}_D(\widehat{h}^*) $ will be large.
> >
> > This can be better understood by comparing it with personal federated learning. Instead of considering the common optimal model $\widehat{h}^*$ defined as $ \widehat{h}^* = \underset{h \in \mathcal{H}}{\operatorname{ argmin}} \mathcal{L}_{D}({h}).$ Personal federated learning focus on the personal optimal models that vary with distributions(clients). Let $\mathcal{L}_D(\mathcal{H}) := \frac{1}{m} \underset{i \in [m]}{\operatorname{ \sum}} \underset{h \in \mathcal{H}}{\operatorname{ \inf}}\underset{Z \sim D_i}{\operatorname{ \mathbb{E}}}\left[\ell(h, Z)\right].$ The personal semi-excess risk in personal federated learning is defined as:
> > $$\mathcal{L}_D(\widehat{h})-\mathcal{L}_D(\mathcal{H})=\frac{1}{m} \underset{i \in [m]}{\operatorname{ \sum}} \underset{Z \sim D_i}{\operatorname{ \mathbb{E}}}[\ell(\widehat{h}, Z)] - \frac{1}{m} \underset{i \in [m]}{\operatorname{ \sum}} \underset{h \in \mathcal{H}}{\operatorname{ \inf}}\underset{Z \sim D_i}{\operatorname{ \mathbb{E}}}\left[\ell(h, Z)\right].$$
> > The relationship between personal semi-excess risk and the semi-excess risk defined in this paper can be shown by the following decomposition:
> > $\mathcal{L}_D(\widehat{h})- \mathcal{L}_D(\mathcal{H}) = \mathcal{L}_D(\widehat{h}) -\mathcal{L}_D(\widehat{h}^*) +\mathcal{L}_D(\widehat{h}^*) - \mathcal{L}_D(\mathcal{H})$. In this paper, we focus on the term $\mathcal{L}_D(\widehat{h}) -\mathcal{L}_D(\widehat{h}^*).$ The term $$\mathcal{L}_D(\widehat{h}^*) - \mathcal{L}_D(\mathcal{H})=\underset{h \in \mathcal{H}}{\operatorname{ \inf}}\frac{1}{m} \underset{i \in [m]}{\operatorname{ \sum}}\underset{Z \sim D_i}{\operatorname{ \mathbb{E}}}[\ell({h}, Z)] - \frac{1}{m} \underset{i \in [m]}{\operatorname{ \sum}} \underset{h \in \mathcal{H}}{\operatorname{ \inf}}\underset{Z \sim D_i}{\operatorname{ \mathbb{E}}}\left[\ell(h, Z)\right]$$ measures the spread of semi-empirical distribution $D=\frac{1}{m} \underset{i \in [m]}{\operatorname{ \sum}} D_i$. If all participating clients are identical, $\mathcal{L}_D(\widehat{h}^*) - \mathcal{L}_D(\mathcal{H})$ will be zero. When the participating clients are quite different, $\mathcal{L}_D(\widehat{h}^*) - \mathcal{L}_D(\mathcal{H})$ will be large.

---

### Author Response · Authors · 2022-11-19
**Paper Revision**

We sincerely appreciate all reviewers for their invaluable and constructive comments. **We hope that we have addressed the reviewer’s concerns. We would be happy to answer any questions and discuss further in case the reviewer believes there are any missing details.** We have carefully revised the manuscript based on the initial reviews. The following is a summary of major changes:

- We add experimental results on the MNIST dataset and synthetic data. We use CNN for MINIST task(see Figure 2 and Figure 3), which allows us to check the validity of our theory for over-parameterized models. Moreover, we add numerical experiments based on the linear regression model(see Figure 4).
- Appendix C, Appendix D and Appendix E: (1)We reformulate the contents in Appendix. (2)we carefully checked the appendix and fix the typos. (3)We refactor the proofs into smaller logical units.
- We add a discussion on how client heterogeneity affects the generalization error(see Remark 2).

---

### Author Response · Authors · 2022-11-23
**Response to Reviewers**

Dear reviewers:

Do my answers address your concerns? If you don't think so, please point out and let me know. I'd be happy to answer any of your concerns.

---

### Decision · Program_Chairs · 2023-01-20

**Decision:**

Accept: poster

**Justification For Why Not Higher Score:**

Clarity issues.

**Justification For Why Not Lower Score:**

I believe that the 7 reviews, many of which are very detailed and technical, give a strong signal towards (weak) acceptance.

**Metareview: Summary, Strengths And Weaknesses:**

The authors study the generalization behavior of federated learning in a meta-distribution framework by characterizing two error terms: the empirical-population gap on clients that participate in training as well as the gap between participating-population at the clients level. They establish such bounds for bounded losses, but also unbounded losses for the case of least squares regression with sub-Weibull noise condition and a small ball inequality.

The paper received 7 reviews, with scores 3, 5, 6, 6, 6, 6, 8. The views were mixed, but overall I am inclined to suggest an acceptance subject to requiring the authors to carefully revise their paper for the camera ready submission, with a particular emphasis on the criticism raised by the reviewers who viewed the work negatively. In reaching this decision, I took into account the silence of some reviewers after the authors responded to the raised criticism, who I believe did so in a reasonable way.

Some of the issued raised by the reviewers:

- Some of the results appear to be somewhat straight-forward extensions of the existing literature.
- Numerous clarity issues were raised.

In addition, the authors sometimes responded with factually incorrect answers (e.g., it is not true that MARINA is a second order method - this is trivial to see, the authors clearly did not read the paper).

Some of the positives:

- This seems to be the first time that small-ball methods are used in the analysis of federated learning. Moreover, the extension of small-ball methods in the two-level framework studied here seems involved.
- Fast learning bounds for non-participating clients under the Bernstein assumption.
- Analysis of the generalization error for participating and non-participating clients with unbounded losses. The results show that the learning bounds will become slow when the losses are heavy-tailed.

AC

**Note From Pc:**

if the above contains the word "oral" or "spotlight" please see: "oral" presentation means -> notable-top-5% and "spotlight" means -> notable-top-25%. As stated in our emails, we are disassociating presentation type from AC recommendations